# Opponent Modeling with In-context Search

**Yuheng Jing**[1,2] **Bingyun Liu**[1,2] **Kai Li**[1,2,†] **Yifan Zang**[1,2]
**Haobo Fu**[6] **Qiang Fu**[6] **Junliang Xing**[5] **Jian Cheng**[1,3,4,†]
[†] denotes corresponding authors
[1] Institute of Automation, Chinese Academy of Sciences
[2] School of Artificial Intelligence, University of Chinese Academy of Sciences
[3] School of Future Technology, University of Chinese Academy of Sciences
[4] AiRiA    [5] Tsinghua University    [6] Tencent AI Lab
{jingyuheng2022,liubingyun2021,kai.li,zangyifan2019,jian.cheng}
@ia.ac.cn, {haobofu,leonfu}@tencent.com, jlxing@tsinghua.edu.cn

## Abstract

Opponent modeling is a longstanding research topic aimed at enhancing decision-making by modeling information about opponents in multi-agent environments. However, existing approaches often face challenges such as having difficulty generalizing to unknown opponent policies and conducting unstable performance. To tackle these challenges, we propose a novel approach based on in-context learning and decision-time search named **O**pponent **M**odeling with **I**n-context **S**earch (**OMIS**). OMIS leverages in-context learning-based pretraining to train a Transformer model for decision-making. It consists of three in-context components: an actor learning best responses to opponent policies, an opponent imitator mimicking opponent actions, and a critic estimating state values. When testing in an environment that features unknown non-stationary opponent agents, OMIS uses pretrained in-context components for decision-time search to refine the actor's policy. Theoretically, we prove that under reasonable assumptions, OMIS without search converges in opponent policy recognition and has good generalization properties; with search, OMIS provides improvement guarantees, exhibiting performance stability. Empirically, in competitive, cooperative, and mixed environments, OMIS demonstrates more effective and stable adaptation to opponents than other approaches. See our project website at `https://sites.google.com/view/nips2024-omis`.

## 1 Introduction

**O**pponent **M**odeling (**OM**) is a pivotal topic in artificial intelligence research, aiming to develop autonomous agents capable of modeling the behaviors, goals, beliefs, or other properties of *adversaries or teammates* (collectively termed as ***opponents***). Such modelings are used to reduce uncertainty in multi-agent environments and enhance decision-making [4, 59, 101, 28, 107, 63, 53, 92, 105, 17, 102, 66, 68]. Despite the methodologies and insights proposed by existing OM approaches, their processes generally boil down to two stages: (1) **Pretraining**: pretrain a model with designed OM methodology on a training set of opponent policies; (2) **Testing**: deploying the pretrained model in a certain way on a testing set of opponent policies to benchmark adaptability to unknown opponents.

For these two processes, different OM approaches usually have their respective focuses: (1) **P**retraining-**F**ocused **A**pproach (**PFA**) [29, 28, 63, 107] focuses on acquiring knowledge of responding to various opponents during pretraining and generalizing it to the testing stage; (2) **T**esting-**F**ocused **A**pproach (**TFA**) [3, 41, 101] focuses on updating the pretrained model during testing to reason and respond to unknown opponents effectively. However, existing PFAs and TFAs have their respective common and noteworthy drawbacks. For PFAs, they have *limited generalization abilities*, as the

38th Conference on Neural Information Processing Systems (NeurIPS 2024).

generalization of their pretrained models often lacks theoretical guarantees. Moreover, PFAs typically involve minimal additional operations during the testing stage, making them practically challenging to handle unknown opponents. For TFAs, they have *performance instability issues*. The *finetuning* (*i.e.*, update the pretrained model) of TFAs during testing can be tricky, as it involves several gradient updates using only a few samples to adjust the policy. Without careful manual hyperparameter tuning, TFAs always perform unstably when facing unknown opponents.

To overcome the inherent issues of PFAs and TFAs, we propose a novel approach named **O**pponent **M**odeling with **I**n-context **S**earch (**OMIS**). The core motivation behind OMIS is '*think before you act*': when facing an opponent with an unknown policy during testing, we first guess about his current policy based on historical context. We then conduct **D**ecision-**T**ime **S**earch (**DTS**) for a few steps using this imagined opponent policy, estimate the returns of each legal action, and choose the best one to act on. Such a process intuitively helps derive a policy more optimal than making a direct decision regarding only the current state. This approach is often reflected in real-life situations, such as the 'deep thinking' strategy employed by professional players in Go, chess, and other board games.

To enable such a DTS, we build three components: an *actor*, to respond appropriately to the current opponent during the DTS; an *opponent imitator*, who imitates the actions of the current opponent, enabling the generation of transitions in an imagined environment during the DTS; a *critic*, who estimates the value of the final search states, as we do not search until the end of the game. We argue that all three components should be *adaptive*, meaning they dynamically adjust based on changes in opponent information. Therefore, we adopt **I**n-**C**ontext **L**earning (**ICL**)-based pretraining to learn *three in-context components*, as ICL can endow them with the needed adaptability.

In summary, the methodology design of OMIS is as follows: (1) For Pretraining, we train a Transformer [84] model for decision-making based on ICL [20, 88, 57, 43]. We build our model with three components: an actor, who learns the best responses to various opponent policies; an opponent imitator, who imitates opponent actions; and a critic, who estimates state values; (2) For Testing, we use the pretrained three in-context components for DTS [80, 81, 13] to refine the actor's original policy. Based on predicting opponent actions and estimating state values, this DTS performs rollouts for each legal action, promptly evaluating and selecting the most advantageous action.

Theoretically, OMIS can provably alleviate the issues present in PFAs and TFAs. For *limited generalization ability* of PFAs, OMIS's pretrained model is proven to converge on opponent policy recognition and to have *good generalization properties*: OMIS's pretrained model can accurately recognize seen opponents and recognize unseen opponents as the most familiar seen ones to some extent. For *performance instability issues* of TFAs, OMIS's DTS avoids any gradient updates and theoretically provides improvement guarantees.

Empirically, extensive comparative experiments and ablation analyses in competitive, cooperative, and mixed environments verify the effectiveness of OMIS in adapting to unknown non-stationary opponent agents. Statistically, OMIS demonstrates better performance and lower variance during testing than other approaches, reflecting the generalizability and stability of opponent adaptation.

## 2 Related Work

**Opponent modeling.** In recent years, OM has seen the rise of various new approaches based on different methods, including those based on representation learning [29, 28, 107, 63, 40], Bayesian learning [106, 19, 24, 54], meta-learning [3, 41, 107, 94], shaping opponents' learning [22, 23, 47], and recursive reasoning [90, 101]. All approaches can be broadly categorized into PFAs and TFAs.

OM based on representation learning and meta-gradient-free meta-learning methods such as Duan et al. [20] typically fall into PFAs. PFAs' generalization on unknown opponents often lacks any theoretical analysis or guarantees. This also leads to PFAs not always performing well empirically. Our work utilizes ICL pretraining to provide good theoretical properties regarding generalization.

OM based on Bayesian learning and meta-gradient-based meta-learning such as Finn et al. [21] typically belong to TFAs. The finetuning of TFAs makes them unstable, as the opponent may continuously change policy during testing, making it challenging to adapt with a small number of samples for updating. Our work employs DTS to avoid finetuning and has improvement guarantees.

**In-context learning.** Algorithmically, ICL can be considered as taking a more agnostic approach by learning the learning algorithm itself [20, 88, 57, 43]. Recent work investigates why and how

pretrained Transformers perform ICL [27, 49, 103, 1, 69]. Xie et al. [95] introduces a Bayesian framework explaining how ICL works. Some work [87, 2, 8] proves Transformers can implement ICL algorithms via in-context gradient descent. Lee et al. [44] proposes supervised pretraining to empirically and theoretically demonstrate ICL abilities in decision-making. Unlike existing decision-related work focusing on single-agent settings, our work explores the theoretical properties and empirical effects of using a Transformer pretrained based on ICL under the setting of OM.

**Decision-time search.** DTS involves searching in a simulated environment before each real action, aiming to obtain a more 'prescient' policy than no search [80, 81, 13, 50]. One of the most representative works is the AlphaGo series [74–76, 72], which achieves remarkable results in games like Go and Atari based on a DTS algorithm called *Monte Carlo Tree Search* (MCTS) and self-play. Our work explores how to make DTS work in the context of OM. The DTS of the AlphaGo series assumes that opponent adopts the same strong policy as the agent we control. In contrast, the DTS in our work dynamically models the opponents' actions, focusing on better adapting to the current opponents.

See App. A for an overview of OM and related work on Transformers for decision-making.

## 3 Preliminaries

We formalize the multi-agent environment using an $n$-agent stochastic game $\langle \mathcal{S}, \{\mathcal{A}^i\}_{i=1}^n, \mathcal{P}, \{R^i\}_{i=1}^n, \gamma, T\rangle$. $\mathcal{S}$ is the state space, $\mathcal{A}^i$ is the action space of agent $i \in [n]$, $\mathcal{A} = \prod_{i=1}^n \mathcal{A}^i$ is the joint action space of all agents, $\mathcal{P} : \mathcal{S} \times \mathcal{A} \times \mathcal{S} \to [0, 1]$ is the transition dynamics, $R^i : \mathcal{S} \times \mathcal{A} \to \mathbb{R}$ is the reward function for agent $i$, $\gamma$ is the discount factor, and $T$ is the horizon for each episode.

Following the tradition in OM, we mark the agent under our control, *i.e.*, the *self-agent*, with the superscript 1, and consider the other $n - 1$ agents as *opponents*, marked with the superscript $-1$. The joint policy of opponents is denoted as $\pi^{-1}(a^{-1}|s) = \prod_{j\neq 1} \pi^j(a^j|s)$, where $a^{-1}$ is the joint actions of opponents. Let the trajectory at timestep $t$ in the current episode be $y_t^{(\text{cur})} = \{s_0, a_0^1, a_0^{-1}, r_0^1, r_0^{-1}, \ldots, s_{t-1}, a_{t-1}^1, a_{t-1}^{-1}, r_{t-1}^1, r_{t-1}^{-1}, s_t\}$. The *historical trajectories* $\mathcal{H}_t := (y_T^{(0)}, \ldots, y_T^{(\text{cur}-1)}, y_t^{(\text{cur})})$ is always available to the self-agent. During the pretraining stage, opponent policies are sampled from a *training set of opponent policies* $\Pi^{\text{train}} := \{\pi^{-1,k}\}_{k=1}^K$. During the testing stage, opponent policies are sampled from a *testing set of opponent policies* $\Pi^{\text{test}}$, which includes an unknown number of unknown opponent policies.

In OM, the self-agent's policy can be generally denoted as $\pi^1(a^1|s, D)$ (abbreviated as $\pi$), which dynamically adjusts based on the *opponent information data* $D$ (referred to as **in-context data** in this paper). $D$ can be directly composed of some part of the data from $\mathcal{H}_t$, or it can be obtained by learning a representation from $\mathcal{H}_t$. Building upon the pretraining, the objective of the self-agent is to maximize its expected *return* (*i.e.*, cumulative discounted reward) during testing:

$$\mathbb{E}_{\substack{s_{t+1}\sim\mathcal{P}(\cdot|s_t,a_t^1,a_t^{-1}),a_t^{-1}\sim\pi^{-1}(\cdot|s_t),\\ \pi^{-1}\sim\Pi^{\text{test}},a_t^1\sim\pi(\cdot|s_t,D),\\ D\sim\mathcal{H}_t,\pi\sim\text{Pretraining}(\Pi^{\text{train}})}} \left[ \sum_{t=0}^{T-1} \gamma^t \cdot R^1(s_t, a_t^1, a_t^{-1}) \right]. \tag{1}$$

## 4 Methodology

In Sec. 4.1, we present how we build the in-context actor, opponent imitator, and critic for OMIS with ICL-based pretraining; in Sec. 4.2, we describe our method of using pretrained in-context components for DTS; in Sec. 4.3, we provide a theoretical analysis of both the ICL and DTS components of OMIS. We provide an overview of OMIS in Fig. 1 and the pseudocode of OMIS in App. B.

### 4.1 In-Context-Learning-based Pretraining

To ensure that the actor learns high-quality knowledge of responding to various opponents, we first solve for the *Best Responses* (BR) against different opponent policies. For each opponent policy $\pi^{-1,k}$ in $\Pi^{\text{train}}$ (where $k \in [K]$), we keep the opponent policy fixed as $\pi^{-1,k}$ and sufficiently train the PPO algorithm [73] to obtain the BR against $\pi^{-1,k}$, denoted as $BR(\pi^{-1,k}) := \pi^{1,k,*}(a|s)$.

To generate training data for pretraining the three components, we continually sample opponent policies from $\Pi^{\text{train}}$ and use their corresponding BR to play against them. For each episode, we sample a $\pi^{-1,k}$ from $\Pi^{\text{train}}$ as opponents and use its BR $\pi^{1,k,*}$ as self-agent to play against it.

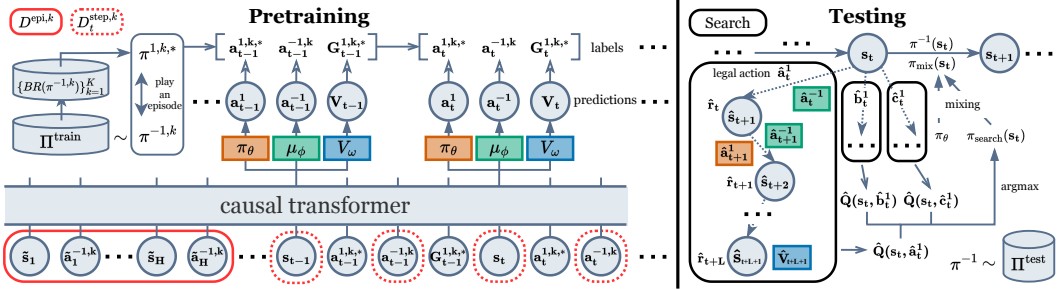

Figure 1: **Left:** The pretraining procedure and architecture of OMIS. The pretraining steps are as follows: (1) Train BRs against all policies in $\Pi^{\text{train}}$. (2) Continuously sample opponent policy from $\Pi^{\text{train}}$ and collect training data by playing against it using its BR. (3) Train a Transformer model using ICL-based supervised learning, where the model consists of three components: an actor $\pi_\theta$, an opponent imitator $\mu_\phi$, and a critic $V_\omega$. **Right:** The testing procedure of OMIS. During testing, OMIS refines $\pi_\theta$ through DTS at each timestep. The DTS steps are as follows: (1) Do multiple $L$-step rollouts for each legal action, where $\pi_\theta$ and $\mu_\phi$ are used to simulate actions for the self-agent and opponent, respectively. $V_\omega$ is used to estimate the value of final search states. (2) Estimate a value $\hat{Q}$ for all legal actions, and the search policy $\pi_{\text{search}}$ selects the legal action with the maximum $\hat{Q}$. (3) Use mixing technique to trade-off between $\pi_{\text{search}}$ and $\pi_\theta$ to choose the real action to be executed.

The procedure of generating *training data* is as follows: for each timestep $t$, we construct **in-context data** $D_t^k := (D^{\text{epi,k}}, D_t^{\text{step,k}})$ about $\pi^{-1,k}$, which is used to provide information about $\pi^{-1,k}$ for self-agent to recognize $\pi^{-1,k}$. $D^{\text{epi,k}} = \{(\tilde{s}_h, \tilde{a}_h^{-1,k})\}_{h=1}^H$ is *episode-wise in-context data*, generated by playing against $\pi^{-1,k}$ using any self-agent policy.[1] It is used to characterize the overall behavioral pattern of $\pi^{-1,k}$ on an episode-wise basis. See the construction process of $D^{\text{epi,k}}$ in App. C. $D_t^{\text{step,k}} = (s_0, a_0^{-1,k}, \dots, s_{t-1}, a_{t-1}^{-1,k})$ is *step-wise in-context data*, generated by the current episode involving $\pi^{-1,k}$ and $\pi^{1,k,*}$. It represents the step-wise specific behavior pattern of $\pi^{-1,k}$.

Furthermore, for each timestep $t$, we collect the *Return-To-Go* (RTG) obtained by the self-agent, denoted as $G_t^{1,k,*} = \sum_{t'=t}^T \gamma^{t'-t} r_{t'}^1 = \sum_{t'=t}^T \gamma^{t'-t} R^1(s_{t'}, a_{t'}^{1,k,*}, a_{t'}^{-1,k})$, where $a^{1,k,*} \sim \pi^{1,k,*}$, $a^{-1,k} \sim \pi^{-1,k}$, and $V_t^{1,k,*} = \mathbb{E}[G_t^{1,k,*}]$. To end with, the *training data for timestep* $t$ is obtained as:

$$\mathfrak{D}_t^k := (s_t, D_t^k, a_t^{1,k,*}, a_t^{-1,k}, G_t^{1,k,*}). \tag{2}$$

After preparing the training data, we use supervised learning to pretrain an actor $\pi_\theta(a_t^1|s_t, D_t^k)$ to learn the BR against $\pi^{-1,k}$, an opponent imitator $\mu_\phi(a_t^{-1,k} \mid s_t, D_t^k)$ to imitate the opponent's policy, and a critic $V_\omega(s_t, D_t^k)$ to estimate the state value of self-agent. Notably, all these components condition on $D_t^k$ as their in-context data. For each episode, the optimization objectives are as follows:

$$\max_\theta \mathbb{E}_{\mathfrak{D}_t^k, t \sim [T], k \sim [K]} \left[ \log \pi_\theta(a_t^{1,k,*} \mid s_t, D_t^k) \right], \tag{3}$$

$$\max_\phi \mathbb{E}_{\mathfrak{D}_t^k, t \sim [T], k \sim [K]} \left[ \log \mu_\phi(a_t^{-1,k} \mid s_t, D_t^k) \right], \tag{4}$$

$$\min_\omega \mathbb{E}_{\mathfrak{D}_t^k, t \sim [T], k \sim [K]} \left[ \left( V_\omega(s_t, D_t^k) - G_t^{1,k,*} \right)^2 \right]. \tag{5}$$

The left side of Fig. 1 illustrates OMIS's architecture and its pretraining procedure. Based on the understanding of ICL in decision-making, we design our architecture upon a causal Transformer [67].

### 4.2 Decision-Time Search with In-Context Components

Following the best practices in OM, we assume the testing environment features *unknown non-stationary opponent agents*, which we denote as $\Phi$. *Unknown* indicates that the self-agent is unable to ascertain the *true policy* $\bar{\pi}^{-1}$ employed by $\Phi$. *Non-stationary* implies that $\Phi$ switches its policy between episodes in some way, with each switch involving randomly sampling a $\bar{\pi}^{-1}$ from $\Pi^{\text{test}}$.

---

[1] $\sim$ is used to mark data in $D^{\text{epi,k}}$; $h$ is an index but not timestep.

Following the general setup in the DTS domain, we assume the ground truth transition dynamic $\mathcal{P}$ is available [75, 76, 12, 13, 9, 46, 38]. Based on the pretrained in-context components $\pi_\theta$, $V_\omega$, $\mu_\phi$, and $\mathcal{P}$, we conduct DTS to play against $\Phi$. At each timestep $t$, we perform $M$ times of rollouts with length $L$ for each legal action $\hat{a}_t^1$ of self-agent.[2] This is done to estimate the value $\hat{Q}(s_t, \hat{a}_t^1)$ for each $\hat{a}_t^1$ under current true opponent policy $\bar{\pi}^{-1}$ and current self-agent policy $\pi_\theta$. The self-agent then executes the legal action with the highest $\hat{Q}$ value in the real environment. Our expectation is that through such a DTS, we can refine the original policy $\pi_\theta$ to better adapt to $\Phi$.

The specific process of the *DTS* is as follows: for each timestep $t$, we first construct **in-context data** $D_t = (D^{\text{epi}}, D_t^{\text{step}})$ about $\Phi$, and its construction method is almost identical to $D_t^k$ mentioned in Sec. 4.1. However, since $\bar{\pi}^{-1}$ is unknowable, we make a slight modification: $D^{\text{epi}}$ is constructed by sampling consecutive segments from the most recent $C$ trajectories in which $\Phi$ participated.

After constructing $D_t$, for any given legal action $\hat{a}_t^1$, we sample the opponents' action by $\hat{a}_t^{-1} \sim \mu_\phi(\cdot|s_t, D_t)$ and transition using $\mathcal{P}$ to obtain $\hat{s}_{t+1}$ and $\hat{r}_t^1$. We append $(s_t, \hat{a}_t^{-1})$ to the end of $D_t^{\text{step}}$ to obtain the updated step-wise in-context data $\hat{D}_{t+1}^{\text{step}}$ and in-context data $\hat{D}_{t+1} = (D^{\text{epi}}, \hat{D}_{t+1}^{\text{step}})$. Following, at the $l$-th step of the rollout for $\hat{a}_t^1$ ($l \in [L]$), we sample self-agent action and opponent action using the following two formulas, respectively:

$$\hat{a}_{t+l}^1 \sim \pi_\theta(\cdot|\hat{s}_{t+l}, \hat{D}_{t+l}), \tag{6}$$

$$\hat{a}_{t+l}^{-1} \sim \mu_\phi(\cdot|\hat{s}_{t+l}, \hat{D}_{t+l}). \tag{7}$$

Transitioning with $\mathcal{P}$ yields $\hat{s}_{t+l+1}$ and $\hat{r}_{t+l}^1$. Next, we append $(\hat{s}_{t+l}, \hat{a}_{t+l}^{-1})$ to the end of $\hat{D}_{t+l}^{\text{step}}$ to obtain $\hat{D}_{t+l+1}^{\text{step}}$ and $\hat{D}_{t+l+1} = (D^{\text{epi}}, \hat{D}_{t+l+1}^{\text{step}})$. After completing the $L$-th step, we use $\hat{V}_{t+L+1} := V_\omega(\hat{s}_{t+L+1}, \hat{D}_{t+L+1})$ to estimate the value of the final search state. When finishing $M$ times of rollouts for $\hat{a}_t^1$, we obtain an estimated value for $\hat{a}_t^1$ by:

$$\hat{Q}(s_t, \hat{a}_t^1) := \frac{1}{M} \sum_{m=1}^{M} \left[ \sum_{t'=t}^{t+L} \gamma_{\text{search}}^{t'-t} \cdot \hat{r}_{t'}^1 + \gamma_{\text{search}}^{L+1} \cdot \hat{V}_{t+L+1} \right]. \tag{8}$$

Here, $\gamma_{\text{search}}$ is the discount factor used in the DTS. After completing rollouts for all legal actions of the self-agent at timestep $t$, we obtain our *search policy* by maximizing $\hat{Q}$:

$$\pi_{\text{search}}(s_t) := \arg\max_{\hat{a}_t^1} \hat{Q}(s_t, \hat{a}_t^1). \tag{9}$$

In practical implementation, we observe that using $\pi_{\text{search}}$ directly is not always effective. This is because we cannot totally precisely estimate the opponents' policy and the state value, making the results obtained from the DTS not sufficiently reliable across all states. To this phenomenon, we propose a simple yet effective **mixing technique** to balance the search policy and the original actor policy in deciding the action to be executed:

$$\pi_{\text{mix}}(s_t) := \begin{cases} \pi_{\text{search}}(s_t), & ||\hat{Q}(s_t, \pi_{\text{search}}(s_t))|| > \epsilon \\ a_t^1 \sim \pi_\theta(\cdot|s_t, D_t), & \text{otherwise} \end{cases}. \tag{10}$$

Here, $\epsilon$ is a threshold hyperparameter. The main motivation for the mixing technique is as follows. We consider the expected return of the action selected by the DTS as the confidence of the search policy. When the confidence exceeds a certain threshold, we tend to consider that $\pi_{\text{search}}$ has a relatively high probability of achieving better results than $\pi_\theta$; otherwise, we prefer to use the original policy $\pi_\theta$. See the testing procedure of OMIS on the right side of Fig. 1.

### 4.3 Theoretical Analysis

Our theoretical analysis unfolds in the following two aspects. (1) We propose Lem. 4.1 and Thm. 4.2 to prove that *OMIS without DTS* (denoted as *OMIS w/o S*) converges to the optimal solution when facing a *true opponent policy* $\bar{\pi}^{-1} \in \Pi^{\text{train}}$; and it recognizes the opponent policy as the policy in $\Pi^{\text{train}}$ with a minimum certain form of KL divergence from $\bar{\pi}^{-1}$ when facing a $\bar{\pi}^{-1} \notin \Pi^{\text{train}}$. (2) Building upon Thm. 4.2, we further propose Thm. 4.3 to prove the policy improvement theorem of OMIS with DTS, ensuring that it leads to enhancements in performance.

---

[2] $\wedge$ is used to mark the terms during DTS.

To begin with, we instantiate a *Posterior Sampling in Opponent Modeling* (PSOM) algorithm (see App. D.1) based on the *Posterior Sampling* (PS) algorithm [61], where PSOM can be proven to share the same guarantees of converging to the optimal solution as PS. Based on some reasonable assumptions, we prove that OMIS w/o S is equivalent to PSOM.

**Lemma 4.1** (Equivalence of OMIS w/o S and PSOM). *Assume that the learned $\pi_\theta$ is consistent and the sampling of $s$ from $\mathcal{T}_{pre}^{-1}$ is independent of opponent policy, then given $\bar{\pi}^{-1}$ and its $D$, we have $P(\xi_T^1|D, \bar{\pi}^{-1}; PSOM) = P(\xi_T^1|D, \bar{\pi}^{-1}; \pi_\theta)$ for all possible $\xi_T^1$.*

Here, '*consistent*' indicates that the network's fitting capability is guaranteed. $\mathcal{T}_{pre}^{-1}(\cdot; \pi^{-1})$ denotes the *probability distribution on all the trajectories involving $\pi^{-1}$ during pretraining*. $\xi_T^1 = (s_1, a_1^{1,*}, \ldots, s_T, a_T^{1,*})$ is *self-agent history*, where $a^{1,*}$ is sampled from the BR to the opponent policy recognized by PSOM. The proof is provided in App. D.2.

**Theorem 4.2.** *When $\bar{\pi}^{-1} = \pi^{-1,k} \in \Pi^{train}$, if the PS algorithm converges to the optimal solution, then OMIS w/o S recognizes the policy of $\Phi$ as $\pi^{-1,k}$, i.e., $\pi_\theta$, $\mu_\phi$, and $V_\omega$ converge to $\pi^{1,k,*}$, $\pi^{-1,k}$, and $V^{1,k,*}$, respectively; When $\bar{\pi}^{-1} \notin \Pi^{train}$, OMIS w/o S recognizes the policy of $\Phi$ as the policies in $\Pi^{train}$ with the minimum $D_{KL}(P(a^{-1}|s, \pi^{-1})||P(a^{-1}|s, \bar{\pi}^{-1}))$.*

Based on this theorem, when OMIS w/o S faces seen opponents, it accurately recognizes the opponent's policy and converge to the BR against it; when facing unseen opponents, it recognizes the opponent's policy as the seen opponent policy with the smallest KL divergence from this unseen opponent policy and produces the BR to the recognized opponent policy. The proof is in App. D.3.

**Theorem 4.3** (Policy Improvement of OMIS's DTS). *Given $\bar{\pi}^{-1}$ and its $D$, suppose OMIS recognizes $\Phi$ as $\pi_\star^{-1}$ and $V_{\pi_\star}^{\pi_{-1}}$ is the value vector on $\mathcal{S}$, where $V(s) := V_\omega(s, D), \pi(a|s) := \pi_\theta(a|s, D)$. Let $\mathcal{G}_L$ be the L-step DTS operator and $\pi' \in \mathcal{G}_L(V_{\pi_\star}^\pi)$, then $V_{\pi_\star}^{\pi'} \geq V_{\pi_\star}^\pi$ holds component-wise.*

Within, OMIS's DTS can be viewed as $\mathcal{G}_L$, and $\pi'$ corresponds to $\pi_{search}$ in Eq. (9). Thm. 4.3 indicates that OMIS's DTS is guaranteed to bring improvement, laying the foundation for performance stability. Additionally, OMIS's DTS avoids gradient updates. The proof is provided in App. D.4.

**Analysis for generalization in OM.** In OM, *generalization* can be typically defined as *performance when facing unknown opponent policies (*e.g.*, opponents like $\Phi$)*. Existing approaches lack rigorous theoretical analysis under this definition of generalization. In Lem. 4.1, we proved that OMIS w/o S is equivalent to PSOM. In Thm. 4.2, we proved that PSOM can accurately recognize seen opponents and recognize unseen opponents as the most similar to the seen ones. Since OMIS w/o S is equivalent to PSOM, OMIS w/o S possesses the same properties. Additionally, Thm. 4.3 proved that OMIS's DTS ensures performance improvement while avoiding instability. These theoretical analyses potentially provide OMIS with benefits in terms of generalization in OM.

## 5 Experiments

### 5.1 Experimental Setup

**Environments.** We consider three sparse-reward benchmarking environments for OM as shown in Fig. 2 (See App. E for detailed introductions of them):

- `Predator Prey` (PP) is a competitive environment with a continuous state space. In PP, the self-agent is a prey (green) whose goal is to avoid being captured by three predators (red) as much as possible. There are two obstacles (black) on the map. The challenge of PP lies in the need to model all three opponents simultaneously and handle potential cooperation among them.

- `Level-Based Foraging` (LBF) is a mixed environment in a grid world. In LBF, the self-agent is the blue one, aiming to eat as many apples as possible. The challenge of LBF is that cooperation with the opponent is necessary to eat apples of a higher level than the self-agent's (the apples and agents' levels are marked in the bottom-right). LBF represents a typical social dilemma.

- `OverCooked` (OC) is a cooperative environment using high-dimensional images as states. In OC, the self-agent is the green one, which aims at collaborating with the opponent (blue) to serve dishes as much as possible. The challenge of OC lies in the high-intensity coordination required between the two agents to complete a series of sub-tasks to serve a dish successfully.

**Baselines.** We consider the following representative PFAs, TFAs, and DTS-based OM approach:

- DRON [29]: Encode hand-crafted features of opponents using a Mixture-of-Expert network while also predicting opponents' actions as auxiliary task (this is the most performant version in [29]).
- LIAM [63]: Use the observations and actions of the self-agent to reconstruct those of the opponent through an auto-encoder, thereby embedding the opponent policy into a latent space.
- MeLIBA [107]: Use *Variational Auto-Encoder* (VAE) to reconstruct the opponent's future actions and condition on the embedding generated by this VAE to learn a Bayesian meta-policy.
- Meta-PG [3]: Execute multiple meta-gradient steps to anticipate changes in opponent policies to enable fast adaptation during testing.
- Meta-MAPG [41]: Compared to Meta-PG, it includes a new term that accounts for the impact of the self-agent's current policy on the opponent's future policy.
- MBOM [101]: Use recursive reasoning in an environment model to learn opponents at different recursion levels and combine them by Bayesian mixing.
- OMIS w/o S: A variant of OMIS, where OMIS directly uses $\pi_\theta$ based on $D_t$ without DTS.
- SP-MCTS [89]: Use a scripted opponent model to estimate the opponent's actions and apply MCTS for DTS. We adopt OMIS w/o S as its original self-agent policy. This is a DTS-based OM approach.

Within, DRON, LIAM, and MeLIBA belong to PFAs; Meta-PG, Meta-MAPG, and MBOM belong to TFAs. See neural architecture design of all approaches in App. F. For a fair comparison, we implement MBOM and SP-MCTS using the ground truth transition $\mathcal{P}$ as the environment model.

**Opponent policy.** We employ a diversity-driven Population Based Training algorithm MEP [104] to train a policy population. Policies from this MEP population are used to construct $\Pi^{\text{train}}$ and $\Pi^{\text{test}}$, ensuring that all opponent policies are performant and exhibit diversity. We measure and visualize the diversity of opponent policies within the MEP population in App. G.

We randomly select 10 policies from the MEP population to form $\Pi^{\text{train}}$. Then, we categorize opponent policies in the MEP population into two types: '*seen*' denotes policies belonging to $\Pi^{\text{train}}$, and '*unseen*' denotes policies not belonging to $\Pi^{\text{train}}$. We set up opponent policies with different ratios of $[seen : unseen]$ to form $\Pi^{\text{test}}$, *e.g.*, $[seen : unseen] = [0 : 10]$ signifies that $\Pi^{\text{test}}$ is composed of 10 opponent policies that were never seen during pretraining.

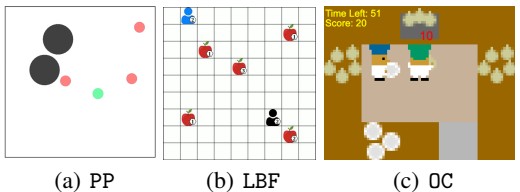

(a) PP    (b) LBF    (c) OC

Figure 2: The benchmarking environments.

During testing, all opponents are the *unknown non-stationary opponent agents* $\Phi$ mentioned in Sec. 4.2. $\Phi$ switches policies by sampling from $\Pi^{\text{test}}$ every $E$ episodes.

**Specific settings.** For the pretraining stage, we train all approaches for 4000 steps. For the testing stage, all approaches use the final checkpoints of pretraining to play against $\Phi$ for 1200 episodes. All bar charts, line charts, and tables report the *average* and *standard deviation* of the mean results over 5 random seeds. See all the hyperparameters in App. H.

### 5.2 Empirical Analysis

We pose a series of questions and design experiments to answer them, aiming to analyze OMIS's opponent adaptation capability and validate each component's effectiveness.

**Question 1.** *Can OMIS effectively and stably adapt to opponents under various $\Pi^{\text{test}}$ configurations?*

We set up 5 different $[seen : unseen]$ ratios to form $\Pi^{\text{test}}$, and we show the average results of all approaches against $\Phi$ corresponding to each ratio in Fig. 3. OMIS exhibits a higher average return and lower variance than other baselines across three environments, highlighting its effectiveness and stability in opponent adaptation under different $\Pi^{\text{test}}$ configurations. It can be observed that OMIS w/o S outperforms existing PFAs (*e.g.*, MeLIBA) in most cases, validating that pretraining based on ICL exhibits good generalization on testing with unknown opponent policies.

The results also indicate that OMIS improves OMIS w/o S more effectively than SP-MCTS does. SP-MCTS sometimes even makes OMIS w/o S worse (*e.g.*, in OC and parts of PP). This might be because (1) the opponent model of SP-MCTS, which estimates opponent actions, is non-adaptive and (2) the trade-off between exploration and exploitation in the MCTS is non-trivial to optimize.

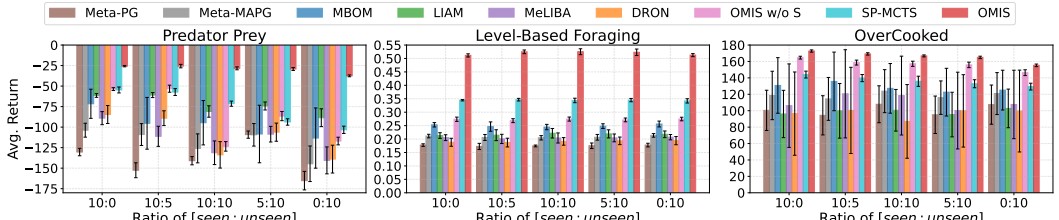

Figure 3: Average results of testing under different $[seen : unseen]$ ratios, where $E = 20$.

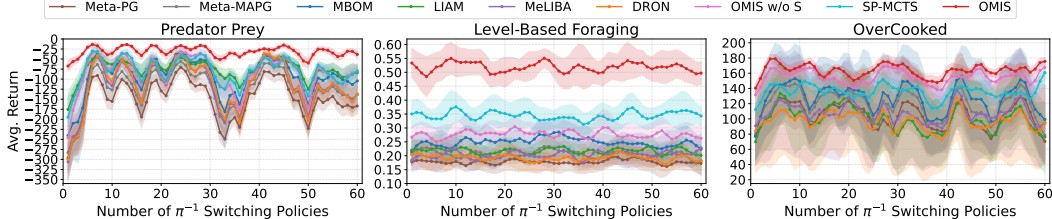

Figure 4: Average results against each true opponent policy during testing, where $[seen : unseen] = [10 : 10]$ and $E = 20$. Each point in X-axis denotes a policy switching of $\Phi$, totaling 60 times. Y-axis denotes the average return against the corresponding switched $\bar{\pi}^{-1}$.

**Question 2.** *Can OMIS adapt well to each one of the true policies adopted by $\Phi$?*

In Fig. 4, we provide the average results of all approaches against each true opponent policy $\bar{\pi}^{-1}$ employed by $\Phi$ corresponding to ratio of $[seen : unseen] = [10 : 10]$. Similar to the observations in Fig. 3, OMIS exhibits higher return and lower variance than other baselines across various true opponent policies in PP, LBF, and OC. TFAs (*e.g.*, Meta-PG) generally show significant performance gaps when facing different true opponent policies. This is likely due to the continuous switching of opponent policies, making finetuning during testing challenging or ineffective.

**Question 3.** *How does OMIS work when the transition dynamics are learned?*

We include a variant named *Model-Based OMIS* (MBOMIS) to verify whether OMIS can work effectively when the transition dynamics are unknown and learned instead. MBOMIS uses the most straightforward method to learn a transition dynamic model $\hat{\mathcal{P}}$: given a state $s$ and action $a$, predicting the next state $s'$ and reward $r$ using *Mean Square Error* (MSE) loss. $\hat{\mathcal{P}}$ is trained using the $(s, a, r, s')$ tuples from the dataset used for pretraining OMIS w/o S. The testing results against unknown non-stationary opponents are shown in Fig. 5. Although MBOMIS loses some performance compared to OMIS, it still effectively improves over OMIS w/o S and generally surpasses other baselines. We also provide quantitative evaluation results of $\hat{\mathcal{P}}$'s estimation during testing in Tab. 1. Observations show that $\hat{\mathcal{P}}$ generally has a small MSE value in predicting the next state and reward without normalization.

**Question 4.** *Is OMIS robust to the frequency of which $\Phi$ switches policies?*

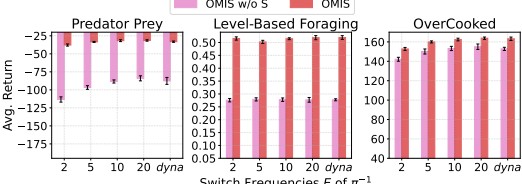

Figure 7: Average results during testing when $\Phi$ adopts different switching frequencies, where $[seen : unseen] = [10 : 10]$.

We employ 5 different frequencies for $\Phi$ to switch policies, *i.e.*, $E = 2, 5, 10, 20, dynamic$ (abbreviated as $dyna$). Here, $E = dyna$ indicates that $\Phi$ randomly selects from $2, 5, 10, 20$ at the start of each switch as the number of episodes until the next switch. Fig. 7 shows the average results against $\Phi$ with different switching frequencies. OMIS and OMIS w/o S exhibit a degree of robustness to different policy switching frequencies $E$ of $\Phi$ in PP, LBF, OC. Notably, as $E$ increases, their performance generally shows a slight upward trend. This suggests that OMIS could gradually stabilize its adaptation to true opponent policies by accumulating in-context data.

**Question 5.** *Is each part of OMIS's method design effective?*

We design various ablated variants of OMIS: (1) OMIS w/o S (See Sec. 5.1); (2) OMIS w/o mixing: a variant where the mixing technique is not used, *i.e.*, using $\pi_{\text{search}}$ instead of $\pi_{\text{mix}}$; (3) OMIS w/o

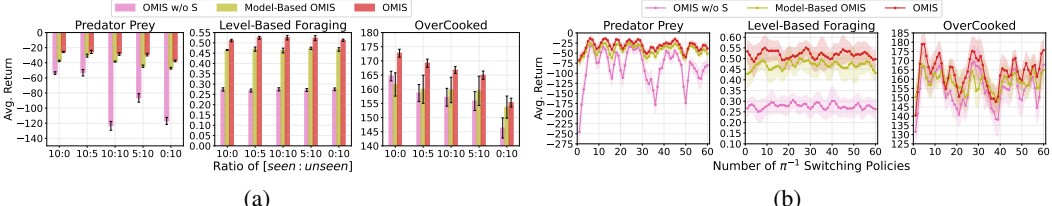

(a)                                     (b)

Figure 5: Results of OMIS using learned dynamics against unknown non-stationary opponents. (a) Average results of testing under different $[seen : unseen]$ ratios, where $E = 20$. (b) Average results against each true opponent policy during testing, where $[seen : unseen] = [10 : 10]$ and $E = 20$.

Table 1: Quantitative evaluation results of the learned dynamic $\hat{\mathcal{P}}$'s estimation during testing. The results are calculated during testing under different $[seen : unseen]$ ratios, where $E = 20$.

| Environment | Evaluation Metric | $[seen : unseen]$ | | | | |
| --- | --- | --- | --- | --- | --- | --- |
| | | $10 : 0$ | $10 : 5$ | $10 : 10$ | $5 : 10$ | $0 : 10$ |
| Predator Prey | Avg. Next State MSE $\downarrow$ | $0.00409 \pm 0.00002$ | $0.00433 \pm 0.00011$ | $0.00410 \pm 0.00005$ | $0.00462 \pm 0.00004$ | $0.00445 \pm 0.00002$ |
| | Avg. Reward MSE $\downarrow$ | $0.13836 \pm 0.01425$ | $0.24080 \pm 0.05558$ | $0.15288 \pm 0.01641$ | $0.26276 \pm 0.03761$ | $0.22110 \pm 0.00567$ |
| Level-Based Foraging | Avg. Next State MSE $\downarrow$ | $0.05759 \pm 0.00187$ | $0.05043 \pm 0.00074$ | $0.05702 \pm 0.00160$ | $0.05525 \pm 0.00154$ | $0.04922 \pm 0.00066$ |
| | Avg. Reward MSE $\downarrow$ | $0.00004 \pm 0.00000$ | $0.00004 \pm 0.00000$ | $0.00004 \pm 0.00000$ | $0.00004 \pm 0.00000$ | $0.00003 \pm 0.00000$ |
| OverCooked | Avg. Next State MSE $\downarrow$ | $0.00028 \pm 0.00001$ | $0.00025 \pm 0.00001$ | $0.00028 \pm 0.00001$ | $0.00028 \pm 0.00001$ | $0.00031 \pm 0.00001$ |
| | Avg. Reward MSE $\downarrow$ | $0.00000 \pm 0.00000$ | $0.00000 \pm 0.00000$ | $0.00000 \pm 0.00000$ | $0.00000 \pm 0.00000$ | $0.00000 \pm 0.00000$ |

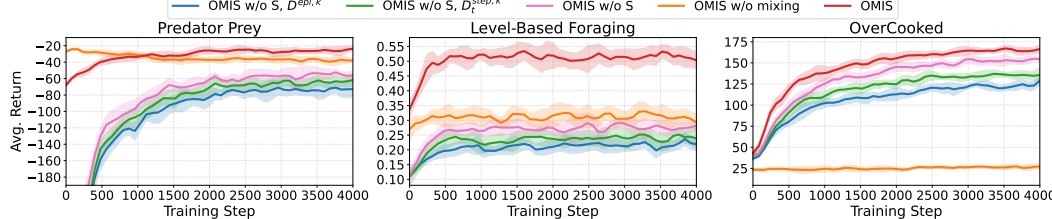

Figure 6: Average performance curves during pretraining against all policies in $\Pi^{\text{train}}$.

S, $D_t^{\text{step},k}$: a variant without DTS where $D_t^{\text{step},k}$ input is excluded from the model; (4) OMIS w/o S, $D^{\text{epi},k}$: a variant without DTS where $D^{\text{epi},k}$ input is excluded from the model.

Fig. 6 shows the average performance curves during pretraining for these variants against all policies in $\Pi^{\text{train}}$. In PP, LBF, and OC, OMIS w/o S consistently performs a lot worse than OMIS. This indicates that the DTS effectively improves the original policy of $\pi_\theta$. OMIS w/o mixing exhibits a notable performance decrease compared to OMIS in LBF and OC. This suggests selective searching based on confidence is more effective than indiscriminately. It can be observed that $D^{\text{epi},k}$ and $D_t^{\text{step},k}$ both play crucial roles in OMIS's adaptation to opponents, with $D^{\text{epi},k}$ making a greater contribution. This could be because capturing the overall behavioral patterns of opponents is more important than focusing on their step-wise changes.

**Question 6.** *Can OMIS effectively characterize opponent policies through in-context data?*

For each policy in $\Pi^{\text{train}}$, we visualize OMIS's attention weights of $D^{\text{epi},k}$ over the final 20 timesteps in an episode in Fig. 8. Each position of tokens in $D^{\text{epi},k}$ has a weight indicated by the depth of color. In all three environments, the attention of OMIS exhibits the following characteristics: (1) Focusing on specific positions of tokens in $D^{\text{epi},k}$ for different opponent policies; (2) Maintaining a relatively consistent distribution for a given opponent policy across various timesteps within the same episode. This implies that OMIS can represent different opponent policies according to distinct patterns of different in-context data. We also provide quantitative analysis on OMIS's attention weights in App. I.

**Question 7.** *How well does the in-context components of OMIS estimate?*

We collect true opponent actions and true RTGs as labels, using *Accuracy* and *MSE* as metrics to evaluate the effectiveness of $\mu_\phi$'s and $V_\omega$'s estimations, respectively. In Tab. 2, we provide estimation results of the in-context components $\mu_\phi, V_\omega$ during testing under different $[seen : unseen]$ ratios. It can be observed that $\mu_\phi$ is estimated relatively accurately in OC, while $V_\omega$ is estimated relatively accurately in PP and LBF. However, $\mu_\phi$ does not estimate very accurately in PP and LBF. This indicates that the functioning of OMIS does not necessarily depend on very precise estimates. In all three



Figure 8: Attention heatmaps of OMIS when playing against different policies in $\Pi^{\text{train}}$.

Table 2: The estimation results of the in-context components of OMIS during testing, where $E = 20$.

| Env. | Predator Prey | | | Level-Based Foraging | | | OverCooked | | |
|---|---|---|---|---|---|---|---|---|---|
| $[seen : unseen]$ | 10 : 0 | 0 : 10 | 10 : 10 | 10 : 0 | 0 : 10 | 10 : 10 | 10 : 0 | 0 : 10 | 10 : 10 |
| Avg. Acc. (%) ↑ | 60.4 ± 0.9 | 46.4 ± 0.6 | 53.0 ± 0.9 | 63.1 ± 0.8 | 56.3 ± 0.9 | 59.2 ± 0.8 | 79.8 ± 0.3 | 64.4 ± 0.6 | 71.9 ± 0.4 |
| Avg. MSE ↓ | 0.01 ± 0.00 | 0.07 ± 0.02 | 0.03 ± 0.01 | 0.01 ± 0.00 | 0.01 ± 0.00 | 0.01 ± 0.00 | 0.12 ± 0.20 | 0.20 ± 0.27 | 0.16 ± 0.22 |

Figure 9: Visualization of OMIS's DTS when playing against an $unseen$ opponent policy.

environments, the estimation accuracy for purely unseen opponents does not decrease significantly, further confirming the good generalization of ICL.

**Question 8.** *How does OMIS's DTS work in real games?*

Fig. 9 visualizes the OMIS's DTS process at a particular timestep during a game against an unseen opponent policy in three environments. We only illustrate two legal actions as an example. It can be observed that OMIS's DTS promptly evaluates each legal action, predicts the opponent's actions during the DTS, and ultimately selects the most advantageous action. We notice the following interesting phenomena: In PP, DTS enables the self-agent to avoid being captured by opponents who use an encirclement policy; In LBF, DTS allows the self-agent to cooperate with opponents to eat apples with a higher level than itself; In OC, DTS helps prevent the self-agent from blocking the path of collaborators, allowing them to serve dishes smoothly.

## 6 Discussion

**Summary.** In this paper, we propose OMIS based on ICL and DTS, aiming to address the challenges of limited generalization abilities and performance instability issues faced by existing OM approaches. The foundations of OMIS lie in two stages: (1) We employ ICL to pretrain a Transformer model consisting of three components: actor, opponent imitator, and critic. We prove that this model converges in opponent policy recognition and has good properties in terms of generalization; (2) Based on the pretrained in-context components, we use a DTS to refine the policy of the original actor. This DTS avoids (the instability-causing) gradient updates and provides improvement guarantees. Extensive experimental results in three environments validate that OMIS adapts effectively and stably to unknown non-stationary opponent agents.

**Limitations and future work.** (1) We only considered opponents with non-stationary switches among several fixed unknown policies during testing. OMIS might face challenges in adapting to opponents who are continuously learning or reasoning; (2) This work focuses on the setting of perfect information games. Effectively incorporating ICL and DTS for OM in imperfect information games is a challenging and meaningful research problem; (3) OMIS only utilizes a naive DTS method to refine its policy. We will continue to explore how to apply more advanced DTS methods to the OM domain for more effective adaptation to unknown opponents. A more **in-depth discussion** is in App. J.

## Acknowledgments

This work is supported in part by the National Science and Technology Major Project (2022ZD0116401); the Natural Science Foundation of China under Grant 62076238, Grant 62222606, and Grant 61902402; the Jiangsu Key Research and Development Plan (No. BE2023016); and the China Computer Federation (CCF)-Tencent Open Fund.

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

# Table of Contents for the Appendix

# A   Extensive Related Work

**Opponent modeling.** The recent research in opponent modeling based on machine learning methodologies can be broadly categorized as follows:

(1) Opponent Modeling with Representation Learning: Embed the opponent policies into a latent space using representation learning methods to enhance the decision-making capabilities of the self-agent. In the study by Grover et al. [28], imitation learning [37] and contrastive learning [39] were utilized to produce policy embeddings for opponent trajectories. Subsequently, these embeddings were integrated with reinforcement learning for policy optimization. Comparable initiatives, exemplified by He et al. [29] and Hong et al. [33], employed auxiliary networks to encode manually crafted opponent features, predict opponent actions, and finetune the policy network to enhance overall performance. Papoudakis et al. [63] suggested employing an autoencoder to leverage the self-agent's observations and actions for reconstructing the opponent's observations and actions. This approach aims to learn embeddings that facilitate decision-making. In comparison, Papoudakis and Albrecht [62] and Zintgraf et al. [107] utilized Variational AutoEncoders (VAE) [42] to capture the high-dimensional distribution of opponent policies.

(2) Opponent Modeling with Bayesian Learning: Detect or deduce the opponent policies in real-time using Bayesian methods and subsequently generate responses based on the inferred information. Bard et al. [10] initially trained a mixture-of-expert counter-strategies against a predefined set of fixed opponent policies. They then utilized a bandit algorithm during testing to dynamically select the most suitable counter-strategy. Building on BPR+ [71, 32], Zheng et al. [106] introduced a rectified belief model to enhance the precision of opponent policy detection. Furthermore, they introduced a distillation policy network to achieve better results. DiGiovanni and Tewari [19] utilized Thompson sampling [82] and change detection methods to address the challenge of opponent switching between stationary policies. Fu et al. [24] employed a bandit algorithm to select either greedy or conservative policies when playing against a non-stationary opponent. The greedy policy underwent training via VAE in conjunction with conditional reinforcement learning and was continuously updated online through variational inference. In contrast, the conservative policy remained a fixed and robust policy. Lv et al. [54] introduced a similar approach to exploit non-stationary opponents.

(3) Opponent Modeling with Meta-learning: Leveraging meta-learning methods [34], train against a set of given opponent policies to adapt to unknown opponent policies during testing swiftly. While most meta-reinforcement learning methods presume that tasks in training and testing exhibit a similar distribution, this category investigates the possible application of meta-reinforcement learning in the context of competing with unknown non-stationary opponents. Al-Shedivat et al. [3] introduced a gradient-based meta-learning algorithm designed for continuous adaptation in non-stationary and competitive environments, showcasing its efficacy in enhancing adaptation efficiency. Building upon analysis on Al-Shedivat et al. [3], Kim et al. [41] introduced a novel meta multi-agent policy gradient algorithm designed to effectively handle the non-stationary policy dynamics inherent in multi-agent reinforcement learning. Zintgraf et al. [107] introduced a meta-learning approach for deep interactive Bayesian reinforcement learning in multi-agent settings. This approach utilizes approximate belief inference and policy adaptation to enhance opponent adaptation. Wu et al. [94] put forward a meta-learning framework called Learning to Exploit (L2E) for implicit opponent modeling. This framework enables agents to swiftly adapt and exploit opponents with diverse styles through limited interactions during training.

(4) Opponent Modeling with Shaping Opponents' Learning: Considering opponents' learning (*i.e.*, conducting gradient updates), estimating the mutual influence between the future opponent policy and the current self-agent's policy. Foerster et al. [22] proposed an approach named LOLA, which modeled the one-step update of the opponent's policy and estimated the learning gradient of the opponent's policy. Foerster et al. [23] introduced a Differentiable Monte-Carlo Estimator operation to explore the shaping of learning dynamics for other agents, building upon the approach presented by Foerster et al. [22]. Letcher et al. [47] further integrated stability guarantees from LookAhead with opponent-shaping capabilities from Foerster et al. [22] to achieve theoretical and experimental improvements. Kim et al. [41] also presented a term closely related to shaping the learning dynamics of other agents' policies. This considers the impacts of the agent's current policy on future opponent policies. Lu et al. [53] proposed a meta-learning approach for general-sum games that can exploit naive learners without requiring white-box access or higher-order derivatives. Willi et al. [92] introduced Consistent LOLA (COLA), a new multi-agent reinforcement learning algorithm that addresses inconsistency

issues with Foerster et al. [22]'s approach. COLA learns consistent update functions for agents by explicitly minimizing a differentiable measure of consistency. Zhao et al. [105] proposed proximal LOLA (POLA), an algorithm that addresses policy parameterization sensitivity issues with LOLA and more reliably learns reciprocity-based cooperation in partially competitive multi-agent environments. Fung et al. [25] further presented that the sample complexity of Lu et al. [53]'s algorithm scales exponentially with the inner state and action space and the number of agents.

(5) Opponent Modeling with Recursive Reasoning: By simulating nested layers of beliefs, predicting the opponent's behavior, and generating the best response based on the expected reasoning process of the opponent towards the self-agent. Wen et al. [90] and Wen et al. [91] suggested conducting recursive reasoning by modeling hypothetical nested beliefs through the agent's joint Q-function. Their work demonstrated enhanced adaptation in stochastic games. Dai et al. [17] introduced recursive reasoning by assuming agents select actions based on GP-UCB acquisition functions [78]. This approach achieved faster regret convergence in repeated games. Yuan et al. [102] proposed an algorithm that utilizes ToM-based recursive reasoning and adaptive reinforcement learning. This approach enables agents to develop a pragmatic communication protocol to infer hidden meanings from context and enhance cooperative multi-agent communication. Yu et al. [101] proposed a model-based opponent modeling approach that employs recursive imagination within an environment model and Bayesian mixing to adapt to diverse opponents.

(6) Opponent Modeling with Theory of Mind (ToM): Reasoning about the opponent's mental states and intentions to predict and adapt to their behavior. This involves modeling their beliefs, goals, and actions to understand opponent dynamics comprehensively. Von Der Osten et al. [86] introduced a multi-agent ToM model designed to predict opponents' actions and infer strategy sophistication in stochastic games. Building upon Zheng et al. [106]'s Bayesian online opponent modeling approach, Yang et al. [99] proposed a ToM approach. This approach leverages higher-level decision-making to play against opponents who are also engaged in opponent modeling. Rabinowitz et al. [66] and Raileanu et al. [68] also delved into methodologies for modeling an opponent's mental state. Rabinowitz et al. [66] utilized three networks to reason about agent actions and goals, simulating a human-like ToM. Raileanu et al. [68] proposed utilizing their own policy to learn the opponent's goals in conjunction with the opponent's observations and actions.

Our work aims to pioneer a new methodology: Opponent Modeling with Decision-Time Search (DTS). We explore how DTS can work in the context of opponent modeling, as intuitively, DTS can make our policy more foresighted. To imbue DTS with opponent awareness and adaptability, we developed a model based on in-context learning to serve as the foundation for DTS.

**Transformers for decision-making.** There is an increasing interest in leveraging Transformers for decision-making by framing the problem as sequence modeling [97, 48]. Chen et al. [15] introduced Decision Transformer (DT), a model that predicts action sequences conditioned on returns using a causal Transformer trained on offline data. Subsequent studies have explored enhancements, such as improved conditioning [26, 65], and architectural innovations [85]. Another interesting direction capitalizes on the generality and scalability of Transformers for multi-task learning [45, 70]. Transformers applied to decision-making have demonstrated meta-learning capabilities as well [55]. Recently, Lee et al. [44] introduced a Transformer-based in-context learning approach that, both empirically and theoretically, surpasses behaviors observed in the dataset in terms of regret, a performance metric where DT falls short [11, 98]. Incorporating these insights, our work employs a causal Transformer architecture to maximize the model's ability for in-context learning, specifically in the realm of opponent modeling.

# B Pseudocode of OMIS

---

**Algorithm 1** Opponent Modeling with In-context Search (OMIS)

---

1: /* Prepare all the opponent policies */
2: Train an opponent policy population using the MEP algorithm to construct $\Pi^{\text{train}}$ and $\Pi^{\text{test}}$.
3: /* In-context-learning-based pretraining (Section 4.1) */
4: Train the BR against each opponent policy in $\Pi^{\text{train}}$ using the PPO algorithm and get $\{BR(\pi^{-1,k})\}_{k=1}^{K}$.
5: Initialize $\pi_\theta$ with parameters $\theta$, $\mu_\phi$ with parameters $\phi$, and $V_\omega$ with parameters $\omega$.
6: **while** not converged **do**
7:     Sample a opponent policy $\pi^{-1,k} \sim \Pi^{\text{train}}$, play an episode against it using $BR(\pi^{-1,k}) := \pi^{1,k,*}(a|s)$,
    and collect the training data $\{\mathfrak{D}_t^k\}_{t=1}^{T}$, where $\mathfrak{D}_t^k := (s_t, D_t^k, a_t^{1,k,*}, a_t^{-1,k}, G_t^{1,k,*})$.
8:     Calculate losses based on **Eqs. (3) to (5)**, and backpropagate to update $\theta, \phi, \omega$.
9: **end while**
10: /* Decision-time search with in-context components (Section 4.2) */
11: Set the policy switching frequency $E$ for the unknown non-stationary opponent agents $\Phi$.
12: **for** num_switch in max_switching_number **do**
13:     Sample a true opponent policy $\bar{\pi}^{-1} \sim \Pi^{\text{test}}$ using uniform distribution, and set $\Phi = \bar{\pi}^{-1}$.
14:     **for** ep in $[E]$ **do**
15:       **for** $t$ in $[T]$ **do**
16:         // Decision-time search
17:         **for** $\hat{a}_t^1$ in all_legal_actions **do**
18:           Rollout $L$ steps for $M$ times using $\pi_\theta, \mu_\phi, V_\omega$, and estimate the value for $\hat{a}_t^1$ by

$$\hat{Q}(s_t, \hat{a}_t^1) := \frac{1}{M} \sum_{m=1}^{M} \left[ \sum_{t'=t}^{t+L} \gamma_{\text{search}}^{t'-t} \cdot \hat{r}_{t'}^1 + \gamma_{\text{search}}^{L+1} \cdot \hat{V}_{t+L+1} \right].$$

19:         **end for**
20:         // Take real actions
21:         The opponents act according to $\Phi$, while simultaneously the self-agent acts according to

$$\pi_{\text{mix}}(s_t) := \begin{cases} \pi_{\text{search}}(s_t), & ||\hat{Q}(s_t, \pi_{\text{search}}(s_t))|| > \epsilon \\ a_t^1 \sim \pi_\theta(\cdot|s_t, D_t), & \text{otherwise} \end{cases}, \text{ where } \pi_{\text{search}}(s_t) := \arg\max_{\hat{a}_t^1} \hat{Q}(s_t, \hat{a}_t^1).$$

22:       **end for**
23:     **end for**
24: **end for**

---

# C Construction Process of $D^{\text{epi,k}}$

The construction of $D^{\text{epi,k}}$ is as follows:

1. Sample $C$ trajectories from all historical games involving $\pi^{-1,k}$;

2. For each trajectory, sample consecutive segments $\{(s_{h'}, a_{h'-1}^{-1,k})\}_{h'=h_s}^{h_s + \frac{H}{C} - 1}$, where $h_s$ is the starting timestep;

3. Concatenate these segments together.

The construction of $D^{\text{epi,k}}$ stems from two intuitions: Firstly, $\pi^{-1,k}$'s gameplay style can become more evident over continuous timesteps, so we sample consecutive fragments. Secondly, $\pi^{-1,k}$ can exhibit diverse behaviors across different episodes, so we sample from multiple trajectories.

# D Proofs of Theorems

## D.1 Algorithm of Posterior Sampling in Opponent Modeling

We instantiate a Bayesian posterior sampling algorithm in the context of opponent modeling, referred to as **Posterior Sampling in Opponent Modeling (PSOM)**. In the PSOM algorithm, we use opponent trajectory $(s_0, a_0^{-1}, \ldots, s_{T-1}, a_{T-1}^{-1})$, which consists of consecutive $(s, a^{-1})$ tuples, to construct the in-context data $D$. Following up, we describe the PSOM algorithm in a most general way [61, 44].

Given the initial distribution of opponent policies $\Pi_0 \leftarrow \Pi_{\text{pre}}$, where $\Pi_{\text{pre}}$ is the *probability distribution* on $\Pi^{\text{train}}$, for $c \in [C]$:

1. Sample an opponent policy $\pi_c^{-1}$ by $\Pi_c$ and compute the BR policy $\pi_c^{1,*}$ for the self-agent;

2. Interact using the self-agent with $\pi_c^{1,*}$ against the opponent with *true opponent policy* $\bar{\pi}^{-1}$, and use the opponent trajectory $(s_0, a_0^{-1}, \ldots, s_{T-1}, a_{T-1}^{-1})$ to construct $D$.

3. Update the posterior distribution $\Pi_c(\pi^{-1}) = P(\pi^{-1}|D)$.

## D.2  Proof of Lemma 4.1

**Lemma 4.1** (Equivalence of OMIS w/o S and PSOM). *Assume that the learned $\pi_\theta$ is consistent and the sampling of $s$ from $\mathcal{T}_{pre}^{-1}$ is independent of opponent policy, then given $\bar{\pi}^{-1}$ and its D, we have $P(\xi_T^1|D, \bar{\pi}^{-1}; PSOM) = P(\xi_T^1|D, \bar{\pi}^{-1}; \pi_\theta)$ for all possible $\xi_T^1$.*

*Proof.* In this section, we use $\pi^{-1}$ to denote opponent policies posteriorly sampled from $\Pi_{\text{pre}}$ by the self-agent and use $\bar{\pi}^{-1}$ to denote the true opponent policy interacted with during the testing stage of OMIS. In the proof, we abbreviate OMIS without DTS as OMIS. For clarity and ease of understanding, all trajectory sequences are indexed starting from 1 in this section (originally starting from 0 in the main text). We abbreviate $D^{\text{epi}}$ as $D$ in the proof, as $D^{\text{epi}}$ is sufficient for completing the proof. Define $\xi_T^1 = (s_1, a_1^{1,*}, \ldots, s_T, a_T^{1,*})$ as *self-agent history*, where $T$ denotes the maximum length of this history (*i.e.*, horizon for each episode) and $a^{1,*}$ is sampled from the best response policy $\pi^{1,*}$ against $\pi^{-1}$. $\mathcal{T}_{\text{pre}}^{-1}(\cdot; \pi^{-1})$ denotes the *probability distribution on all the trajectories involving $\pi^{-1}$ during pretraining*.

Let $\pi^{-1} \sim \Pi_{\text{pre}}$ and $D$ contain sampled trajectory fragments of $\pi^{-1}$ and let $s_{\text{query}} \in \mathcal{S}, a^{1,*} \in \mathcal{A}^1, \xi_{T-1}^1 \in (\mathcal{S} \times \mathcal{A}^1)^{T-1}$ and $t \in [0, T-1]$ be arbitrary, the full joint probability distribution during OMIS's pretraining stage can be denoted as:

$$P_{\text{pre}}(\pi^{-1}, D, \xi_{T-1}^1, t, s_{\text{query}}, a^{1,*}) = \Pi_{\text{pre}}(\pi^{-1}) \mathcal{T}_{\text{pre}}^{-1}(D; \pi^{-1}) \mathfrak{S}_T(s_{1:T}) \mathcal{S}_{\text{query}}(s_{\text{query}}) \pi^{1,*}(a^{1,*}|s_{\text{query}}; \pi^{-1})$$
$$\times \text{Unif}[0, T-1] \prod_{i \in [T]} \pi^{1,*}(a_i^1|s_i; \pi^{-1}).$$
$$(11)$$

Herein, $\mathfrak{S}_T \in \Delta(\mathcal{S}^T)$, which is independent of opponent policy. $\mathcal{S}_{\text{query}}$ is set to the uniform over $\mathcal{S}$. In addition, we sample $t \sim \text{Unif}[0, T-1]$ and truncating $\xi_t^1$ from $\xi_{T-1}^1$ (or, equivalently, sample $\xi_t^1 \sim \Delta((\mathcal{S} \times \mathcal{A}^1)^t)$ directly).

We define the random sequences and subsequences of the *self-agent trajectory under PSOM algorithm* as $\Xi_{\text{PSOM}}(t; D) = (S_1^{\text{PSOM}}, A_1^{1,\text{PSOM}}, \ldots, S_t^{\text{PSOM}}, A_t^{1,\text{PSOM}})$. This trajectory is generated in the following manner:

$$\pi_{\text{PSOM}}^{-1} \sim P(\pi^{-1}|D), S_1^{\text{PSOM}} \sim \rho,$$
$$A_i^{1,\text{PSOM}} \sim \pi^{1,*}(\cdot|S_i^{\text{PSOM}}; \pi_{\text{PSOM}}^{-1}), A_i^{-1,\text{PSOM}} \sim \bar{\pi}^{-1}(\cdot|S_i^{\text{PSOM}}), i \geq 1,$$
$$S_{i+1}^{\text{PSOM}} \sim \mathcal{P}(\cdot|S_i^{\text{PSOM}}, A_i^{1,\text{PSOM}}, A_i^{-1,\text{PSOM}}), i \geq 2.$$

Within, $\rho$ denotes the initial distribution on $\mathcal{S}$. Analogously, we define the random sequences and subsequences of the *self-agent trajectory under OMIS algorithm* as $\Xi_{\text{pre}}(t; D) = (S_1^{\text{pre}}, A_1^{1,\text{pre}}, \ldots, S_t^{\text{pre}}, A_t^{1,\text{pre}})$. This trajectory is generated in the following manner:

$$S_1^{\text{pre}} \sim \rho,$$
$$A_i^{1,\text{pre}} \sim P_{\text{pre}}(\cdot|S_i^{\text{pre}}, D, \Xi_{\text{pre}}(i-1; D)), A_i^{-1,\text{pre}} \sim \bar{\pi}^{-1}(\cdot|S_i^{\text{pre}}), i \geq 1,$$
$$S_{i+1}^{\text{pre}} \sim \mathcal{P}(\cdot|S_i^{\text{pre}}, A_i^{1,\text{pre}}, A_i^{-1,\text{pre}}), i \geq 2.$$

To simplify matters, we will refrain from explicitly referencing $D$ for $\Xi$ in our notations, except when required to avoid confusion. Next, we introduce a common assumption to ensure the neural network fits the pretraining data distribution.

**Assumption D.1** (Learned $\pi_\theta$ is consistent). *For any given $(S_i^{pre}, D, \Xi_{pre}(i-1;D))$, $P_{pre}(A_i^{1,pre}|S_i^{pre}, D, \Xi_{pre}(i-1;D)) = \pi_\theta(A_i^{1,pre}|S_i^{pre}, D, \Xi_{pre}(i-1;D))$ for all possible $A_i^{1,pre}$.*

Upon Assump. D.1, we will limit our attention to $P_{\text{pre}}$ for the rest of the proof.

To prove that $\forall \xi_T^1, P(\xi_T^1|D, \bar{\pi}^{-1}; \text{PSOM}) = P(\xi_T^1|D, \bar{\pi}^{-1}; \pi_\theta)$ (*i.e.*, Lemma 4.1), it is equivalent to prove that

$$P(\Xi_{\text{PSOM}}(T) = \xi_T^1) = P(\Xi_{\text{pre}}(T) = \xi_T^1). \tag{12}$$

We will prove that $\forall t \in [T]$,

$$P(\Xi_{\text{PSOM}}(t) = \xi_t^1) = P(\Xi_{\text{pre}}(t) = \xi_t^1) \tag{13}$$

using **Mathematical Induction** and then introduce a lemma for the evidence of Eq. (12).

To begin with, we propose a lemma to assist in proving Eq. (13) for the base case when $t = 1$.

**Lemma D.2.** *If the sampling of $s$ from $\mathcal{T}_{pre}^{-1}$ is independent of opponent policy, then $P_{pre}(\pi^{-1}|D) = P(\pi_{PSOM}^{-1} = \pi^{-1}|D)$.*

*Proof.* Assuming the sampling of $s$ from $\mathcal{T}_{\text{pre}}^{-1}$ is independent of opponent policy, we have:

$$P(\pi_{\text{PSOM}}^{-1} = \pi^{-1}|D) \propto \Pi_{\text{pre}}(\pi^{-1})P(D|\pi^{-1}) \tag{14a}$$

$$\propto \Pi_{\text{pre}}(\pi^{-1}) \prod_{j \in [|D|]} \pi^{-1}(a_j^{-1}|s_j) \tag{14b}$$

$$\propto \Pi_{\text{pre}}(\pi^{-1}) \prod_{j \in [|D|]} \pi^{-1}(a_j^{-1}|s_j)\mathcal{T}_{\text{pre}}^{-1}(s_j) \tag{14c}$$

$$= \Pi_{\text{pre}}(\pi^{-1})\mathcal{T}_{\text{pre}}^{-1}(D; \pi^{-1}) \tag{14d}$$

$$\propto P_{\text{pre}}(\pi^{-1}|D). \tag{14e}$$

$\propto$ denotes that the two sides are equal up to multiplicative factors independent of $\pi^{-1}$. Eq. (14a) is derived through the Bayesian posterior probability formula. Eq. (14b) uses the fact that $s$ in posterior sampling is independent of opponent policy. Eq. (14c) holds because of the assumption that the sampling of $s$ from $\mathcal{T}_{\text{pre}}^{-1}$ is independent of opponent policy. Eq. (14d) uses the definition of $\mathcal{T}_{\text{pre}}^{-1}$. Eq. (14e) is derived through the Bayesian posterior probability formula. $\square$

Now, we prove that Eq. (13) holds when $t = 1$:

$$P(\Xi_{\text{PSOM}}(1) = \xi_1^1) = P(S_1^{\text{PSOM}} = s_1, A_1^{1,\text{PSOM}} = a_1^1) \tag{15a}$$

$$= \rho(s_1) \int_{\pi^{-1}} P(A_1^{1,\text{PSOM}} = a_1^1, \pi_{\text{PSOM}}^{-1} = \pi^{-1}|S_1^{\text{PSOM}} = s_1)\mathrm{d}\pi^{-1} \tag{15b}$$

$$= \rho(s_1) \int_{\pi^{-1}} \pi^{1,*}(a_1^1|s_1; \pi^{-1})P_{\text{PSOM}}(\pi_{\text{PSOM}}^{-1} = \pi^{-1}|D, S_1^{\text{PSOM}} = s_1)\mathrm{d}\pi^{-1} \tag{15c}$$

$$= \rho(s_1) \int_{\pi^{-1}} \pi^{1,*}(a_1^1|s_1; \pi^{-1})P_{\text{PSOM}}(\pi_{\text{PSOM}}^{-1} = \pi^{-1}|D)\mathrm{d}\pi^{-1} \tag{15d}$$

$$= \rho(s_1) \int_{\pi^{-1}} \pi^{1,*}(a_1^1|s_1; \pi^{-1})P_{\text{pre}}(\pi^{-1}|D)\mathrm{d}\pi^{-1} \tag{15e}$$

$$= \rho(s_1)P_{\text{pre}}(a_1^1|s_1, D) \tag{15f}$$

$$= P(\Xi_{\text{pre}}(1) = \xi_1^1). \tag{15g}$$

Eqs. (15a) to (15c), (15f) and (15g) are derived using Bayesian law of total probability and conditional probability formula based on Eq. (11). Eq. (15d) holds because the sampling of $s_1$ is independent of $\pi^{-1}$. Eq. (15e) is derived through Lem. D.2.

Next, we start proving Eq. (13) for the other cases when $t \neq 1$. We utilize the inductive hypothesis to demonstrate the validity of the entire statement. Suppose that $P(\Xi_{\text{PSOM}}(t-1) = \xi_{t-1}^1) =$

$P(\Xi_{\mathrm{pre}}(t-1) = \xi_{t-1}^1)$, since

$$P(\Xi_{\mathrm{PSOM}}(t) = \xi_t^1) =$$
$$P(\Xi_{\mathrm{PSOM}}(t-1) = \xi_{t-1}^1)P(S_t^{\mathrm{PSOM}} = s_t, A_t^{1,\mathrm{PSOM}} = a_t^1|\Xi_{\mathrm{PSOM}}(t-1) = \xi_{t-1}^1)$$

and

$$P(\Xi_{\mathrm{pre}}(t) = \xi_t^1) =$$
$$P(\Xi_{\mathrm{pre}}(t-1) = \xi_{t-1}^1)P(S_t^{\mathrm{pre}} = s_t, A_t^{1,\mathrm{pre}} = a_t^1|\Xi_{\mathrm{pre}}(t-1) = \xi_{t-1}^1),$$

to prove that $P(\Xi_{\mathrm{PSOM}}(t) = \xi_t^1) = P(\Xi_{\mathrm{pre}}(t) = \xi_t^1)$, it is equivalent to prove:

$$
\begin{aligned}
&P(S_t^{\mathrm{PSOM}} = s_t, A_t^{1,\mathrm{PSOM}} = a_t^1|\Xi_{\mathrm{PSOM}}(t-1) = \xi_{t-1}^1)\\
&= P(S_t^{\mathrm{pre}} = s_t, A_t^{1,\mathrm{pre}} = a_t^1|\Xi_{\mathrm{pre}}(t-1) = \xi_{t-1}^1).
\end{aligned}
\tag{16}
$$

By expanding Eq. (16), we can get:

$$
\begin{aligned}
&P(S_t^{\mathrm{PSOM}} = s_t, A_t^{1,\mathrm{PSOM}} = a_t^1|\Xi_{\mathrm{PSOM}}(t-1) = \xi_{t-1}^1)\\
&= \mathcal{P}(s_t|s_{t-1}, a_{t-1}^1; \bar{\pi}^{-1})P(A_t^{1,\mathrm{PSOM}} = a_t^1|S_t^{\mathrm{PSOM}} = s_t, \Xi_{\mathrm{PSOM}}(t-1) = \xi_{t-1}^1)
\end{aligned}
\tag{17a}
$$

$$
\begin{aligned}
&= \int_{a_{t-1}^{-1}} \mathcal{P}(s_t|s_{t-1}, a_{t-1}^1, a_{t-1}^{-1})\bar{\pi}^{-1}(a_{t-1}^{-1}|s_{t-1})\mathrm{d}a_{t-1}^{-1}\\
&\cdot \int_{\pi^{-1}} P(A_t^{1,\mathrm{PSOM}} = a_t^1, \pi_{\mathrm{PSOM}}^{-1} = \pi^{-1}|S_t^{\mathrm{PSOM}} = s_t, \Xi_{\mathrm{PSOM}}(t-1) = \xi_{t-1}^1)\mathrm{d}\pi^{-1}.
\end{aligned}
\tag{17b}
$$

In Eq. (17b), the first integral term is the same for PSOM and OMIS, while the term inside the second integral term satisfies:

$$
\begin{aligned}
&P(A_t^{1,\mathrm{PSOM}} = a_t^1, \pi_{\mathrm{PSOM}}^{-1} = \pi^{-1}|S_t^{\mathrm{PSOM}} = s_t, \Xi_{\mathrm{PSOM}}(t-1) = \xi_{t-1}^1)\\
&= \pi^{1,*}(a_t^1|s_t; \pi^{-1})P(\pi_{\mathrm{PSOM}}^{-1} = \pi^{-1}|S_t^{\mathrm{PSOM}} = s_t, \Xi_{\mathrm{PSOM}}(t-1) = \xi_{t-1}^1).
\end{aligned}
\tag{18}
$$

Based on Eq. (17) and Eq. (18), to prove that Eq. (16) holds, it is equivalent to prove:

$$P(\pi_{\mathrm{PSOM}}^{-1} = \pi^{-1}|S_t^{\mathrm{PSOM}} = s_t, \Xi_{\mathrm{PSOM}}(t-1) = \xi_{t-1}^1) = P_{\mathrm{pre}}(\pi^{-1}|s_t, D, \xi_{t-1}^1).\tag{19}$$

We prove that Eq. (19) holds through the following derivation:

$$P(\pi_{\mathrm{PSOM}}^{-1} = \pi^{-1}|S_t^{\mathrm{PSOM}} = s_t, \Xi_{\mathrm{PSOM}}(t-1) = \xi_{t-1}^1)$$

$$= \frac{P(S_t^{\mathrm{PSOM}} = s_t, \Xi_{\mathrm{PSOM}}(t-1) = \xi_{t-1}^1|\pi_{\mathrm{PSOM}}^{-1} = \pi^{-1})P(\pi_{\mathrm{PSOM}}^{-1} = \pi^{-1}|D)}{P(S_t^{\mathrm{PSOM}} = s_t, \Xi_{\mathrm{PSOM}}(t-1) = \xi_{t-1}^1)}\tag{20a}$$

$$\propto P_{\mathrm{pre}}(\pi^{-1}|D)\prod_{i\in[t-1]}\mathcal{P}(s_{i+1}|\xi_i^1, \bar{\pi}^{-1})\pi^{1,*}(a_i^1|s_i; \pi^{-1})\tag{20b}$$

$$\propto P_{\mathrm{pre}}(\pi^{-1}|D)\prod_{i\in[t-1]}\pi^{1,*}(a_i^1|s_i; \pi^{-1})\tag{20c}$$

$$\propto P_{\mathrm{pre}}(\pi^{-1}|D)\mathfrak{S}_{\mathrm{query}}(s_t)\mathfrak{S}_{t-1}(s_{1:t-1})\prod_{i\in[t-1]}\pi^{1,*}(a_i^1|s_i; \pi^{-1})\tag{20d}$$

$$\propto P_{\mathrm{pre}}(\pi^{-1}|s_t, D, \xi_{t-1}^1).\tag{20e}$$

Eq. (20a) is derived through the Bayesian posterior probability formula. In Eq. (20b), we decompose the probability of observing the sequence of observations $s$ and actions $a^1$. Eqs. (20c) and (20d) use the fact that the sampling of $s$ is only related to the true opponent policy $\bar{\pi}^{-1}$ and is independent of $\pi^{-1}$. Eq. (20e) is derived by the definition of $P_{\mathrm{pre}}(\pi^{-1}|s_t, D, \xi_{t-1}^1)$.

Therefore, we finish the proof of $P(\Xi_{\mathrm{PSOM}}(t) = \xi_t^1) = P(\Xi_{\mathrm{pre}}(t) = \xi_t^1)$, where

$$P(\Xi_{\mathrm{PSOM}}(t) = \xi_t^1)$$

$$= P(\Xi_{\text{pre}}(t-1) = \xi^1_{t-1})\mathcal{P}(s_t|s_{t-1}, a^1_{t-1}; \bar{\pi}^{-1}) \int_{\pi^{-1}} \pi^{1,*}(a^1_t|s_t; \pi^{-1}) P_{\text{pre}}(\pi^{-1}|s_t, D, \xi^1_{t-1}) \mathrm{d}\pi^{-1} \tag{21a}$$

$$= P(\Xi_{\text{pre}}(t-1) = \xi^1_{t-1})\mathcal{P}(s_t|s_{t-1}, a^1_{t-1}; \bar{\pi}^{-1}) P_{\text{pre}}(a^1_t|s_t, D, \xi^1_{t-1}) \tag{21b}$$

$$= P(\Xi_{\text{pre}}(t) = \xi^1_t). \tag{21c}$$

Based on Mathematical Induction, Eq. (13) holds for any $t \in [T]$. Hence, Eq. (12) is satisfied. This concludes the proof. $\square$

### D.3 Proof of Theorem 4.2

**Theorem 4.2.** *When $\bar{\pi}^{-1} = \pi^{-1,k} \in \Pi^{train}$, if the PS algorithm converges to the optimal solution, then OMIS w/o S recognizes the policy of $\Phi$ as $\pi^{-1,k}$, i.e., $\pi_\theta$, $\mu_\phi$, and $V_\omega$ converge to $\pi^{1,k,*}$, $\pi^{-1,k}$, and $V^{1,k,*}$, respectively; When $\bar{\pi}^{-1} \notin \Pi^{train}$, OMIS w/o S recognizes the policy of $\Phi$ as the policies in $\Pi^{train}$ with the minimum $D_{KL}(P(a^{-1}|s, \pi^{-1})||P(a^{-1}|s, \bar{\pi}^{-1}))$.*

*Proof.* In the proof, we abbreviate OMIS without DTS as OMIS. We denote the in-context data consisting of $(s, a^{-1})$ tuples as $D$, and the in-context data consisting of $(s, a^1, s'^1, r^1)$ tuples as $D'$, where $s'$ is the next state of $s$ transitioned to. Note that the original PS algorithm uses $D'$ while PSOM and OMIS use $D$ to recognize the policy of $\Phi$.

To begin, we propose a lemma and its corollary to prove the convergence guarantee of the PSOM algorithm and to analyze its properties in opponent policy recognition.

**Lemma D.3.** *Let $f(\pi^{-1}; D) = -\int_{s,a^{-1}} P(s, a^{-1}; D) \log(P(a^{-1}|s, \pi^{-1})) \mathrm{d}s \mathrm{d}a^{-1}$ and $\pi_\star^{-1} = \arg\min_{\pi^{-1} \in \Pi^{train}} f(\pi^{-1}; D)$, then $\forall \pi^{-1} \in \{\pi^{-1}|f(\pi^{-1}; D) \neq f(\pi_\star^{-1}; D)\}$, we have $\frac{P(\pi^{-1}|(s,a^{-1})_{1:n})}{P(\pi_\star^{-1}|(s,a^{-1})_{1:n})} \xrightarrow{P} 0$.*

*Proof.* Here, $\pi_\star^{-1}$ denotes the equivalent class of opponent policies to which PSOM converges with non-zero probability. $P(s, a^{-1}; D)$ is the distribution of $(s, a^{-1})$ tuples in $D$. $n$ is the number of the $(s, a^{-1})$ tuples. To prove that $\frac{P(\pi^{-1}|(s,a^{-1})_{1:n})}{P(\pi_\star^{-1}|(s,a^{-1})_{1:n})} \xrightarrow{P} 0$ under the given conditions, it is equivalent to prove:

$$L_{\pi^{-1},n} = -\log \frac{P(\pi^{-1}|(s,a^{-1})_{1:n})}{P(\pi_\star^{-1}|(s,a^{-1})_{1:n})} \xrightarrow{P} +\infty. \tag{22}$$

By expanding Eq. (22), we can get:

$$L_{\pi^{-1},n} = -\log \frac{P(\pi^{-1}|(s,a^{-1})_{1:n})}{P(\pi_\star^{-1}|(s,a^{-1})_{1:n})} = -\log \frac{P(\pi^{-1})}{P(\pi_\star^{-1})} - \sum_{i=1}^n \log(\frac{P(a^{-1}|\pi^{-1}, s)}{P(a^{-1}|\pi_\star^{-1}, s)}). \tag{23}$$

According to the definition of $\pi_\star^{-1}$ and the condition $f(\pi^{-1}; D) \neq f(\pi_\star^{-1}; D)$, we have:

$$\mathbb{E}_{(s,a^{-1})\sim P(\cdot; D)}[-\sum_{i=1}^n \log(\frac{P(a^{-1}|\pi^{-1}, s)}{P(a^{-1}|\pi_\star^{-1}, s)})] = f(\pi^{-1}; D) - f(\pi_\star^{-1}; D) = \mathcal{C} > 0. \tag{24}$$

Here, $\mathcal{C}$ is a positive constant. Therefore, based on the law of large numbers, we have $\lim_{n\to\infty} P(|\frac{L_{\pi^{-1},n}}{n} - \mathcal{C}| > \epsilon) = 0$, where $\epsilon$ is any positive number. Hence, Eq. (22) is satisfied, and the proof ends. $\square$

**Corollary D.4** (Corollary of Lem. D.3). *When $\bar{\pi}^{-1} \in \Pi^{train}$, we have $\bar{\pi}^{-1} \in \pi_\star^{-1}$; When $\bar{\pi}^{-1} \notin \Pi^{train}$, $\pi_\star^{-1}$ is the equivalent class of policies in $\Pi^{train}$ with the minimum $D_{KL}(P(a^{-1}|s, \pi^{-1})||P(a^{-1}|s, \bar{\pi}^{-1}))$.*

*Proof.* Since $P(s, a^{-1}; D) = P(s; D)P(a^{-1}|s, \bar{\pi}^{-1})$ holds, we have

$$\pi_\star^{-1} = \arg\min_{\pi^{-1} \in \Pi^{train}} -\int_{s,a^{-1}} P(s, a^{-1}; D) \log(P(a^{-1}|s, \pi^{-1})) \mathrm{d}s \mathrm{d}a^{-1} \tag{25a}$$

$$= \arg\min_{\pi^{-1}\in\Pi^{\text{train}}} - \int_{s,a^{-1}} P(s;D)P(a^{-1}|s,\bar{\pi}^{-1})\log(P(a^{-1}|s,\pi^{-1}))\mathrm{d}s\mathrm{d}a^{-1} \tag{25b}$$

$$= \arg\min_{\pi^{-1}\in\Pi^{\text{train}}} - \int_{s,a^{-1}} P(s;D)P(a^{-1}|s,\bar{\pi}^{-1})\log(P(a^{-1}|s,\pi^{-1}))\mathrm{d}s\mathrm{d}a^{-1}$$
$$+ \int_{s,a^{-1}} P(s;D)P(a^{-1}|s,\bar{\pi}^{-1})\log(P(a^{-1}|s,\bar{\pi}^{-1}))\mathrm{d}s\mathrm{d}a^{-1} \tag{25c}$$

$$= \arg\min_{\pi^{-1}\in\Pi^{\text{train}}} - \int_{s,a^{-1}} P(s;D)P(a^{-1}|s,\bar{\pi}^{-1})\frac{\log(P(a^{-1}|s,\pi^{-1}))}{\log(P(a^{-1}|s,\bar{\pi}^{-1}))}\mathrm{d}s\mathrm{d}a^{-1} \tag{25d}$$

$$= \arg\min_{\pi^{-1}\in\Pi^{\text{train}}} \int_s P(s;D)D_{KL}(P(a^{-1}|s,\pi^{-1})||P(a^{-1}|s,\bar{\pi}^{-1}))\mathrm{d}s \tag{25e}$$

$$= \arg\min_{\pi^{-1}\in\Pi^{\text{train}}} \mathbb{E}_{s\sim P(\cdot;D)}\left[D_{KL}(P(a^{-1}|s,\pi^{-1})||P(a^{-1}|s,\bar{\pi}^{-1}))\right] \tag{25f}$$

When $D$ is the in-context data of the opponent policy $\bar{\pi}^{-1} \in \Pi^{\text{train}}$, $D_{KL}(P(a^{-1}|s,\pi^{-1})||P(a^{-1}|s,\bar{\pi}^{-1})) = 0$ in Eq. (25f), and $\pi_\star^{-1}$ is the equivalent class of opponent policies that have the same action distribution as $\bar{\pi}^{-1}$, *i.e.*, $\forall a^{-1}, P(a^{-1}|s,\pi_\star^{-1}) = P(a^{-1}|s,\bar{\pi}^{-1})$. When $D$ is the in-context data of an opponent policy $\bar{\pi}^{-1} \notin \Pi^{\text{train}}$, $\pi_\star^{-1}$ is the equivalent class of opponent policies that minimizes the expected KL divergence $D_{KL}(P(a^{-1}|s,\pi^{-1})||P(a^{-1}|s,\bar{\pi}^{-1}))$. □

Using a similar proving method as in Lem. D.3, it can be proved straightforward that the PS algorithm can converge: Let $f(\pi^{-1};D') = -\int_{s,a^1,s'^1,r^1} P(s,a^1,s'^1,r^1;D')\log(P(s'^1,r^1|s,a^1,\pi^{-1}))\mathrm{d}s\mathrm{d}a^1\mathrm{d}s'^1\mathrm{d}r^1$ and $\pi_\star'^{-1} = \arg\min_{\pi^{-1}\in\Pi^{\text{train}}} f(\pi^{-1};D')$, then $\forall \pi^{-1} \in \{\pi^{-1}|f(\pi^{-1};D') \neq f(\pi_\star'^{-1};D')\}$, we have $\frac{P(\pi^{-1}|(s,a^1,s'^1,r^1)_{1:n})}{P(\pi_\star'^{-1}|(s,a^1,s'^1,r^1)_{1:n})} \xrightarrow{P} 0$.

Next, we introduce a lemma to prove that if the PS algorithm converges to the optimal solution, PSOM converges to the optimal solution.

**Lemma D.5.** *Given $s, a^1, \bar{\pi}^{-1}, \pi_\star^{-1}$, if $\forall a^{-1}, P(a^{-1}|s,\pi_\star^{-1}) = P(a^{-1}|s,\bar{\pi}^{-1})$ holds, it can be deduced that $\forall s'^1, r^1, P(s'^1,r^1|s,a^1,\pi_\star^{-1}) = P(s'^1,r^1|s,a^1,\bar{\pi}^{-1})$, but the reverse is not true.*

*Proof.* For the forward deduction (*i.e.*, ⇒), we have:

$$\forall s'^1, r^1, P(s'^1,r^1|s,a^1,\pi_\star^{-1})$$
$$= \sum_{a^{-1}} P(a^{-1}|s,\pi_\star^{-1})P(s'^1,r^1|s,a^1,s',a^{-1}) \tag{26a}$$
$$= \sum_{a^{-1}} P(a^{-1}|s,\bar{\pi}^{-1})P(s'^1,r^1|s,a^1,s',a^{-1}) \tag{26b}$$
$$= P(s'^1,r^1|s,a^1,\bar{\pi}^{-1}). \tag{26c}$$

For the backward deduction (*i.e.*, ⇐), counterexamples exist. For example, when $P(s'^1,r^1|s,a^1,s',a^{-1})$ takes equal values for some $a^{-1} \in \bar{\mathcal{A}}^{-1} \subset \mathcal{A}^{-1}$ and $\forall a^{-1} \in \mathcal{A}^{-1}\backslash\bar{\mathcal{A}}^{-1}, P(a^{-1}|s,\pi_\star^{-1}) = P(a^{-1}|s,\bar{\pi}^{-1})$; $\sum_{a^{-1}\in\bar{\mathcal{A}}^{-1}} P(a^{-1}|s,\pi_\star^{-1}) = \sum_{a^{-1}\in\bar{\mathcal{A}}^{-1}} P(a^{-1}|s,\bar{\pi}^{-1})$ hold, $P(a^{-1}|s,\pi_\star^{-1}) = P(a^{-1}|s,\bar{\pi}^{-1})$ does not necessarily hold for all $a^{-1} \in \mathcal{A}^{-1}$. This means when $\bar{\pi}^{-1} \in \Pi^{\text{train}}$, the PS algorithm may lead to distributions on opponent policies with non-zero probability other than the equivalence class of $\bar{\pi}^{-1}$, resulting in potential suboptimality compared to PSOM. □

According to Lem. D.5, $\pi_\star^{-1} \subset \pi_\star'^{-1}$. Based on Cor. D.4, we have $\bar{\pi}^{-1} \in \pi_\star^{-1}$. Thus, we conclude that $\bar{\pi}^{-1} \in \pi_\star^{-1} \subset \pi_\star'^{-1}$. Hence, if PS converges to the optimal solution, PSOM converges to the optimal solution.

Based on Lemma 4.1, it can be inferred that OMIS is equivalent to PSOM. Thus, all the proofs in this section also hold for OMIS. We have the following conclusions:

1. Based on Cor. D.4 and Lem. D.5, when $\bar{\pi}^{-1} \in \Pi^{\text{train}}$, if the PS algorithm converges to the optimal solution, then OMIS converges to the optimal solution. If the true opponent policy is $\pi^{-1,k}$, OMIS recognizes the current policy of $\Phi$ as $\pi^{-1,k}$. In this case, $\pi_\theta$ converges to $\pi^{1,k,*}$. Similarly to the PSOM algorithm in App. D.1, when we replace $\pi^{1,*}$ with $\pi^{-1}$ and $V^{1,*}$, we can derive the algorithms with the same theoretical guarantees. Thus, $\mu_\phi$ and $V_\omega$ converge to $\pi^{-1,k}$ and $V^{1,k,*}$, respectively.

2. Based on Cor. D.4, when $\bar{\pi}^{-1} \notin \Pi^{\text{train}}$, OMIS recognizes the current policy of $\Phi$ as the policies in $\Pi^{\text{train}}$ with the minimum KL divergence $D_{KL}(P(a^{-1}|s, \pi^{-1})||P(a^{-1}|s, \bar{\pi}^{-1}))$.

This concludes the proof. $\qquad\square$

### D.4 Proof of Theorem 4.3

**Theorem 4.3** (Policy Improvement of OMIS's DTS). *Given $\bar{\pi}^{-1}$ and its $D$, suppose OMIS recognizes $\Phi$ as $\pi_\star^{-1}$ and $V_{\pi_\star^{-1}}^\pi$ is the value vector on $\mathcal{S}$, where $V(s) := V_\omega(s, D), \pi(a|s) := \pi_\theta(a|s, D)$. Let $\mathcal{G}_L$ be the $L$-step DTS operator and $\pi' \in \mathcal{G}_L(V_{\pi_\star^{-1}}^\pi)$, then $V_{\pi_\star^{-1}}^{\pi'} \geq V_{\pi_\star^{-1}}^\pi$ holds component-wise.*

*Proof.* To begin with, we do not consider the mixing technique in the proof. Based on Theorem 4.2, given $\bar{\pi}^{-1}$ and its $D$, OMIS recognize the policy of $\Phi$ as $\pi_\star^{-1}$, which means $\pi_\theta, \mu_\phi$, and $V_\omega$ converge to $\pi^{1,\star}, \pi_\star^{-1}$, and $V^{1,\star}$, respectively. When $\bar{\pi}^{-1} \in \Pi^{\text{train}}$, since the labels in the pretraining data may not be optimal, there is space for improvement in the $\pi$ (*i.e.*, $\pi^{1,\star}$). When $\bar{\pi}^{-1} \notin \Pi^{\text{train}}$, $\pi$ may not be the best response against $\bar{\pi}^{-1}$, thus there is still space for policy improvement. Furthermore, disregarding the impact of $D_t^{\text{step}}$ on $\mu_\phi$, $\mu_\phi$ can be treated as a fixed policy during the DTS process. Thus, the virtual environment for the DTS is stationary.

A $L$-greedy policy w.r.t. the value function $V_{\pi_\star^{-1}}$, belongs to the following set of policies,

$$\arg\max_{\pi_0} \max_{\pi_1,\dots,\pi_{L-1}} \mathbb{E}_{|\cdot}^{\pi_0 \dots \pi_{L-1}} \left[ \sum_{l=0}^{L-1} \gamma_{\text{search}}^l R(s_l, \pi_l(s_l); \pi_\star^{-1}) + \gamma_{\text{search}}^L V_{\pi_\star^{-1}}(s_L) \right] \tag{27a}$$

$$= \arg\max_{\pi_0} \mathbb{E}_{|\cdot}^{\pi_0} \left[ R(s_0, \pi_0(s_0); \pi_\star^{-1}) + \gamma_{\text{search}}(\mathcal{T}^{L-1} V_{\pi_\star^{-1}})(s_1) \right] \tag{27b}$$

where the notation $\mathbb{E}_{|\cdot}^{\pi_0 \dots \pi_{L-1}}$ means that we condition on the trajectory induced by the choice of actions $(\pi_0(s_0), \pi_1(s_1), \dots, \pi_{L-1}(s_{L-1}))$ and the starting state $s_0 = \cdot$.[3] The $\pi_\star^{-1}$ terms in $R$ means opponents take actions by $\mu_\phi$ conditioned on $D$. We define $\mathcal{T}^{\pi^1}$ as the operator choosing actions using $\pi^1$ for one step, where $\pi^1$ is any self-agent policy. We define $\mathcal{T}$ as the Bellman optimality operator, where

$$\mathcal{T} V_{\pi_\star^{-1}} = \max_{\pi^1} \mathcal{T}^{\pi^1} V_{\pi_\star^{-1}}. \tag{28}$$

Following up, we define $\mathcal{T}^L$ (shown in Eq. (27b)) as the $L$-step Bellman optimality operator, where

$$\mathcal{T}^L V_{\pi_\star^{-1}} = \max_{\pi_0,\dots,\pi_{L-1}} \mathbb{E}_{|\cdot}^{\pi_0 \dots \pi_{L-1}} \left[ \sum_{l=0}^{L-1} \gamma_{\text{search}}^l R(s_l, \pi_l(s_l); \pi_\star^{-1}) + \gamma_{\text{search}}^L V_{\pi_\star^{-1}}(s_L) \right]. \tag{29}$$

Since the argument in Eq. (27b) is a vector, the maximization is component-wise, *i.e.*, we want to find the choice of actions that will jointly maximize the entries of the vector. Thus, the $L$-greedy policy chooses the first optimal action of a non-stationary, optimal control problem with horizon $L$. Since $\pi$ is maximized to select actions during OMIS's DTS, Eq. (9) can be considered equivalent to Eq. (27).[4]

The set of $L$-greedy policies w.r.t. $V_{\pi_\star^{-1}}$, *i.e.*, the $L$-step DTS operator, $\mathcal{G}_L(V_{\pi_\star^{-1}})$, can be expressed as follows:

$$\forall V_{\pi_\star^{-1}}, \pi^1, \mathcal{T}_L^{\pi^1} V_{\pi_\star^{-1}} \stackrel{\text{def}}{=} \mathcal{T}^{\pi^1} \mathcal{T}^{L-1} V_{\pi_\star^{-1}}, \tag{30a}$$

---

[3]We use $\pi_l, l = 0, \dots, L - 1$ to denote $\pi$ at each step for simplicity as they can be different based on $D$.

[4]In Eq. (8), the number of rollout steps is actually $L + 1$ as we need to perform a rollout of 1 step for each legal action first. However, this does not affect the conclusions.

$$\forall V_{\pi_\star^{-1}}, \mathcal{G}_L(V_{\pi_\star^{-1}}) = \{\pi' : \mathcal{T}_L^{\pi'} V_{\pi_\star^{-1}} = \mathcal{T}^L V_{\pi_\star^{-1}}\}. \tag{30b}$$

For Eq. (30a), the operator $\mathcal{T}^{\pi^1}\mathcal{T}^{L-1}$ represents choosing actions using $\pi^1$ in the first step and selecting the optimal actions from all possibilities in the subsequent $L-1$ steps. For Eq. (30b), the policy $\pi'$ derived from $\mathcal{G}_L(V_{\pi_\star^{-1}})$ satisfies that choosing actions using $\pi$ in the first step and selecting the optimal actions from all possibilities in the subsequent $L-1$ steps is equivalent to choosing all possible optimal actions in all L steps.

We observe that

$$V_{\pi_\star^{-1}} = \mathcal{T}^\pi V_{\pi_\star^{-1}} \leq \mathcal{T} V_{\pi_\star^{-1}}. \tag{31}$$

Then, sequentially using Eqs. (30a), (30b) and (31), we have

$$V_{\pi_\star^{-1}} = (\mathcal{T}^\pi)^L V_{\pi_\star^{-1}} \leq \mathcal{T}^L V_{\pi_\star^{-1}} = \mathcal{T}_L^{\pi'} V_{\pi_\star^{-1}} = \mathcal{T}^{\pi'}(\mathcal{T}^{L-1} V_{\pi_\star^{-1}}). \tag{32}$$

This leads to the following inequalities:

$$V_{\pi_\star^{-1}} \leq \mathcal{T}^{\pi'}(\mathcal{T}^{L-1} V_{\pi_\star^{-1}}) \tag{33a}$$

$$\leq \mathcal{T}^{\pi'}(\mathcal{T}^{L-1} \mathcal{T} V_{\pi_\star^{-1}}) = \mathcal{T}^{\pi'}(\mathcal{T}^L V_{\pi_\star^{-1}}) \tag{33b}$$

$$= \mathcal{T}^{\pi'}\left(\mathcal{T}^{\pi'}\mathcal{T}^{L-1} V_{\pi_\star^{-1}}\right) = \left(\mathcal{T}^{\pi'}\right)^2 (\mathcal{T}^{L-1} V_{\pi_\star^{-1}}) \tag{33c}$$

$$\leq \cdots$$

$$\leq \lim_{n \to \infty} \left(\mathcal{T}^{\pi'}\right)^n (\mathcal{T}^{L-1} V_{\pi_\star^{-1}}) = V_{\pi_\star^{-1}}^{\pi'}. \tag{33d}$$

Within, Eq. (33a) holds because of Eq. (32), Eq. (33b) is due to Eq. (31) and the monotonicity of $\mathcal{T}^{\pi'}$ and $\mathcal{T}$ (and thus of their composition), Eq. (33c) is derived by Eq. (32), and Eq. (33d) is due to the fixed point property of $\mathcal{T}^{\pi'}$. Lastly, notice that $V_{\pi_\star^{-1}} = V_{\pi_\star^{-1}}^{\pi'}$ if and only if (see Eq. (31)) $\mathcal{T} V_{\pi_\star^{-1}} = V_{\pi_\star^{-1}}$, which holds if and only if $\pi$ is the optimal policy. This concludes the proof.

$\square$

# E Detailed Introductions of the Environments

`Predator Prey` [52] is a competitive environment with a three vs. one setup and a continuous state space. The environment consists of three predators (in red), one prey (in green), and two obstacles (in black). The goal of the predators is to capture (*i.e.*, collide with) the prey as much as possible, while the goal of the prey is to be captured as little as possible. The environment features sparse rewards, where each time a predator captures the prey, the capturing predator receives a reward of $10$, and the prey receives a reward of $-10$. Additionally, the environment provides a small, dense reward to the prey to prevent it from running out of the map boundaries. Here, the prey is the self-agent, and the three predators serve as opponents. From the perspective of the self-agent, the environment is highly unstable, as there are three opponents with unknown policies in the environment. The challenge in this environment is that the self-agent must model the behavior of three opponents simultaneously and adapt to various potential coordination strategies employed by the opponents (*e.g.*, surrounding from three different directions). For specific implementation of this environment, we adopt the open-source code of `Multi-Agent Particle Environment`, which is available at `https://github.com/openai/multiagent-particle-envs`.

`Level-Based Foraging` [16, 64] is a mixed environment in a 9×9 grid world containing two players: the self-agent (in blue) and the opponent (in black), along with five apples (in red). At the beginning of each episode, the two players and the five apples are randomly generated in the environment and assigned a level marked in their bottom-right corner. The goal of the self-agent is to eat as many apples as possible. All players can move in four directions or eat an apple. Eating an apple can be successfully done only under the following conditions: one or two players are around the apple, and all players who take the action of eating an apple have a summed level at least equal to the level of the apple. The environment has sparse rewards, representing the players' contributions to eating all the apples in the environment. The environment is essentially a long-term social dilemma and can be viewed as an extension of the Prisoner's Dilemma [6]. The challenge in this environment is that the self-agent must learn to cooperate to eat high-level apples while greedily eating low-level apples simultaneously. For specific implementation of this environment, we adopt the open-source code of `lb-foraging`, which is available at `https://github.com/semitable/lb-foraging`.

`OverCooked` [14] is a cooperative environment with high-dimensional images as states. One of the chefs is the self-agent (green), while the other chef is the opponent (blue). The two chefs must collaborate to complete a series of subtasks and serve dishes. All players share the same sparse rewards, earning $20$ for each successful dish served. The more successful the dish servings, the higher the reward. The goal of the self-agent is to collaborate as effectively as possible with the other chef to maximize the return. The challenge in this environment is for the self-agent to not only be able to complete subtasks such as getting onions, putting onions into the pot, and serving dishes but also to coordinate intensively with the opponent. It requires the self-agent to allocate subtasks effectively with the opponent, ensuring that it does not negatively impact the opponent (*e.g.*, not blocking the opponent's path). For specific implementation of this environment, we adopt the open-source code of `Overcooked-AI`, which is available at `https://github.com/HumanCompatibleAI/overcooked_ai`.

# F Neural Architecture Design

For **OMIS**, we adopt the neural architecture design as follows:

The *backbone* of the OMIS architecture is mainly implemented based on the causal Transformer, *i.e.*, GPT2 [67] model of Hugging Face [93]. The backbone is a GPT2 decoder composed of 3 self-attention blocks. Each self-attention block consists of a single-head attention layer and a feed-forward layer. Residual connections [30] and LayerNorm [7] are utilized after each layer in the self-attention block. Within each attention layer, dropout [79] is added to the residual connection and attention weight.

In the backbone, except for the fully connected layer in the feed-forward layer (the feed-forward layer consists of a fully connected layer that increases the number of hidden layer nodes and a projection layer that recovers the number of hidden layer nodes), which consists of $128$ nodes with GELU [31] activation functions, the other hidden layers are composed of $32$ nodes without activation functions. The modality-specific linear layers for self-agent actions, opponents actions, and RTGs comprise $32$

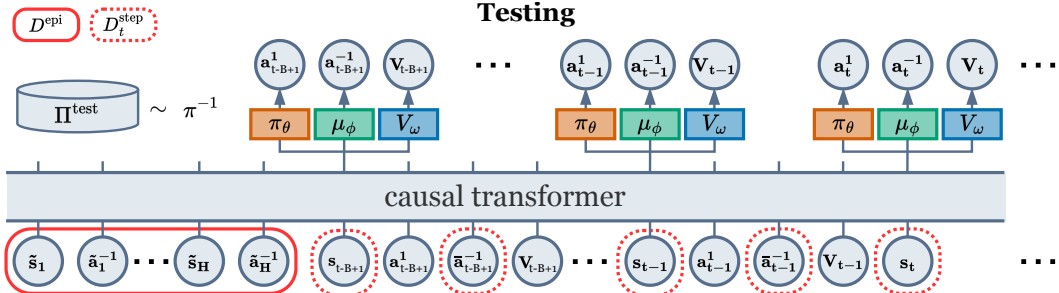

Figure 10: The architecture of OMIS during testing.

nodes without activation functions. The modality-specific linear (and convolutional) layers for states comprise 32 nodes with LeakyReLU [96] activation functions.

For *input encoding*, states $s$, self-agent actions $a^1$, opponents actions $a^{-1}$, RTGs $G^1$ are fed into modality-specific linear layers. For `OC`, additional convolutional layers are added before the linear layers to encode the state $s$ better. A positional episodic timestep encoding is added. We adopt the same timestep encoding as in Chen et al. [15]. In addition, an agent index encoding is added to each token to distinguish the inputs from different agents.

For *output decoding*, the sequences of embedded tokens are fed into the backbone, which autoregressively outputs the self-agent actions $a^1$, opponents actions $a^{-1}$, values $V$ at the positions of state $s$ tokens using a causal self-attention mask. The $\pi_\theta$ who outputs $a^1$, the $\mu_\phi$ who outputs $a^{-1}$, and the $V_\omega$ who outputs $V$ all consists of linear layers.

During pretraining, for each timestep $t$ given $\pi^{-1,k} \sim \Pi^{\text{train}}$, the input sequence is $(\tilde{s}_1, \tilde{a}_1^{-1,k}, \ldots, \tilde{s}_H, \tilde{a}_H^{-1,k}, \quad s_{t-B+1}, a_{t-B+1}^{1,k,*}, a_{t-B+1}^{-1,k}, G_{t-B+1}^{1,k,*}, \ldots, s_{t-1}, a_{t-1}^{1,k,*}, a_{t-1}^{-1,k}, G_{t-1}^{1,k,*}, s_t)$, where $B$ is the maximum sequence length as Transformer model has a token capacity. The output prediction sequence is $(a_{t-B+1}^1, a_{t-B+1}^{-1}, V_{t-B+1}, \ldots, a_{t-1}^1, a_{t-1}^{-1}, V_{t-1})$. The output label sequence is $(a_{t-B+1}^{1,k,*}, a_{t-B+1}^{-1,k}, G_{t-B+1}^{1,k,*}, \ldots, a_{t-1}^{1,k,*}, a_{t-1}^{-1,k}, G_{t-1}^{1,k,*})$.

During testing, for each *timestep* $t$ given $\Phi = \bar{\pi}^{-1}$, $\bar{\pi}^{-1} \sim \Pi^{\text{test}}$,[5] the input sequence is $(\tilde{s}_1, \tilde{a}_1^{-1}, \ldots, \tilde{s}_H, \tilde{a}_H^{-1}, s_{t-B+1}, a_{t-B+1}^1, \bar{a}_{t-B+1}^{-1}, V_{t-B+1}, \ldots, s_{t-1}, a_{t-1}^1, \bar{a}_{t-1}^{-1}, V_{t-1}, s_t)$, where $\bar{a}^{-1}$ is the true actions of $\Phi$. The output sequence is $(a_{t-B+1}^1, a_{t-B+1}^{-1}, V_{t-B+1}, \ldots, a_{t-1}^1, a_{t-1}^{-1}, V_{t-1})$. We demonstrated the architecture of OMIS during pretraining in Fig. 1, see the architecture of OMIS during testing in Fig. 10.

For **all the baselines**, we adopt the neural architecture design as follows:

We replace the original *backbone* of the baselines (*e.g.*, linear layers, recurrent layers, LSTM [56], and more) with the same GPT2 backbone as OMIS. For *input encoding* and *input encoding*, we encode and decode states $s$ and actions $a$ using the same modality-specific layers as OMIS. We encode and decode rewards $r$ using the modality-specific layers used to encode RTGs in OMIS. Note that we only modified the neural architectures of all the baselines to ensure fair comparisons. All the baselines are still pretrained and tested according to their respective methodologies.

## G Diversity of Opponent Policies

As mentioned in Sec. 5.1, we run the Maximum Entropy Population-based training algorithm (MEP) to generate a diversified opponent policy population. Nevertheless, a quantitative analysis is still necessary to measure the similarity/dissimilarity between different opponent policies within the MEP population. We calculate the pair-wise KL divergence between different opponent policies to measure their dissimilarity. The results for `PP`, `LBF`, and `OC` are shown in Figs. 11 to 13, respectively.

---

[5]This timestep $t$ can be the real timestep, and it also can be a virtual timestep during the DTS.

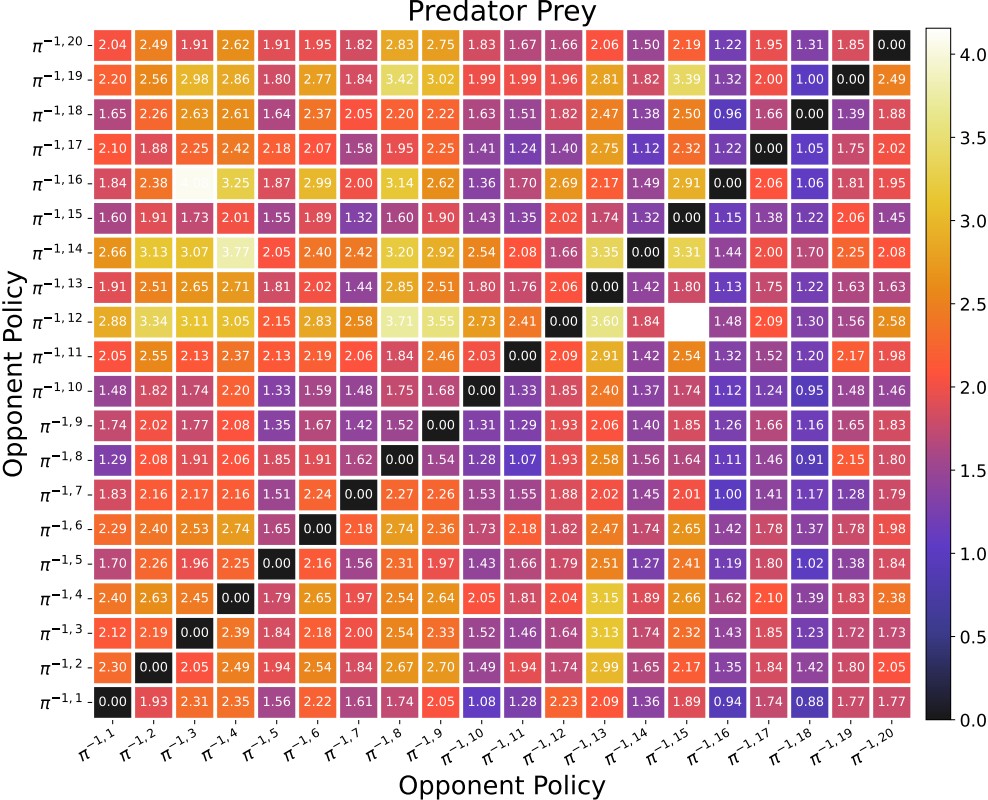

Figure 11: Pair-wise KL divergence of all policies within MEP population in PP

For any given policies $\pi_i$ and $\pi_j$, we estimate the KL divergence between them by:

$$D_{KL}(\pi_i || \pi_j) = \mathbb{E}_{s \sim P(s)} \left[ \sum_{a \in \mathcal{A}} \pi_i(a|s) \cdot \log \frac{\pi_i(a|s)}{\pi_j(a|s)} \right]. \tag{34}$$

Here, $P(s)$ denotes the state distribution. Ideally, $P(s)$ should cover the entire state space $\mathcal{S}$. However, in practical situations, covering the entire state space in even slightly large environments can be intractable.

To maximize the coverage of the state space by $P(s)$, we employ the following sampling method: Within the MEP population, there are a total of 20 opponent policies. For each opponent policy $\pi^{-1,k}$, we sample 1000 episodes. In these 1000 episodes, the opponents' policy are fixed to $\pi^{-1,k}$ while the self-agent traverses through all the opponent policies, resulting in the self-agent using per opponent policy for 50 episodes.

In Figs. 11 to 13, $\pi^{-1,k}, k = 1, 2, \ldots, 10$ denotes *seen* opponent policies, while $\pi^{-1,k}, k = 11, 12, \ldots, 20$ denotes *unseen* opponent policies. The lighter the color in the heatmap, the higher the KL divergence value, indicating a lower similarity between the two policies.

In the PP and OC environments, there is relatively large dissimilarity between all pairs of opponent policies. Assuming a dissimilarity threshold of $1.0$ (*i.e.*, two policies are dissimilar if their KL divergence is greater than $1.0$), the dissimilarity rates for PP and OC are $93.75\%$ and $91.5\%$, respectively. In contrast, the dissimilarity rate for LBF is $66.25\%$, indicating relatively smaller differences between opponent policies. This could be attributed to the fact that the state space of LBF is much smaller than PP and OC, making it difficult for well-trained opponent policies to exhibit significant behavioral diversity. Nonetheless, overall, we can consider the MEP opponent policy population we generated to be adequately diverse.

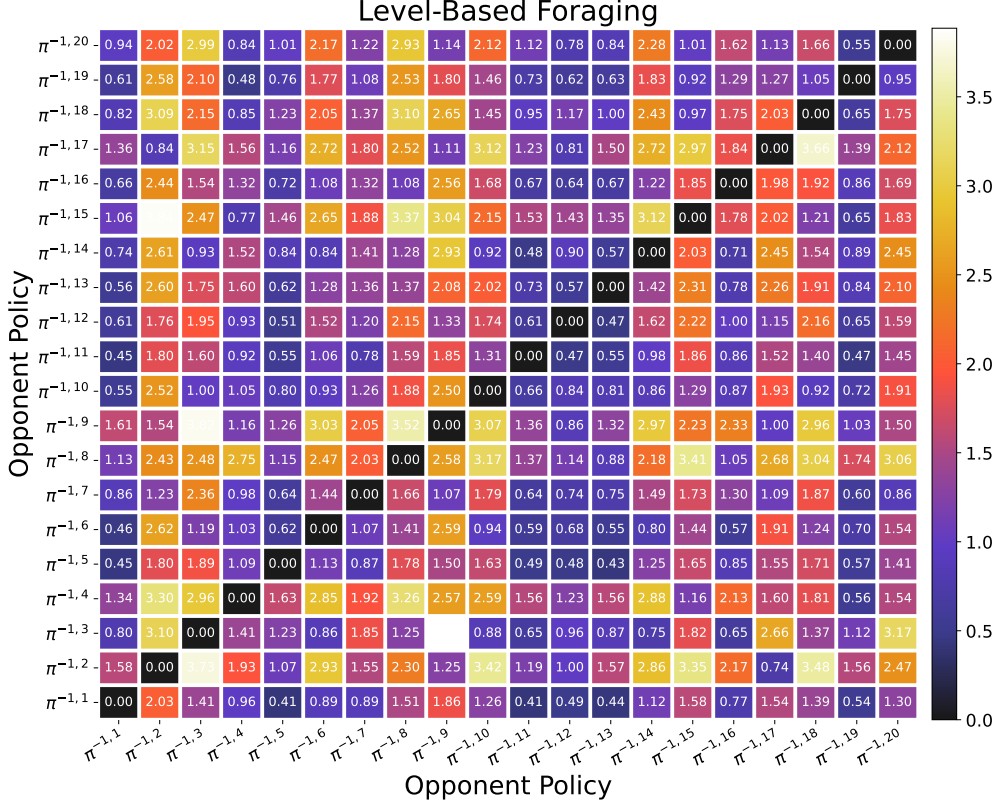

Figure 12: Pair-wise KL divergence of all policies within MEP population in LBF

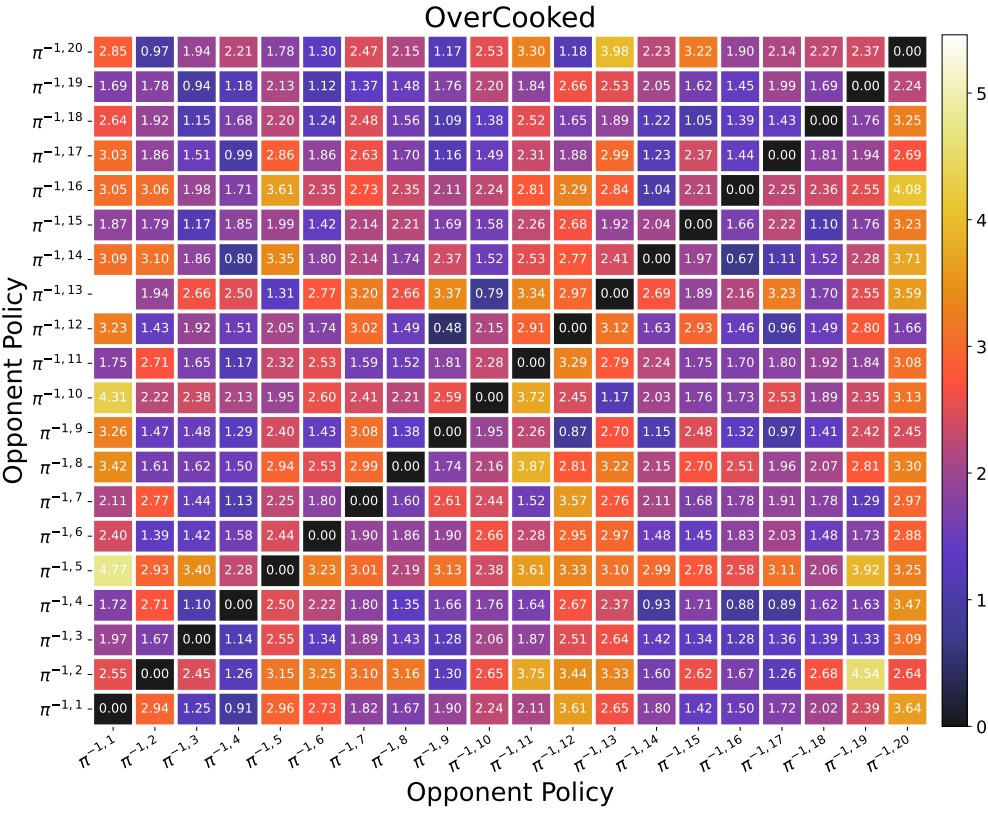

Figure 13: Pair-wise KL divergence of all policies within MEP population in OC

# H Hyperparameters

## H.1 Hyperparameters for Opponent Policies Training

As mentioned in Sec. 5.1, we employ a diversity-driven Population-Based Training algorithm MEP [104] to train a policy population, which is further used to create the opponent policy sets $\Pi^{\text{train}}$ and $\Pi^{\text{test}}$. For specific implementation of MEP, we adopt the open-source code of `maximum_entropy_population_based_training`, which is available at `https://github.com/ruizhaogit/maximum_entropy_population_based_training`. For the three environments, we use the same hyperparameters as this open-source code to train the MEP populations.

## H.2 Hyperparameters for In-Context-Learning-based Pretraining

| Hyperparameter Name | PP | LBF | OC |
|---|---|---|---|
| dimensionality of states | 16 | 21 | $(5, 4, 20)$ (image-like) |
| dimensionality of actions | 5 | 6 | 6 |
| horizon for each episode ($T$) | 100 | 50 | 400 |
| agent index of the self-agent | 3 | 0 | 0 |
| agent indexes of the opponents | $0, 1, 2$ | 1 | 1 |
| total number of episodes for training BRs | 50000 | 50000 | 50000 |
| discount factor for training BRs | 1.0 | 1.0 | 1.0 |
| batch size for training BRs | 4096 | 4096 | 4096 |
| number of updating epochs at each training step for training BRs | 10 | 10 | 10 |
| learning rate of the actor for training BRs | $5 \times 10^{-4}$ | $5 \times 10^{-4}$ | $5 \times 10^{-4}$ |
| learning rate of the critic for training BRs | $5 \times 10^{-4}$ | $5 \times 10^{-4}$ | $5 \times 10^{-4}$ |
| number of linear layers for training BRs (add 3 additional convolutional layers for OC) | 3 | 3 | $3 + 3$ |
| number of nodes of hidden layers for training BRs | 32 | 32 | 32 |
| clipping factor of PPO [73] for training BRs | 0.2 | 0.2 | 0.2 |
| maximum norm of the gradients for training BRs (clip if exceeded) | 5.0 | 5.0 | 5.0 |
| number of opponent policies in $\Pi^{\text{train}}$ ($K$) | 10 | 10 | 10 |
| sequence length of episode-wise in-context data $D^{\text{epi},k}$ ($H$) | 15 | 15 | 15 |
| number of trajectories randomly sampled to construct $D^{\text{epi},k}$ ($C$) | 3 | 3 | 3 |
| maximum sequence length for OMIS's GPT2 backbone ($B$) (see App. F for detailed descriptions) | 20 | 20 | 20 |
| reward scaling factor for pretraining $\pi_\theta, \mu_\phi, V_\omega$ (all the rewards are multiplied by $\frac{1}{\text{reward scaling factor}}$ to reduce the variance of training) | 100 | 1 | 100 |
| total number of training steps for pretraining $\pi_\theta, \mu_\phi, V_\omega$ | 4000 | 4000 | 4000 |
| discount factor for pretraining $\pi_\theta, \mu_\phi, V_\omega$ ($\gamma$) | 1.0 | 1.0 | 1.0 |
| batch size for pretraining $\pi_\theta, \mu_\phi, V_\omega$ | 64 | 64 | 64 |
| number of updating epochs at each training step for pretraining $\pi_\theta, \mu_\phi, V_\omega$ | 10 | 10 | 10 |
| weighting coefficient for pretraining $\pi_\theta$ | 1.0 | 1.0 | 1.0 |
| weighting coefficient for pretraining $\mu_\phi$ | 0.8 | 0.8 | 0.8 |
| weighting coefficient for pretraining $V_\omega$ | 0.5 | 0.5 | 0.5 |
| warm-up epochs for pretraining $\pi_\theta, \mu_\phi, V_\omega$ (the learning rate is multiplied by $\frac{\text{num\_epoch}+1}{\text{warm-up epochs}}$ to allow it to increase linearly during the initial warm-up epochs of training) | 10000 | 10000 | 10000 |
| learning rate for AdamW [51] optimizer for pretraining $\pi_\theta, \mu_\phi, V_\omega$ | $6 \times 10^{-4}$ | $6 \times 10^{-4}$ | $6 \times 10^{-4}$ |
| weight decay coefficient for AdamW optimizer for pretraining $\pi_\theta, \mu_\phi, V_\omega$ | $1 \times 10^{-4}$ | $1 \times 10^{-4}$ | $1 \times 10^{-4}$ |
| maximum norm of the gradients for pretraining $\pi_\theta, \mu_\phi, V_\omega$ | 0.5 | 0.5 | 0.5 |
| number of nodes of hidden layers for OMIS's GPT2 backbone (see App. F for detailed descriptions) | 32 | 32 | 32 |
| dropout factor for OMIS's GPT2 backbone | 0.1 | 0.1 | 0.1 |
| number of self-attention blocks for OMIS's GPT2 backbone | 3 | 3 | 3 |
| number of attention head for OMIS's GPT2 backbone | 1 | 1 | 1 |
| random seeds | $0, 1, 2, 3, 4$ | $0, 1, 2, 3, 4$ | $0, 1, 2, 3, 4$ |

### H.3   Hyperparameters for Decision-Time Search with In-Context Components

| Hyperparameter Name | PP | LBF | OC |
| --- | --- | --- | --- |
| total number of episodes for testing | 1200 | 1200 | 1200 |
| number of rollouts for self-agent's each legal action for DTS ($M$) | 3 | 3 | 3 |
| length of each rollout for DTS ($L$) | 3 | 3 | 3 |
| discount factor for DTS ($\gamma_{\text{search}}$) | 0.7 | 0.7 | 0.7 |
| threshold of mixing technique for DTS ($\epsilon$) | 10 | 0 | 0 |
| sequence length of episode-wise in-context data $D^{\text{epi}}$ ($H$) | 15 | 15 | 15 |
| number of the most recent trajectories used to construct $D^{\text{epi}}$ ($C$) | 3 | 3 | 3 |
| maximum sequence length for OMIS's GPT2 backbone ($B$) | 20 | 20 | 20 |
| number of nodes of hidden layers for OMIS's GPT2 backbone | 32 | 32 | 32 |
| dropout factor for OMIS's GPT2 backbone | 0.1 | 0.1 | 0.1 |
| number of self-attention blocks for OMIS's GPT2 backbone | 3 | 3 | 3 |
| number of attention head for OMIS's GPT2 backbone | 1 | 1 | 1 |
| random seeds | $0, 1, 2, 3, 4$ | $0, 1, 2, 3, 4$ | $0, 1, 2, 3, 4$ |

## I   Quantitative Analysis of Attention Weights Learned by OMIS

To rigorously evaluate whether OMIS can effectively characterize opponent policies, we conduct a quantitative analysis of the attention weights learned by OMIS by calculating the pair-wise *Pearson Correlation Coefficients* (PCC) between the attention vectors. The relevant results are shown in Fig. 14. The first column is the heatmaps of the pair-wise PCC statistics of all attention vectors, and the second column shows the corresponding $p$-value plots for the statistics in the first column, with pairs marked in white for $p < 0.05$ and black otherwise.

The observations reveal that the attention vectors of the same opponent policy have *strong pair-wise correlations* (*i.e.*, statistics close to $1$ and $p < 0.05$) across multiple timesteps. In contrast, the attention vectors of different opponent policies generally have no strong pair-wise correlations with each other. Although there is some pair-wise correlation between the attention vectors of different opponent policies, each opponent policy generally has the strongest pair-wise correlation with its own other attention vectors. These observations indicate that the attention weights learned by OMIS can be distinguished by different opponent policies and maintain consistency for the same opponent policy to some extent. Therefore, this analysis further demonstrates OMIS's ability to represent opponent policies based on in-context data.

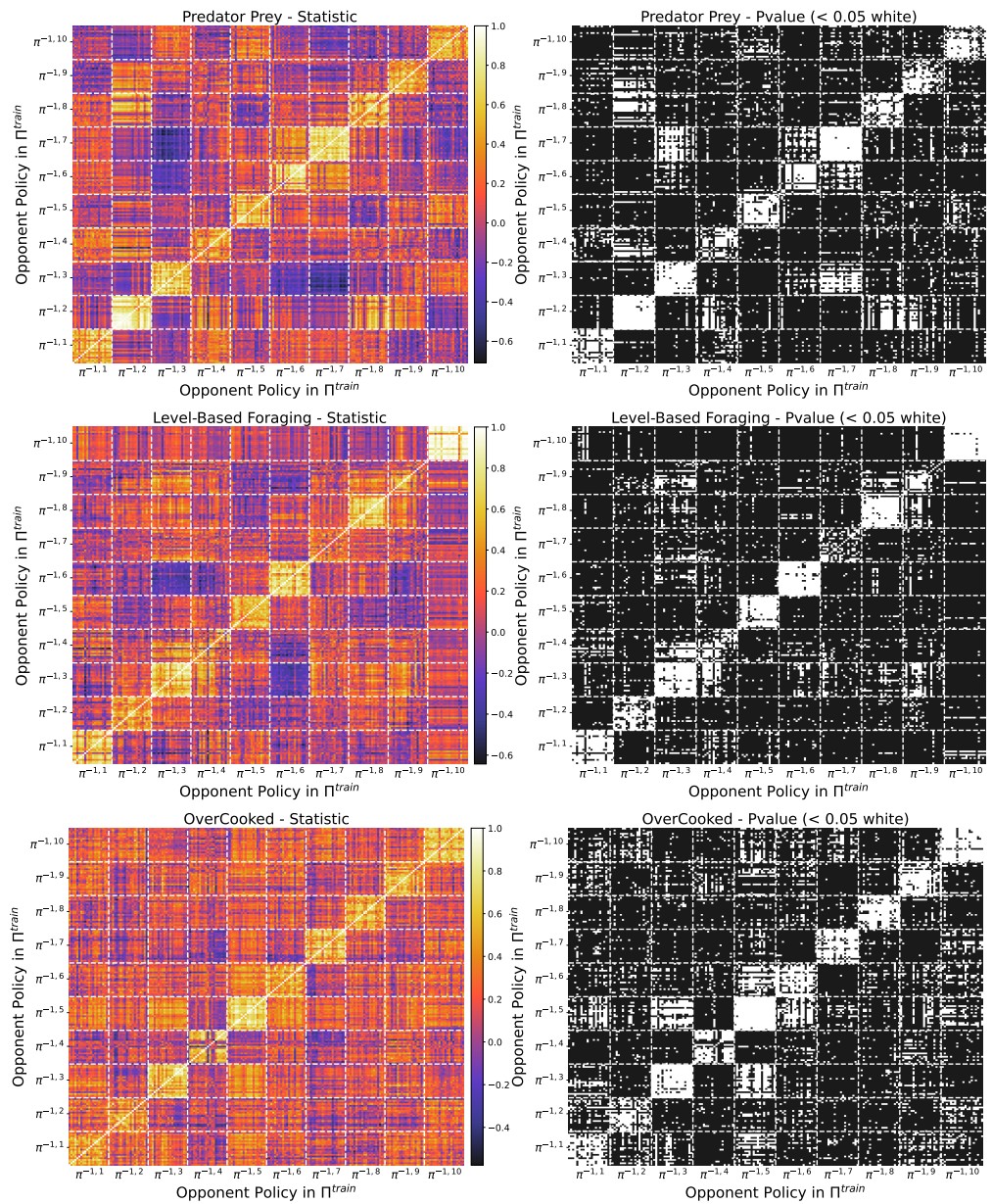

Figure 14: Pair-wise PCC statistics and $p$-values between the attention weights learned by OMIS. The attention vectors on $D^{\mathrm{epi},k}$ are calculated over the final 20 timesteps against each opponent policy.

## J    In-depth Discussion

In Sec. 6 of the main text, we analyzed this study's limitations and future work from four perspectives. Herein, we would like to point out that there are currently many potential feasible solutions for each aspect. The OMIS proposed in this paper can be viewed as a complete framework that tackled the main problems in existing OM works during the pretraining and testing stages. This framework can be modified for other settings (such as the opponents are learning, imperfect information, etc.). Moreover, this framework also represents a minimalist approach, focusing on generic opponent modeling settings, while more complex settings can be considered as new research problems to explore in the future.

(1) For the settings where opponents are learning, according to the observations in Laskin et al. [43], ICL has the ability to model a sequence taken during the learning process. Therefore, we can potentially model continuously updating opponents by using the complete $(s, a^{-1})$ sequences during opponent learning as in-context data. Strictly speaking, regardless of the type of opponent, as long as we have their in-context data and their best response policy, we can use the OMIS framework to learn to respond to that opponent. Another possible solution is to leverage the idea of Opponent Modeling with Shaping Opponents' Learning [22, 23, 47, 41, 53, 92, 105, 25] (see App. A), explicitly modeling the opponent's gradient updates during testing to shape their learning process.

(2) For imperfect information settings, there is a vast of research in the field of imperfect information online search [58, 12, 13, 83, 35, 38, 50], with many mature methods that can be adapted to work within the OMIS framework. Yet, this adaptation is non-trivial, as such DTS methods often require explicit or learned beliefs about the true state, introducing significant additional computational complexity. Interestingly, a recently proposed Update-Equivalence Framework [77] suggests that we can effectively search in imperfect information settings without relying on beliefs.

(3) For more complex decision-time searches, numerous advanced DTS methods [75, 76, 12, 13, 9, 46, 38, 36, 5, 100, 18, 60] can seamlessly integrate with our framework. This is because the OMIS pretraining stage learns all the key components needed for DTS: an actor, a critic, and an opponent imitator. The actor provides a good prior decision for the self-agent during the DTS, the critic estimates the value of a given terminal state during the DTS, and the opponent imitator estimates the most probable action for the opponent during the DTS.

## K    Compute Resources

- CPU: AMD EPYC 7742 64-Core Processor $\times 2$
- GPU: NVIDIA GeForce RTX 3090 24G $\times 8$
- MEM: 500G
- Maximum total computing time: pretraining + testing $\approx 40h$

## L    Broader Impacts

This paper presents work that aims to advance the field of Machine Learning. There are many potential societal consequences of our work, none of which we feel must be specifically highlighted here.

