# OpenReview forum: "Opponent Modeling with In-context Search"
_NeurIPS.cc/2024/Conference — NeurIPS 2024 poster_

### Official Review · Reviewer_YkYd · 2024-06-17

**Soundness:** 3
**Presentation:** 4
**Contribution:** 2
**Rating:** 4
**Confidence:** 4

**Summary:**

This paper introduces an approach to opponent modeling in multi-agent environments, aimming to address the challenges of generalization and performance instability when trained agents interacting with unknown opponents during the test time. The proposed method, Opponent Modeling with In-context Search (OMIS), leverages in-context learning-based pretraining to train a Transformer model, which includes three components: an actor learning best responses to opponent policies, an opponent imitator mimicking opponent actions, and a critic estimating state values. OMIS uses these pretrained components for decision-time search to refine the actor’s policy during testing. The paper theoretically proves that OMIS converges in opponent policy recognition and has good generalization properties under certain conditions, while the decision-time search improves performance stability without gradient updates. Empirical results in competitive, cooperative, and mixed environments show that OMIS effectively and stably adapts to opponents with unknown non-stationary policies.

**Strengths:**

1. Focus on Generalization to Unseen Opponents:
   The paper focuses on a critical challenge in multi-agent reinforcement learning: the ability to generalize to unseen opponent policies during testing. The proposed OMIS approach leverages in-context learning and decision-time search to enhance the agent's adaptability and robustness against such opponents.

2. Clear and Coherent Writing:
   The paper is well written with logical structure and good prsentation apporaches, which makes it easy to follow.

3. Comprehensive Literature Review:
   The authors have conducted a comprehensive literature review, covering various approaches to opponent modeling, including representation learning, Bayesian learning, meta-learning, and decision-time search.

4. Thorough Experimental Evaluation:
   The empirical evaluation of OMIS is thorough and robust, encompassing a variety of environments (competitive, cooperative, and mixed) and comparing against several baseline methods.

**Weaknesses:**

1. Lack of novelty: I assume that two of the most important feature of this work is in-context learning-based pretraining and in-context search. However, I hardly find these two ideas or the combination of these two ideas applied to marl novel.

2. Strong assumptions: OMIS relies on the access to the environment dynamics which is a strong assumption in real problems and it may give an unfair advatange to OMIS when compared to other baselines. In addition, I understand that the authors have conducted experiments to test OMIS's robustness to random E (which is the number of episodes before the opponents switching to another policy), but it is still unclear to me that whether OMIS requires to know the exact time points when opponent switch policies.

3. Scalability: The experiments conducted in the paper are relatively simple and conceptual compared to the complexity of the OMIS, I am concerned with how OMIS can perform in more complex environments with regrad to associated cost of generating diverse training opponent policies population, collecting training trajectories, pretraining and computing resources for search.

**Questions:**

1. Could you clarify if OMIS needs to know the exact time points when opponents switch policies during the test stage?

2. Is it possible to provide OMIS-dyna under all ratio settings?

3. For OMIS w/o S, could you analyze the reasons why it can still outperform other baselines in LBF and OC but falls behind in competitive PP when the ratio of unseen opponents is high?

4. Could you analyze the reasons why OMIS can outperform other baselines while the in-context components of OMIS do not estimate very accurately in PP and LBF?

5. During the search stage, these levels of estimation errors can quickly accumulate into large compounding errors. Therefore, I wonder how the results of deep search can benefit from the pre-training of the in-context components of OMIS, especially the opponent imitator.

**Limitations:**

Please refer to the Questions part.

---

> ### Author Rebuttal · Authors · 2024-08-05
>
> # Response to Reviewer **YkYd**
>
> Thank you very much for your recognition of our paper's problem setting, paper writing, experimental results, and the valuable feedback you provided. In response to your comments, we would like to make the following clarifications and feedback. We hope our explanations and analyses can eliminate concerns and make you find our work stronger.
>
> > **W1**. Lack of novelty: … However, I hardly find these two ideas or the combination of these two ideas applied to marl novel.
> >
>
> We all know that *learning* and *search* are two major “magic keys” in machine learning, as well as common paradigms in MARL. We argue that the novelty of OMIS lies in using **in-context** **learning (ICL)** and **in-context** **search (ICS)** to address the challenges of OM, with "**in-context**" being the core aspect. In other words, we offer a new perspective that learning and search should be in-context (adaptive). We utilize ICL-based pretraining to respond to opponents based on **in-context data (ICD)** adaptively. Building upon this foundation, we further improve this adaptive capability through ICS. Our ICS is capable of inferring opponent actions, making responses, and estimating corresponding values based on the opponent's ICD. Lastly, both our proposed ICL and ICS are theoretically well-grounded, whereas many existing OM approaches lack theoretical analysis. To end with, we would like to argue that novelty is not always about creating entirely new methodologies. *Offering a systematic approach that effectively addresses a long-standing problem in a field is also a form of innovation.*
>
> > **W2**. Strong assumptions: OMIS relies on the access to the environment dynamics … it may give an unfair advatange to OMIS when compared to other baselines.
> >
>
> We include results and corresponding analyses for the version of OMIS where the dynamics are learned. Please refer to the **Global Response**.
>
> > (1) **W2**. In addition, …
> (2) **Q1**. Could you clarify if OMIS needs to know the exact time points when opponents switch policies during the test stage?
> >
>
> OMIS does not require knowing the exact time points when opponents switch policies, as demonstrated in Figure 7, where the timing of policy switches is strictly unknowable to OMIS. Even in this situation, OMIS works well and generally outperforms other baselines (see Figures 3 and 4).
>
> If the switch times were precisely known, it might be able to further enhance OMIS. Detecting whether opponents have switched policies is an independent research problem [1,2]. In fact, we could leverage these methods for detecting opponent switches to make $D^{\text{epi}}$ as pure as possible, thereby achieving better result.
>
> [1] Efficiently detecting switches against non-stationary opponents. AAMAS, 2017.
>
> [2] A deep bayesian policy reuse approach against non-stationary agents. NIPS, 2018.
>
> > **W3**. Scalability: The experiments conducted in the paper are relatively simple, …
> >
>
> As you mentioned, the experiments in our paper are generally not that complex. Despite this, *current algorithms in the OM domain still struggle to work well*, facing challenges such as having difficulty generalizing to unknown opponent policies and performing unstably (see Figures 3 and 4). In contrast, our approach performs well in these commonly used benchmarks. Furthermore, we would like to emphasize that *OverCooked (OC) is a relatively complex environment* with a high-dimensional state space where all states are image-like. Training even a workable policy on OC is actually challenging and requires more than a few computational resources.
>
> > **Q2**. Is it possible to provide OMIS-dyna under all ratio settings?
> >
>
> We provide the results for OMIS-dyna under all ratio settings. Please refer to the **Global Response**.
>
> > **Q3**. For OMIS w/o S, … why it can still outperform other baselines in LBF and OC but falls behind in competitive PP when the ratio of unseen opponents is high?
> >
>
> The possible reason for this result is that PP is a continuous environment with an uncountable state space, exacerbating the degree of out-of-distribution of unseen opponents' states. OMIS w/o S relies on $(s, a^{-1})$ tuples for generalization, whereas other baselines do not depend on this for generalization. This reliance introduces additional challenges for OMIS w/o S. In contrast, the state spaces of LBF and OC environments are countable, and many of the states of unseen opponents can be familiar (though the actions in those states may be novel). This makes generalization for OMIS w/o S easier.
>
> > (1) **Q4**. … why OMIS can outperform other baselines while the in-context components of OMIS do not estimate very accurately in PP and LBF?
> (2) **Q5**. … , I wonder how the results of deep search can benefit from the pre-training of the in-context components of OMIS, especially the opponent imitator.
> >
>
> We argue that the necessity of search arises precisely because there are estimation errors in the opponent's actions and value predictions. Otherwise, we could directly solve for the best response to the current opponent's policy, and there would be no need for search.
>
> The search domain generally does not emphasize the concept of compound errors; rather, it focuses on the trade-off between exploration and exploitation. In fact, the inaccuracy of the opponent imitator can be viewed as accurately sampling the opponent's actions most of the time while exploring other actions occasionally, meeting the trade-off need for search. Given that OMIS's critic is relatively accurate, combined with the reward signal during search, we can consider our value estimates to have relatively high confidence.
>
> All your questions and feedback have greatly contributed to improving our manuscript. We welcome further comments from you and will seriously consider your suggestions for revisions. If you feel that we have addressed your concerns, we hope you will reconsider your rating.

---

> > ### Comment · Reviewer_YkYd · 2024-08-13
> >
> > Thank the authros for their detailed response. I appreciate the efforts for conducting these many experiments requested by me and other reviewers. Specifically, I am glad to see the new results of 1) learning dynamics 2) a search-based baseline 3)OMIS dyna under all ratios.
> >
> > However, two of my concerns still persists and I tend to keep my current ratings.
> >
> > 1) I still not find the OMIS novel. I understand in-context learning / in-context search quite popular recently, but application of it on a common MARL framework (i.e. learning + search) seems not novel to me. I appreciate the theoretical analysis but that does not necessarily lead to novelty. It seems we might not be able to achieve an agreement on this point. Therefore I will leave this to AC to decide based on all reviewes reviews.
> >
> > 2) I still have concern about the approach's scalability. OMIS requests much pre-trained learning and data prepration offline and search/inference computing resources online. With this scale of training resources, "experiments conducted in the paper are relatively simple and conceptual ". Consider OMIS setting, I don't think that "OverCooked (OC) is a relatively complex environment with a high-dimensional state space where all states are image-like" adresses my concern. OC is fully cooperative. In addition, high-dimensional state space does not necessarily mean complex games with complex adversarial strategies to learn.
> >
> > Lastly, I want to explain why I care about the approach's novelty and scalability. As I mentioned in my first review, I appreciate the authors "focuses on a critical challenge in multi-agent reinforcement learning". However, if OMIS request these many computation resources for achieving SOTA on "relatively simple and conceptual" games (offline: pretraining on there mouldes each is a GPT2 decoder composed of 3 self-attention blocks, building opponent policies pool for training, corresponding training data preparation, learning environment dynamics and online search + inference), how will OMIS contribute to the community? Does it provide a new perspecitve on how to solve the challenge or scalable way to sovle complex problems?

---

> > > ### Author Response · Authors · 2024-08-13
> > > **Response to your new comments (1/3)**
> > >
> > > Dear Reviewer,
> > >
> > > We are pleased that our rebuttal has addressed most of your concerns. Regarding the remaining two concerns about novelty and scalability, we agree that these are indeed very important issues. At the same time, they provide us with an opportunity to further clarify the advantages of our approach in terms of both novelty and scalability. **We hope that our discussion will also help the other reviewers and the AC gain a deeper understanding of our work.**
> > >
> > > ---
> > >
> > > > I still not find the OMIS novel. I understand in-context learning / in-context search quite popular recently, but application of it on a common MARL framework (i.e. learning + search) seems not novel to me.
> > > >
> > >
> > > In-context learning is indeed a popular concept right now, but **we have not chosen to use it simply because of its popularity**. On the contrary, *our method is built upon a deep understanding of the opponent modeling problem.* Although various approaches already exist in the field of opponent modeling, such as those based on representation learning [1,2,3], Bayesian learning [4,5], meta-learning [6,7], shaping opponents' learning [8,9], and recursive reasoning [10,11]. However, we argue that ***opponent modeling is fundamentally a sequence-to-sequence problem***.  Specifically, **the input sequence consists of the historical data generated from interactions with the opponent, which we refer to as in-context data, while the output sequence represents the optimal sequence of actions that the self-agent needs to take.** From this new, simplified perspective that more closely aligns with the essence of the problem, we adopted the Transformer model as the foundational architecture for our approach. The Transformer is currently one of the most effective sequence-to-sequence models and inherently possesses in-context learning capabilities, which, *to our knowledge, has never been utilized in the opponent modeling literature before*. In terms of **in-context learning**, existing work typically analyzes its capabilities in the context of language modeling. However, we have rigorously demonstrated, through theoretical proof, the unique properties of in-context learning in the domain of decision-making, particularly in opponent modeling, i.e., *our approach possesses the following properties:  when the opponent's policy is a seen one, OMIS w/o S can accurately recognize the opponent's policy and converge to the best response against it; when the opponent's policy is an unseen one, OMIS w/o S recognizes the opponent policy as the seen opponent policy with the smallest KL divergence from this unseen opponent policy and produces the best response to the recognized opponent policy.* This makes our approach one of the very few in the opponent modeling domain that comes with performance guarantees in terms of generalization.
> > >
> > > In terms of **in-context search**, to the best of our knowledge, **our work is the first to propose this concept**. The fundamental difference between our search method and existing ones is that ours is an **adaptive search**. Existing search methods typically assume that the opponent follows a fixed policy, such as a scripted or RL-trained policy [12,13,14,15] ([12] is the new search-based baseline we added during the rebuttal period at the request of the reviewers). These fixed opponent policies often have a significant gap compared to ground-truth opponent policies. In contrast, *our in-context search method can predict the opponents' actions based on interaction data, enabling a targeted and adaptive search*. **This search approach is not only novel in the OM domain but also introduces a new methodological paradigm in the RL field.** Additionally, *we have rigorously proven that OMIS's search is guaranteed to improve upon OMIS w/o S without requiring any gradient updates.*
> > >
> > > In summary, our paper offers **a completely new perspective on opponent modeling** and introduces **an entirely new algorithmic framework** with **novel theoretical properties**. We hope that this algorithm, which incorporates in-context learning and in-context search capabilities along with strong theoretical guarantees, addresses your concerns about the novelty of our work.
> > >
> > > [1] Learning policy representations in multiagent system, ICML 2018.
> > >
> > > [2] Deep interactive bayesian reinforcement learning via meta-learning, AAMAS 2021.
> > >
> > > [3] Agent modelling under partial observability for deep reinforcement learning, NIPS 2021.
> > >
> > > [4] A deep bayesian policy reuse approach against non-stationary agents, NIPS 2018.
> > >
> > > [5] Greedy when sure and conservative when uncertain about the opponents, ICML 2022.
> > >
> > > [6] Continuous adaptation via meta-learning in nonstationary and competitive environments, ICLR 2018.
> > >
> > > [7] A policy gradient algorithm for learning to learn in multiagent reinforcement learning, ICML 2021.
> > >
> > > [8] Learning with opponent-learning awareness, AAMAS 2018.

---

> > > ### Author Response · Authors · 2024-08-13
> > > **Response to your new comments (2/3)**
> > >
> > > [9] Stable opponent shaping in differentiable games, ICLR 2018.
> > >
> > > [10] Probabilistic recursive reasoning for multi-agent reinforcement learning, ICLR 2019.
> > >
> > > [11] Model-based opponent modeling, NIPS 2022.
> > >
> > > [12] Know your Enemy: Investigating Monte-Carlo Tree Search with Opponent Models in Pommerman, ALA at AAMAS 2023.
> > >
> > > [13] Mastering the game of go with deep neural networks and tree search. Nature, 2016.
> > >
> > > [14] Mastering the game of Go without human knowledge. Nature, 2017.
> > >
> > > [15] A general reinforcement learning algorithm that masters chess, shogi, and go through self-play. Science, 2018.
> > >
> > > > I still have concern about the approach's scalability. OMIS requests much pre-trained learning and data prepration offline and search/inference computing resources online. With this scale of training resources, "experiments conducted in the paper are relatively simple and conceptual "… Does it provide a scalable way to sovle complex problems?
> > > >
> > >
> > > Regarding **scalability**, we strongly argue that **our approach was designed with scalability as a key consideration from the outset**. Our OMIS algorithm, which incorporates in-context learning and in-context search capabilities, perfectly aligns with the principles outlined in **Richard S. Sutton**'s "***The Bitter Lesson***" [16]:
> > >
> > > ### ***One thing that should be learned from the bitter lesson is the great power of general purpose methods, of methods that continue to scale with increased computation even as the available computation becomes very great. The two methods that seem to scale arbitrarily in this way are search and learning.***
> > >
> > > This stands in stark contrast to existing opponent modeling approaches, such as those based on Bayesian learning [4,5], meta-learning [6,7], shaping opponents' learning [8,9], and recursive reasoning [10,11]. These approaches typically involve *more complex and cumbersome methodologies*, making them inherently much harder to scale. In contrast, our approach only requires **learning** a Transformer model and then performing a **search** based on it, making it ***essentially scalable***.
> > >
> > > Regarding your concern about **the complexity of the experimental environments**, we agree that they may not be very complex. However, the benchmarks used in this paper are recognized as some of the *most representative in the opponent modeling domain* [1-11]. While some of these environments may seem "simple and conceptual," *they often present significant challenges even for current SOTA opponent modeling approaches* such as MBOM [11], which sometimes perform poorly in them. Our approach, which is both simple and scalable, has been able to effectively outperform most existing approaches. To the best of our knowledge, there aren't more complex environments currently used in the opponent modeling domain. If you have suggestions for better benchmarking environments, we would be more than happy to evaluate our approach to them.
> > >
> > > [16] http://www.incompleteideas.net/IncIdeas/BitterLesson.html?ref=blog.heim.xyz, Rich Sutton.

---

> ### Author Response · Authors · 2024-08-11
> **Requesting feedback and any remaining concerns**
>
> Dear Reviewer,
>
> We sincerely appreciate your thorough reading and the valuable feedback you provided during the review process. Your constructive comments have helped us strengthen our manuscript, particularly in clarifying our work's novelty and validating our approach's effectiveness when the environment model is unavailable.
>
> We are eager to have the opportunity to have a further discussion with you and will do our best to address any remaining concerns or questions you may have. We sincerely request your feedback so that we can conduct additional experiments or make revisions to further improve the current version of our submission promptly.
>
> ---
>
> Here is a summary of our rebuttal, which we hope adequately addresses all the concerns you raised:
>
> 1. We clarified that **the novelty of OMIS** lies in using **in-context learning (ICL)** and **in-context search (ICS)** to address the challenges of opponent modeling (OM), with “**in-context**” being the core aspect, to address your concerns about “**our contributions on novelty**”. Additionally, we presented our argument that novelty is not always about creating entirely new methodologies. *Offering a systematic approach that effectively addresses a long-standing problem in a field is also a form of innovation.*
> 2. We have supplemented our rebuttal with **experimental results for OEOM under learned dynamic transitions** to address your concern regarding “**OEOM's strong dependence on the environment model**”, as detailed in the **Global Response**. These results support the effectiveness of OEOM under such conditions, and we will include them in the experimental section of the revision.
> 3. We have provided the “**results for OMIS-dyna under all ratio settings”** as you requested, as detailed in the **Global Response**.
> 4. To address your concern about “**why OMIS w/o S was unable to outperform other baselines in the PP environment**”, we provided a detailed explanation based on **the characteristics of the environment** and **the principles underlying the OMIS w/o S algorithm**.
> 5. Regarding your concern about “**how search can improve OMIS w/o S even when the opponent imitator's estimates are inaccurate**”, we clarified that **the necessity of search arises because there are estimation errors** in the opponent's actions and value predictions. Additionally, we explained that *OMIS's inaccurate estimates can be seen as equivalent to a search mechanism that balances exploration and exploitation*, which is needed in the search domain to refine the original policy.

---

> ### Author Response · Authors · 2024-08-13
> **Response to your new comments (3/3)**
>
> > However, if OMIS request these many computation resources for achieving SOTA on "relatively simple and conceptual" games (offline: pretraining on there mouldes each is a GPT2 decoder composed of 3 self-attention blocks, building opponent policies pool for training, corresponding training data preparation, learning environment dynamics and online search + inference), how will OMIS contribute to the community?
> >
>
> Regarding **computation resources**, we would like to clarify that **we did not train three separate Transformer networks** offline. Instead, **we trained a single Transformer backbone and used three output heads for the actor, critic, and opponent imitator, all of which share this Transformer backbone**. In fact, the Transformer model we used is very **lightweight**, with a memory footprint of only **≤3MB**, so it requires minimal computational resources.
>
> Regarding the **complexity of the algorithmic process**, during the pretraining stage, our approach only requires straightforward supervised training using data generated by any RL algorithm. During the testing stage, the pretrained Transformer can either be directly used (which already outperforms most baselines in our experiments, see Fig. 3 and Fig. 4 in the main text) or further enhanced with our very simple search method. In contrast, other approaches involve a lot of complex procedures. Take MBOM [11] as an example. In our implementation, MBOM also uses a Transformer backbone of the same scale. However, its overall process includes several complex steps: building an opponent policies pool for training, generating training data using opponent policies and conducting pretraining, learning environment dynamics (if necessary), online learning and finetuning multiple nested opponent models, conducting online planning and inference, and finally, using explicit Bayesian methods to mix the nested opponent models. Another example is Meta-MAPG [7], where our implementation also uses a Transformer backbone of the same scale. Its overall process is as follows: building an opponent policies pool for training, generating training data using opponent policies, and conducting pretraining with meta-gradient methods (which involves both outer and inner loops of meta-learning). The online stage requires continuous inference and finetuning sample collection from the opponent, followed by gradient updates to the learned model for continuous finetuning (which is highly sensitive to hyperparameters, making it challenging to work effectively in practice).
>
> ---
>
> We hope our detailed response has addressed the remaining concerns you had regarding novelty and scalability. We believe that **OMIS offers a simple, scalable, and effective solution for the opponent modeling community**. Even with limited time remaining, we are more than willing to engage in further discussions if you have any remaining concerns. If we have resolved your concerns, we sincerely hope you will reconsider your score.
>
> Best regards,
>
> All authors

---

### Official Review · Reviewer_eKwk · 2024-07-08

**Soundness:** 3
**Presentation:** 3
**Contribution:** 3
**Rating:** 7
**Confidence:** 2

**Summary:**

This paper proposes a novel approach to opponent modeling called OMIS, which combines ICL and decision-time search to improve performance and stability in three distinct game settings over baselines. It also shows ablations that validate the need for specific components such as mixing technique, search, episode-wise in-context data for episodes, and step-wise in-context data.

**Strengths:**

- Well written paper with great attention to detail
- Novel combination of known components (ICL + In-context search)
- Strong empirical results and ablations
- Extensive comparison to known baselines
- Extensive Appendix with proofs, codebase, and website
- Question and Answer format helps to pre-empt some questions.

**Weaknesses:**

- The theoretical explanations in Section 3 are dense. It’s unclear if this section provides additional benefit within the Methodology section. You may consider moving Section 4.3 to the beginning of the Methodology section, so that the transition between the methods and experiment are clearer. Alternatively, you could summarize the key steps in a summary paragraph at the beginning of the Experimental setup to explain why you have chosen these Environments and Baselines.

**Questions:**

- Why do you specifically select 10 policies from the MEP population to form the training policies?
- Why do the test policies need to contain seen policies? When testing, why not have all unseen policies?

**Limitations:**

Yes.

---

> ### Author Rebuttal · Authors · 2024-08-05
>
> # Response to Reviewer **eKwk**
>
> Thank you very much for your recognition of our paper's methodologies, theories, empirical results, paper writing, and the valuable feedback you provided. In response to your comments, we would like to make the following clarifications and feedback. We hope our explanations and analyses can eliminate concerns and make you find our work stronger.
>
> > The theoretical explanations in Section 3 are dense. It’s unclear if this section provides additional benefit within the Methodology section. You may consider moving Section 4.3 to the beginning of the Methodology section …
> >
>
> Thank you very much for your valuable suggestions. The connection between our methodology and experiments is indeed not smooth. In the revision, we will move Section 4.3 to the beginning of the Methodology section to make the transition between the methods and experiments clearer.
>
> > Why do you specifically select 10 policies from the MEP population to form the training policies?
> >
>
> Selecting 10 policies from the MEP population is just one of the possible reasonable design choices. We follow this choice for the following reasons:
>
> (1) MEP is an efficient approach for generating a diverse and high-strength population of policies, making it a reasonable method for opponent policy generation.
>
> (2) For all opponent modeling approaches, we use the same 10 policies from the MEP population as training opponent policies, facilitating fair comparison with other baselines.
>
> (3) The specific selection of which 10 policies from the MEP population does not matter, as all policies have appropriate differences between each other, as can be observed from the quantitative analysis of policy diversity in Appendix G.
>
> (4) Considering computational resource constraints, we select only 10 policies as training opponent policies.
>
> > Why do the test policies need to contain seen policies? When testing, why not have all unseen policies?
> >
>
> Although testing against seen opponents may seem trivial, existing opponent modeling approaches still struggle to handle even seen opponents, as can be observed in Figures 3 and 4. This is because their methodological designs often rely heavily on intuition but lack rigorous theoretical analysis. In contrast, OMIS provides good theoretical properties in terms of generalization (see Theorems 4.2 and 4.3), ensuring accurate recognition of seen opponent policies during pretraining and providing the most appropriate responses. Our experimental results (Figures 3 and 4) further validate the effectiveness of our approach, as it generally outperforms other baselines against seen opponents.
>
> All your questions and feedback have greatly contributed to improving our manuscript. With the valuable input from you and all other reviewers, the quality of our work can be significantly enhanced. We welcome further comments from you and will seriously consider your suggestions for revisions. If you feel that we have addressed your concerns, we hope you will reconsider your rating.

---

> > ### Comment · Reviewer_eKwk · 2024-08-07
> >
> > Thank you for your response to my comments. I have read your rebuttal and will provide further comments soon, as needed.

---

> > > ### Author Response · Authors · 2024-08-11
> > >
> > > Dear Reviewer,
> > >
> > > Thank you very much for your diligent review and the highly valuable feedback, which has greatly contributed to improving our manuscript. If you have any remaining concerns or questions about our paper, we warmly welcome further comments and are eager to engage in a constructive discussion with you.

---

> > > > ### Comment · Reviewer_eKwk · 2024-08-11
> > > >
> > > > Thank you again for addressing my questions. They have been largely answered. I will maintain my score.

---

### Official Review · Reviewer_G1Ke · 2024-07-10

**Soundness:** 3
**Presentation:** 3
**Contribution:** 2
**Rating:** 6
**Confidence:** 4

**Summary:**

This paper addresses the problem of opponent modeling and leverages in-context learning to tackle the challenges posed by opponents using non-stationary and unknown policies. Specifically, the proposed method, OMIS, employs PPO to train a best-response policy for each opponent policy in the training set. OMIS then uses these best-response policies to generate in-context data. Based on this data, OMIS trains three in-context components using transformers: the self-agent actor, the opponent imitator, and the critic. With these components, OMIS can perform multiple rollouts for each action and select the action with the highest expected return to interact with opponents. By relying on in-context data, OMIS effectively mitigates the adverse effects of non-stationary and unknown opponent policies.

**Strengths:**

1. The proposed method is meticulously designed and clearly articulated, ensuring the reliability and validity of the results.

2. The paper provides a comprehensive review of existing literature, showcasing a deep understanding of the relevant background.

3. The paper is well-written, logically organized, and structured, making it straightforward for readers to comprehend and follow the arguments and conclusions.

**Weaknesses:**

1. This work implicitly assumes the presence of a perfect transition model. OMIS relies on this transition model to perform multiple rollouts and identify the best action. However, in many application domains, such transition models do not exist, making OMIS inapplicable.

2. The mixing technique in Equation 10 is quite simple and may not generalize across different domains. Additionally, the hyperparameter requires fine-tuning for different environments.

**Questions:**

1. OMIS requires virtual transition dynamics, which are exact replicas of the true dynamics, to perform rollouts. However, if the virtual transition dynamics need to be learned from data, as in model-based RL approaches, how will OMIS perform under these conditions?

2. Lemma 4.1 assumes that "the sampling of $s$ from $\tau_{pre}^{-1}$ is independent of the opponent's policy." However, $\tau_{pre}^{-1}$ is the probability distribution over all trajectories involving the opponent's policy during pre-training. Thus, the opponent's policy should influence the distribution $\tau_{pre}^{-1}$, and sampling from $\tau_{pre}^{-1}$ should be correlated with the opponent's policy. So is the assumption reasonable?

3. Equation 10 considers the expected return of the action selected by the search as the confidence of the search policy. How does this design address the issue of value function overestimation?

**Limitations:**

The authors have discussed the limitations and potential negative societal impact of their work.

---

> ### Author Rebuttal · Authors · 2024-08-05
>
> # Response to Reviewer **G1Ke**
>
> Thank you very much for your recognition of our paper's writing, methodologies, literature reviews, and the valuable feedback you provided. In response to your comments, we would like to make the following clarifications and feedback. We hope our explanations and analyses can eliminate concerns and make you find our work stronger.
>
> > (1) This work implicitly assumes the presence of a perfect transition model. OMIS relies on this transition model …, making OMIS inapplicable.
> (2) OMIS requires virtual transition dynamics ... However, if the virtual transition dynamics need to be learned from data, …, how will OMIS perform under these conditions?
> >
>
> We include results and corresponding analyses for the version of OMIS where the dynamics are learned. Please refer to the **Global Response**.
>
> > The mixing technique in Equation 10 is quite simple and …, the hyperparameter requires fine-tuning for different environments.
> >
>
> Thank you for pointing that out. In fact, more complex hybrid techniques combining both search and original policies have been proposed by others as well, such as [1]. However, their technique also introduces hyperparameters, and our experimental verification found that the approach proposed by [1] yielded comparatively poorer results. In contrast, our proposed mixing technique is a simple yet effective approach. Additionally, we empirically found that it is quite straightforward to find suitable values for the hyperparameter $\epsilon$. The value of $\epsilon$ only needs to follow one principle: *it should be slightly smaller than the absolute value of the sparse rewards in the environment.* We argue that our technique has an advantage building upon its simplicity and effectiveness.
>
> [1] Modeling strong and human-like gameplay with KL-regularized search. ICML, 2022.
>
> > Lemma 4.1 assumes that … sampling from $\mathcal{T}_{\text{pre}}^{-1}$ should be correlated with the opponent's policy. So is the assumption reasonable?
> >
>
> Thank you for carefully reading our paper and raising this interesting question. We also hope you understand that *such a gap between theoretical analysis and practical algorithms is quite common in deep reinforcement learning.*
>
> Yes, opponent policies do influence the distribution $\mathcal{T} _ {\text{pre}}^{-1}$. Here, we assume that sampling states $s$ from the distribution $\mathcal{T} _ {\text{pre}}^{-1}$ is independent of opponent policies. This is not a strong assumption. During the pretraining stage, we generate many trajectories w.r.t. each opponent policy. Therefore, given any opponent policy $\pi^{-1}$ in $\Pi^{\text{train}}$, sampling $s$ from $\mathcal{T} _ {\text{pre}}^{-1}(·;\pi^{-1})$ can be approximately considered as sampling over the entire state space $\mathcal{S}$, thus making it independent of the opponent policy.
>
> In our practical implementation, we made every effort to approximate this assumption. For example, we generated 1000 trajectories w.r.t. each opponent policy, utilized MEP to ensure sufficient diversity among opponent policies, etc. Despite these approximations, we found that both OMIS w/o S and OMIS achieved impressive results empirically. This suggests that they are not heavily reliant on this assumption to a certain extent.
>
> > Equation 10 considers the expected return ... How does this design address the issue of value function overestimation?
> >
>
> This is an important question, and we greatly appreciate you bringing it up.
>
> Generally, when optimizing the Bellman optimality equation in RL, using the max operation for temporal difference bootstrapping can lead to maximization bias, which further causes overestimation issues. However, *our search does not use the max operation for bootstrapping*, so there should be no overestimation problem.
>
> Even if overestimation occurs, we also utilized a search-specific discount factor $\gamma _ {\text{search}}$ (see Eq. 8) to balance the negative impact of overestimation in the value function. When $\gamma _ {\text{search}}$ is small, we rely more on relatively reliable rewards to estimate the action value $\hat{Q}$ and less on the value function. Our experiments found that this approach effectively addresses the overestimation issue in the value function, resulting in stable performance improvements from the search process.
>
> All your comments have greatly contributed to the improvement of our paper. If you have any new comments, please feel free to provide them, and we will promptly address them. If you find that we have addressed your concerns, we hope you will reconsider your rating.

---

> > ### Comment · Reviewer_G1Ke · 2024-08-13
> >
> > Thanks for the response. I have read the rebuttal and will keep the current rating.

---

> > > ### Author Response · Authors · 2024-08-13
> > >
> > > Dear Reviewer,
> > >
> > > We believe that all your concerns have been addressed, and we sincerely thank you for taking the time to read our response. Furthermore, we are grateful for your appreciation and support of our paper.
> > >
> > > Best regards,
> > >
> > > All authors

---

> ### Author Response · Authors · 2024-08-11
> **Requesting feedback and any remaining concerns**
>
> Dear Reviewer,
>
> We are deeply grateful for your thoroughness and the constructive feedback you provided during the review process. Your valuable insights have helped us improve our paper, especially in validating the effectiveness of our approach when the ground-truth environment model is unavailable.
>
> We would love the opportunity to have an active discussion with you and address any remaining concerns or questions you may have. We sincerely request your feedback so that we can conduct additional experiments or make revisions to further improve our submission promptly.
>
> ---
>
> Here is a summary of our rebuttal, which we hope adequately addresses all the concerns you raised:
>
> 1. We have supplemented our rebuttal with **experimental results for OEOM under learned dynamic transitions** to address your concern regarding “**OEOM's strong dependence on the environment model**”, as detailed in the **Global Response**. These results support the effectiveness of OEOM under such conditions, and we will include them in the experimental section of the revision.
> 2. We clarified that the hyperparameters for OEOM's mixing technique are easy to find and **provided guidelines for setting them** to address your concern about “**its reliance on hyperparameter optimization”**. Specifically, *the hyperparameter $\epsilon$ only needs to be slightly smaller than the absolute value of the sparse rewards in the environment.*
> 3. To address your question about “**the reasonableness of the assumption in Lemma 4.1**”, we provided a detailed explanation of the assumption's meaning and clarified **how we conducted experiments to ensure that this assumption was met** as closely as possible.
> 4. Regarding your concern about “**the issue of value function overestimation**”, we analyzed the scenarios where value overestimation might occur and clarified that **our approach does not fall into this category**. Additionally, we explained and analyzed how OEOM's search-specific discount factor $\gamma_{\text{search}}$ mechanism helps to further avoid overestimation.

---

### Official Review · Reviewer_xSNm · 2024-07-16

**Soundness:** 2
**Presentation:** 1
**Contribution:** 2
**Rating:** 6
**Confidence:** 3

**Summary:**

The paper addresses the challenges of opponent modeling in multi-agent environments, particularly the difficulties in generalizing to unknown opponent policies. The authors propose a Opponent Modeling with In-context Search (OMIS), which combines in-context learning-based pretraining and decision-time search. OMIS utilizes a Transformer model with three components: an actor, an opponent imitator, and a critic, to enhance decision-making. The method proves to converge in opponent policy recognition and generalize well without search, while offering performance stability with search. Empirical results show that OMIS outperforms existing approaches in competitive, cooperative, and mixed environments.

**Strengths:**

Using in-context learning for opponent modeling is novel and interesting, though the proposed method is somewhat limited by the need of dynamics and the presentation is unclear.

**Weaknesses:**

- The proposed method requires transition dynamics while the baselines compared with it do not require, which makes experiments unfair. It makes more sense to take model-based approaches (e.g., MCTS) as a baseline. This may be a reference: https://arxiv.org/pdf/2305.13206.
- The presentation is unclear. Notations are unnecessarily complex and the reasoning in some paragraphs are unclear. See my questions.

**Questions:**

- Line 39: It's disconnected from the previous paragraph. How does pre-training with a transformer model deal with any of the issue abovementioned?
- Line 41: How is in-context learning related to pre-training issues?
- Line 42: How are these three components related to pre-training issues? The current version only describes how the proposed method is implemented but didn't explain "why" it should be implemented in this way. A good introduction should sell your main insight instead of describing your implementation.
- Line 48: Again, what issues are you talking about there? What are limited generalization abilities? What are good properties? What are performance instability issues? All these need explanation. I guess that you refer instability issue to the sentence in Line 38, "TFA always perform unstably when facing unknown opponents...". If so, instability performance is too ambiguous.
- Line 147: The phrase "self-agent is unable to ascertain the true policy $\bar{\pi}^{-1}$ employed by $\pi^{-1}$" reads weird. What do you mean by true policy is unclear. Isn't $\bar{\pi}^{-1}$ just a policy chosen by a non-stationary player?
- Line 158: What are D^epi and D^step?
- Line 159: $D^epi$ samples segments? It reads weird.
- Section 5.2: How many random seeds do you train? What does the error bar represent?
- Line 311, Section 5.2: From the heatmap shown in Figure 6, I don't think there are clear patterns between each opponent policy. Since this claim is only backed up by qualitative observation, I don't think you can say the results imply OMIS can represent different opponent policies. One suggestion is to run a statistical test (e.g., randomization test) to test the significance of the hypothesis on each attention vector.
- Line 319: I initially had a question, "What do you mean by using opponent actions and true RTGs as labels?" but I figured out what $\mu$ and $V$ mean by looking back to the method section. I suggest the author write the role of each math symbol before you refer to it if the definition of this math symbol is far away from the current text.

**Limitations:**

Yes, it's addresed.

---

> ### Author Rebuttal · Authors · 2024-08-05
>
> # Response to Reviewer **xSNm**
>
> Thank you very much for your valuable feedback. In response to your comments, we would like to make the following clarifications and feedback. We hope our explanations and analyses can eliminate concerns and make you find our work stronger.
>
> > the proposed method is somewhat limited by the need of dynamics
> >
>
> We include results and corresponding analyses for the version of OMIS where the dynamics are learned. Please refer to the **Global Response**.
>
> > The proposed method requires transition dynamics while the baselines compared with it do not require, which makes experiments unfair …
> >
>
> We compare our work with the paper [1] you mentioned. Please refer to the **Global Response**.
>
> [1] https://arxiv.org/pdf/2305.13206
>
> > The presentation is unclear …
> >
>
> In fact, the presentation of our paper received consistent approval from reviewers G1Ke, eKwk, and YkYd. However, as you mentioned, there is still significant room for improvement in terms of clarity. We apologize for any inconvenience this may have caused and appreciate your suggested advice. We will reorganize the logic and simplify the mathematical symbols in the revision. We will address your concerns one by one in our subsequent responses.
>
> > (1) Line 39: … How does pre-training with a transformer model deal with any of the issue abovementioned?
> (2) Line 41: How is in-context learning related to pre-training issues?
> (3) Line 42: How are these three components related to pre-training issues? …
> >
>
> Your suggestion is very pertinent; the transition in Lines 39-42 is indeed not smooth. Actually, we explained the motivation for our approach at the beginning of the methodology section (Lines 106-120). In the revision, we will move the overview of the motivation to Lines 39-42 to emphasize the rationale behind our work rather than its implementation.
>
> The in-context-learning (ICL)-based pretraining we used theoretically provides good generalization guarantees: the pretrained model can accurately recognize seen opponents and recognize unseen opponents as the most familiar seen ones to some extent. This theoretical property endows our approach with the potential to address pretraining issues effectively.
>
> For the three components: the actor is the core of the ICL-based pretraining, ensuring good generalization to handle pretraining issues. The critic and opponent imitator are indispensable modules for OMIS search during testing.
>
> > Line 48: Again, what issues are you talking about there? What are limited generalization abilities? What are good properties? What are performance instability issues? …
> >
>
> We apologize for the misunderstanding caused by our unclear expression. In the revision, we will clarify that "limited generalization abilities" refers to the lack of theoretical guarantees for generalization during the pretraining stage in existing approaches. "Good properties" refer to the characteristics of OMIS described in Theorem 4.2—accurately recognizing seen opponents and recognizing unseen opponents as the most similar to the seen ones. "Performance instability issues" refer to the problem mentioned in Line 38, as you guessed, which is also reflected in the results of our experiments, such as Figure 3.
>
> > Line 147: … What do you mean by true policy is unclear. Isn't $\bar{\pi}^{-1}$ just a policy chosen by a non-stationary player?
> >
>
> Here, $\pi^{-1}$ can be understood as the non-stationary opponent agent, while $\bar{\pi}^{-1}$ is a mnemonic representing the actual policy used by $\pi^{-1}$ at a given time, which is unknown to the self-agent. We will emphasize the distinction between the two symbols in the revision.
>
> > Line 158: What are $D^{\text{epi}}$ and $D^{\text{step}}_t$?
> >
>
> Both $D^{\text{epi}}$ and $D^{\text{step}} _ t$ are generated from the interactions between OMIS and the non-stationary opponent $\pi^{-1}$ during testing. $D^{\text{epi}} = \{(\tilde{s} _ h, \tilde{a} _ h^{-1})\} _ {h=1}^{H}$ is episode-wise in-context data, constructed similarly to the process in Appendix C, except that $(s, a^{-1})$ tuples are sampled from the most recent $C$ trajectories in which $\pi^{-1}$ participated. $D^{\text{step}} _ t = (s _ 0, a _ 0^{-1}, \dots, s _ {t-1}, a _ {t-1}^{-1})$ is step-wise in-context data. We will add these explanations to the revision.
>
> > Line 159: $D^{\text{epi}}$ samples segments? …
> >
>
> The expression here is indeed inaccurate. We mean that $D^{\text{epi}}$ is constructed by sampling several consecutive segments from the opponent's trajectories.
>
> > Section 5.2: How many random seeds do you train? What does the error bar represent?
> >
>
> We trained with 5 random seeds, as stated in the "Specific settings" paragraph of Section 5.1. The error bars represent the standard deviation of the test results across these above 5 random seeds.
>
> > Line 311, Section 5.2: … Since this claim is only backed up by qualitative observation, I don't think you can say the results imply OMIS can …
> >
>
> We incorporate your suggestion and add a quantitative analysis of attention weights learned by OMIS. Please refer to the **Global Response**.
>
> > Line 319: I initially had a question, "What do you mean by using opponent actions and true RTGs as labels?" …
> >
>
> Your suggestion is beneficial; we will emphasize the meaning of each symbol before it appears in the revision.
>
> All your comments have greatly contributed to the improvement of our paper. If you have any new comments, please feel free to provide them, and we will promptly address them. If you find that we have addressed your concerns, we hope you will reconsider your rating.

---

> > ### Comment · Reviewer_xSNm · 2024-08-08
> >
> > > We include results and corresponding analyses for the version of OMIS where the dynamics are learned. Please refer to the Global Response.
> >
> > The new result addresses my concern. Thanks.
> >
> > > We compare our work with the paper [1] you mentioned. Please refer to the Global Response.
> >
> > [1] https://arxiv.org/pdf/2305.13206
> >
> > The comparison seems to show positive results, but it'd be great if you could make the comparison more fair by comparing with [1] (or the other similar methods) in more hyperparameter choices. As you stated, SP-MCTS may be sensitive to hyperparameters, and the authors of [1] were likely to optimize their hyperparameters to the tasks evaluated in their paper. Nevertheless, the tasks evaluated in [1] and this submission are different, and your hyperparameters (stated in Appendix H.2.) might also be tuned for your tasks. One can thus question: is the performance gain of your method resulting from better hyperparameters? The answer to this question is unclear after rebuttal still.
> >
> > > In fact, the presentation of our paper received consistent approval from reviewers G1Ke, eKwk, and YkYd. However, as you mentioned, there is still significant room for improvement in terms of clarity. We apologize for any inconvenience this may have caused and appreciate your suggested advice. We will reorganize the logic and simplify the mathematical symbols in the revision. We will address your concerns one by one in our subsequent responses.
> >
> > I believe everyone has different standards for presentation. I do agree that I can roughly understand your high-level idea and motivation. However, if reading each paragraph more carefully, the reasoning is not immediately clear. As readers, we may have to fill in the gap in our reasoning, guessing what we're trying to communicate. It will lead to confusion and even inconsistent conclusions from different readers. As I believe the insight/reasoning written in the paper is as important as the results.
> >
> > > Your suggestion is very pertinent; the transition in Lines 39-42 is indeed not smooth.
> >
> > Ok, I feel Lines 106-112 is what I was looking for in the Intro. It's much better than the current text in the intro. Without this motivation, the implementation details written in intro reads like noise for readers.
> >
> > > Here, can be understood as the non-stationary opponent agent, while
> >  is a mnemonic representing the actual policy used by  at a given time, which is unknown to the self-agent. We will emphasize the distinction between the two symbols in the revision.
> >
> > I don't understand what you mean by "$\bar{\pi}^{-1}$" is a mnemonic representing the actual policy $\pi^{-1}$. Are you trying to say $\bar{\pi}^{-1}$ is an estimated $\pi^{-1}$?
> >
> > > Generalization abilities
> >
> > I read Appendix D.3 and still don't understand how it's related to the generalization abilities you're talking about. It looks like you;re proving PSOM will converge to the optimal solution, but how is it related to generalization?
> >
> > > Good properties & Performance instability
> >
> > Thanks for clarification.
> >
> > > Line 158:
> >
> > So why do you need two symbols? It seems that you can get step data from episode data.
> >
> > > Line 159:
> >
> > Got it.
> >
> > > We trained with 5 random seeds, as stated in the "Specific settings" paragraph of Section 5.1. The error bars represent the standard deviation of the test results across these above 5 random seeds.
> >
> > I'd like to see a 95% confidence interval estimated by the bootstrapping method and reporting aggregated performance (e.g., IQM and probability of improvement), as recommended in https://arxiv.org/abs/2108.13264. It will strengthen the statistical significance of the results.
> >
> > > Line 311
> >
> > Thanks, it's very helpful.
> >
> > > Line 319:
> >
> > Please do.
> >
> > Due to the additional results, I'm increasing my rating and will reconsider again my rating if my follow-up comments are addressed.

---

> > > ### Author Response · Authors · 2024-08-09
> > > **Response to your follow-up comments (1/2)**
> > >
> > > Thank you for taking the time to read our rebuttal so thoroughly and respond. This level of attention is rare in the review process, and we are deeply touched by it. We are also very glad that most of your concerns have been addressed. We will now carefully respond to each of your remaining concerns one by one.
> > >
> > > > The comparison seems to show positive results, but it'd be great if you could make the comparison more fair by comparing with [1] (or the other similar methods) in more hyperparameter choices … One can thus question: is the performance gain of your method resulting from better hyperparameters? The answer to this question is unclear after rebuttal still.
> > > >
> > >
> > > We sincerely apologize; you are right. Due to the tight rebuttal timeline, we did not have the opportunity to hyperparameter-tune SP-MCTS and only used its default hyperparameters. To ensure a fair comparison, we are currently running experiments with a thorough hyperparameter search for SP-MCTS. We will report the results to you as soon as they become available, and we kindly ask for your patience. Due to the rebuttal policy restrictions, we are not allowed to add additional figures or modify the PDF. Therefore, we plan to present the results in a markdown table, and we hope for your understanding.
> > >
> > > > (1) … As readers, we may have to fill in the gap in our reasoning, guessing what we're trying to communicate. It will lead to confusion and even inconsistent conclusions from different readers. As I believe the insight/reasoning written in the paper is as important as the results. (2) … Without this motivation, the implementation details written in intro reads like noise for readers.
> > > >
> > >
> > > I believe your point is well taken. A good paper requires rigorous and logical reasoning to provide readers with accurate understanding and insight. In our revision, we will move Lines 106-112 to the Introduction section and refine them to help readers better grasp the core idea of our work. Once again, thank you for your valuable suggestion.
> > >
> > > > I don't understand what you mean by "$\bar{\pi}^{-1}$" is a mnemonic representing the actual policy $\pi^{-1}$. Are you trying to say $\bar{\pi}^{-1}$ is an estimated $\pi^{-1}$?
> > > >
> > >
> > > In our context, $\pi^{-1}$ represents the non-stationary opponent agent. This agent can adopt a variety of policies during testing. $\bar{\pi}^{-1}$ is used to denote the actual policy employed by this agent in a particular episode. For instance, if the agent adopts the policy $\pi^{-1,1} \in \Pi^{\text{train}}$ in the first testing episode, then $\bar{\pi}^{-1}$ for the first testing episode would be $\pi^{-1,1}$.
> > >
> > > We apologize for the confusion caused by our use of $\pi^{-1}$, as it typically represents a policy in RL and game theory. In our revision, we will replace $\pi^{-1}$ with a new symbol to represent the non-stationary opponent agent, thereby improving clarity.
> > >
> > > > I read Appendix D.3 and still don't understand how it's related to the generalization abilities you're talking about. It looks like you;re proving PSOM will converge to the optimal solution, but how is it related to generalization?
> > > >
> > >
> > > In opponent modeling, generalization is typically defined as performance when facing unknown opponent policies. Existing approaches lack rigorous theoretical analysis under this definition of generalization. In Theorem 4.2, we proved that (1) PSOM can converge to the optimal solution, and in Lemma 4.1, we proved that OMIS w/o S is equivalent to PSOM. This implies that when the opponent's policy is a seen one, OMIS w/o S can accurately recognize the opponent's policy and converge to the best response against it. (2) PSOM recognizes an unseen opponent policy as the seen opponent policy with the smallest KL divergence from this unseen opponent policy and produces the best response to the recognized opponent policy. Since OMIS w/o S is equivalent to PSOM, OMIS w/o S possesses the same properties. These properties potentially provide OMIS w/o S with benefits in terms of the defined generalization, which is also validated in the experiments in the main text (see Figures 3 and 4).

---

> > > > ### Comment · Reviewer_xSNm · 2024-08-11
> > > >
> > > > > $\bar{\pi}^{-1}$
> > > >
> > > > So it is a realization of policy sampled from $\pi^{-1}$?
> > > >
> > > > > Appendix D.3
> > > >
> > > > Thanks, it's much clearer now. I'd suggest you draft a paragraph in rebuttal and detail your plan to incorporate this paragraph into the manuscript.
> > > >
> > > > > On the other hand, $\mathcal{D}^{step}_{t}$ is constructed using the opponent's trajectories up to the t-th timestep within the e-th testing episode.
> > > >
> > > > It seems that $\mathcal{D}^{step}_{t,e}$ is just a partial trajectory of the the episode, and $\mathcal{D}^{epi}_e$ is a full trajectory. I think you can just use time-slice index to index trajectories to represent partial trajectories?
> > > >
> > > > > SP-MCTS hyperparameters
> > > >
> > > > Thanks for the new results. I'm now good with the comparison with SP-MCTS.
> > > >
> > > > > IQM
> > > >
> > > > You should plot IQM aggregated over tasks as well. Also how many random seeds are used in each task?
> > > >
> > > > I think my primary concern about experiments is addressed. Though my concern about writing clarity still persists, I understand the author's idea better now after the rebuttal. I will update my rating to weak acceptance. I'm hesitating to increase the rating further because (1) how the author will make the writing clearer is unclear in the rebuttal (it'd be great if the author could provide a more detailed revision plan, though I'm not sure if that's sufficient to communicate the revision plan well) and (2) I'm not sure about the significance of this method since I'm not update-to-date on opponent modeling literature.

---

> > > > > ### Author Response · Authors · 2024-08-12
> > > > > **Response to your new comments (1/2)**
> > > > >
> > > > > Dear Reviewer,
> > > > >
> > > > > Thank you so much for taking the time to carefully read our new responses and for increasing your score. We have greatly benefited from our discussions with you, and your feedback has significantly helped us improve our work. Here, we provide detailed responses to each of your new comments.
> > > > >
> > > > > ---
> > > > >
> > > > > > So it is a realization of policy sampled from $\pi^{-1}$?
> > > > > >
> > > > >
> > > > > Yes, your understanding is correct.
> > > > >
> > > > > > I'd suggest you draft a paragraph in rebuttal and detail your plan to incorporate this paragraph into the manuscript.
> > > > > >
> > > > >
> > > > > We will add a new subsection titled "Analysis for Generalization" in Section 4 - Methodology to elaborate on the following analysis:
> > > > >
> > > > > 1. The definition of generalization in the context of opponent modeling;
> > > > > 2. The issues with generalization in existing opponent modeling work;
> > > > > 3. The relationship between the theoretical properties of our approach, OMIS, and the defined generalization.
> > > > >
> > > > > > It seems that $\mathcal{D}^{step}_{t,e}$ is just a partial trajectory of the the episode, and $\mathcal{D}^{epi}_e$ is a full trajectory. I think you can just use time-slice index to index trajectories to represent partial trajectories?
> > > > > >
> > > > >
> > > > > Yes, this is a beneficial suggestion. We will use the time-slice index to index trajectories to represent partial trajectories to simplify the notation.
> > > > >
> > > > > > You should plot IQM aggregated over tasks as well.
> > > > > >
> > > > >
> > > > > Following your suggestion, we further calculate the IQM aggregated over tasks, where "tasks" refer to different ratios of [seen:unseen] opponent policies. The results are presented in the three tables below.
> > > > >
> > > > > ### Predator Prey
> > > > >
> > > > > | Approach | Meta-PG | Meta-MAPG | MBOM | LIAM | MeLIBA | DRON | OMIS w/o S | OMIS |
> > > > > | --- | --- | --- | --- | --- | --- | --- | --- | --- |
> > > > > | IQM Aggregated over All Ratios | -150.41+11.36-11.36 | -130.69+17.54-17.54 | -89.50+20.08-17.77 | -73.19+7.57-8.95 | -111.96+15.82-18.95 | -120.92+30.89-33.88 | -93.81+20.55-19.02 | **-30.97+5.51-5.51** |
> > > > >
> > > > > ### Level-Based Foraging
> > > > >
> > > > > | Approach | Meta-PG | Meta-MAPG | MBOM | LIAM | MeLIBA | DRON | OMIS w/o S | OMIS |
> > > > > | --- | --- | --- | --- | --- | --- | --- | --- | --- |
> > > > > | IQM Aggregated over All Ratios | 0.18+0.00-0.00 | 0.21+0.00-0.00 | 0.25+0.01-0.01 | 0.22+0.00-0.00 | 0.21+0.00-0.00 | 0.19+0.00-0.00 | 0.28+0.00-0.00 | **0.52+0.01-0.01** |
> > > > >
> > > > > ### OverCooked
> > > > >
> > > > > | Approach | Meta-PG | Meta-MAPG | MBOM | LIAM | MeLIBA | DRON | OMIS w/o S | OMIS |
> > > > > | --- | --- | --- | --- | --- | --- | --- | --- | --- |
> > > > > | IQM Aggregated over All Ratios | 112.27+4.14-2.60 | 120.13+8.47-10.32 | 135.86+6.33-6.60 | 98.81+6.35-5.11 | 121.11+5.50-4.60 | 92.89+2.39-2.16 | 155.35+5.28-4.76 | **162.80+5.89-4.94** |
> > > > >
> > > > > We report IQM aggregated over all [seen:unseen] ratios in the three tables above and calculate a 95% confidence interval using the bootstrapping method. The "+" indicates the upper confidence interval, and the "-" indicates the lower confidence interval. We highlight the best results in bold.
> > > > >
> > > > > From the overall results above, it can be observed that OMIS effectively outperforms the baselines mentioned in the main text over all [seen:unseen] ratios and consistently improves upon the results of OMIS w/o S. We will include these results in our revisions, too.

---

> > > ### Author Response · Authors · 2024-08-09
> > > **Response to your follow-up comments (2/2)**
> > >
> > > > So why do you need two symbols? It seems that you can get step data from episode data.
> > > >
> > >
> > > In fact, these two symbols are different. Suppose the current testing episode index is $e$. Then, $D^{\text{epi}}$ is constructed using the opponent's trajectories from testing episodes $e-C, \ldots, e-2, e-1$. On the other hand, $D^{\text{step}}_t$ is constructed using the opponent's trajectories up to the $t$-th timestep within the $e$-th testing episode. We apologize for the confusion caused by the redundancy in our symbol definitions. We will revise and clarify our notation in the revision to improve clarity.
> > >
> > > > I'd like to see a 95% confidence interval estimated by the bootstrapping method and reporting aggregated performance (e.g., IQM and probability of improvement), as recommended in https://arxiv.org/abs/2108.13264. It will strengthen the statistical significance of the results.
> > > >
> > >
> > > Using mean as performance and standard deviation (std.) as confidence is common in RL [3,4,5,6]. However, as you mentioned, IQM and probability of improvement are more reasonable and can strengthen the statistical significance of the results [2]. Due to the rebuttal policy restrictions, we are not allowed to add additional figures or modify the PDF. We will adopt the methods you suggested and redraw the figures in the experimental section in our revision to improve the statistical significance of the results. We greatly appreciate this suggestion.
> > >
> > > [2] https://arxiv.org/abs/2108.13264
> > >
> > > [3] Addressing Function Approximation Error in Actor-Critic Methods, Fujimoto et al, 2018. RL Algorithm: TD3.
> > >
> > > [4] Scalable trust-region method for deep reinforcement learning using Kronecker-factored approximation, Wu et al, 2017. RL Algorithm: ACKTR.
> > >
> > > [5] Bridging the Gap Between Value and Policy Based Reinforcement Learning, Nachum et al, 2017. RL Algorithm: PCL.
> > >
> > > [6] Empirical Design in Reinforcement Learning, Andrew Patterson et al, 2023. JMLR
> > >
> > > Thank you once again for taking our manuscript so seriously. We sincerely hope that we have addressed your new concerns. We will promptly share the new experimental results with you as soon as they are available. We also warmly welcome any further in-depth discussions with you.

---

> > > ### Author Response · Authors · 2024-08-10
> > > **Supplementary Results (1/2)**
> > >
> > > Dear Reviewer, thank you once again for carefully reading our rebuttal and providing useful suggestions. Here, we are providing some of the experimental results you requested, and we hope these will address your concerns.
> > >
> > > ## **1 Results against SP-MCTS with hyperparameter search**
> > >
> > > We conduct a hyperparameter search for SP-MCTS's $c_{\text{puct}}$ to facilitate a more fair comparison. The results are presented in the three tables below.
> > >
> > > ### Predator Prey
> > >
> > > | Approach | [seen:unseen]=10:0 | [seen:unseen]=10:5 | [seen:unseen]=10:10 | [seen:unseen]=5:10 | [seen:unseen]=0:10 |
> > > | --- | --- | --- | --- | --- | --- |
> > > | OMIS w/o S | $\textcolor{blue}{-53.93+1.95-1.16}$ | $-\textcolor{blue}{56.79+2.04-1.53}$ | $\textcolor{blue}{-69.62+3.47-2.71}$ | $\textcolor{blue}{-137.10+8.13-10.68}$ | $\textcolor{blue}{-117.43+4.41-3.25}$ |
> > > | SP-MCTS ($c_{\text{puct}}$=0.5) | $\textcolor{red}{-46.29+1.52-4.01}$ | -69.08+3.86-3.76 | -72.37+3.31-4.81 | $\textcolor{red}{-58.30+1.79-3.33}$ | $\textcolor{red}{-86.15+3.25-1.70}$ |
> > > | SP-MCTS ($c_{\text{puct}}$=1.0) | -52.57+0.48-0.50 | $\textcolor{red}{-58.56+3.35-4.67}$ | $\textcolor{red}{-56.91+3.68-2.69}$ | -103.32+9.74-5.15 | -97.07+2.44-2.17 |
> > > | SP-MCTS ($c_{\text{puct}}$=1.2) | -54.32+2.46-4.00 | -62.75+2.29-2.16 | -60.58+2.85-2.36 | -131.77+2.37-4.67 | -102.06+2.04-5.72 |
> > > | SP-MCTS ($c_{\text{puct}}$=2.0) | -55.52+1.88-3.11 | -89.72+1.81-9.72 | -94.84+2.98-0.89 | -85.95+1.17-5.74 | -108.19+1.39-3.46 |
> > > | SP-MCTS ($c_{\text{puct}}$=5.0) | -59.02+1.56-1.53 | -86.18+3.93-4.39 | -72.02+3.10-2.88 | -79.54+3.92-3.40 | -113.90+3.05-3.99 |
> > > | OMIS | **-25.46+0.34-1.06** | **-24.41+2.01-1.17** | **-29.12+0.36-0.96** | **-34.71+2.56-2.73** | **-37.76+1.31-0.62** |
> > >
> > > ### Level-Based Foraging
> > >
> > > | Approach | [seen:unseen]=10:0 | [seen:unseen]=10:5 | [seen:unseen]=10:10 | [seen:unseen]=5:10 | [seen:unseen]=0:10 |
> > > | --- | --- | --- | --- | --- | --- |
> > > | OMIS w/o S | $\textcolor{blue}{0.28+0.00-0.01}$ | $\textcolor{blue}{0.27+0.01-0.01}$ | $\textcolor{blue}{0.28+0.01-0.01}$ | $\textcolor{blue}{0.28+0.01-0.01}$ | $\textcolor{blue}{0.27+0.01-0.00}$ |
> > > | SP-MCTS ($c_{\text{puct}}$=0.5) | $\textcolor{red}{0.41+0.01-0.02}$ | $\textcolor{red}{0.40+0.01-0.01}$ | $\textcolor{red}{0.41+0.01-0.02}$ | $\textcolor{red}{0.40+0.01-0.01}$ | $\textcolor{red}{0.40+0.01-0.01}$ |
> > > | SP-MCTS ($c_{\text{puct}}$=1.0) | 0.36+0.01-0.01 | 0.36+0.00-0.00 | 0.37+0.01-0.01 | 0.37+0.00-0.00 | 0.36+0.01-0.01 |
> > > | SP-MCTS ($c_{\text{puct}}$=1.2) | 0.34+0.00-0.00 | 0.34+0.01-0.01 | 0.34+0.00-0.00 | 0.34+0.01-0.01 | 0.34+0.00-0.01 |
> > > | SP-MCTS ($c_{\text{puct}}$=2.0) | 0.31+0.01-0.01 | 0.30+0.01-0.00 | 0.31+0.01-0.01 | 0.31+0.00-0.00 | 0.31+0.00-0.00 |
> > > | SP-MCTS ($c_{\text{puct}}$=5.0) | 0.28+0.00-0.00 | 0.28+0.00-0.01 | 0.28+0.01-0.00 | 0.28+0.01-0.00 | 0.27+0.01-0.00 |
> > > | OMIS | **0.51+0.01-0.00** | **0.51+0.01-0.01** | **0.52+0.01-0.00** | **0.52+0.01-0.01** | **0.51+0.01-0.00** |
> > >
> > > ### OverCooked
> > >
> > > | Approach | [seen:unseen]=10:0 | [seen:unseen]=10:5 | [seen:unseen]=10:10 | [seen:unseen]=5:10 | [seen:unseen]=0:10 |
> > > | --- | --- | --- | --- | --- | --- |
> > > | OMIS w/o S | $\textcolor{blue}{164.49+1.60-1.29}$ | $\textcolor{blue}{160.99+2.87-1.59}$ | $\textcolor{blue}{157.34+3.56-2.04}$ | $\textcolor{blue}{149.40+4.12-1.54}$ | $\textcolor{blue}{146.87+2.42-3.68}$ |
> > > | SP-MCTS ($c_{\text{puct}}$=0.5) | 89.83+27.76-33.50 | 92.63+27.45-29.54 | 82.02+30.18-29.40 | 69.96+30.06-30.06 | 71.24+30.15-23.86 |
> > > | SP-MCTS ($c_{\text{puct}}$=1.0) | 134.82+7.77-10.76 | 133.02+8.37-14.51 | 131.11+7.12-11.60 | 130.30+7.88-14.37 | 119.94+8.23-17.33 |
> > > | SP-MCTS ($c_{\text{puct}}$=1.2) | $\textcolor{red}{144.59+2.92-4.51}$ | $\textcolor{red}{140.97+3.11-3.97}$ | 137.19+2.50-3.12 | 130.04+3.46-1.96 | 130.62+1.95-5.31 |
> > > | SP-MCTS ($c_{\text{puct}}$=2.0) | 143.43+5.18-6.01 | 136.27+4.42-3.37 | $\textcolor{red}{137.28+6.03-6.06}$ | $\textcolor{red}{133.36+5.79-3.74}$ | $\textcolor{red}{133.54+4.30-6.15}$ |
> > > | SP-MCTS ($c_{\text{puct}}$=5.0) | 128.16+3.93-3.61 | 126.41+4.46-3.31 | 126.11+4.15-4.36 | 124.13+4.47-3.92 | 122.59+5.05-4.43 |
> > > | OMIS | **172.15+1.95-0.30** | **162.21+1.04-1.38** | **162.13+0.51-0.82** | **162.33+1.46-1.24** | **155.08+1.63-1.09** |
> > >
> > > Following your suggestion, we report the aggregated performance in the three tables above using IQM and calculate a 95% confidence interval using the bootstrapping method. The "+" indicates the upper confidence interval, and the "-" indicates the lower confidence interval. The true transition dynamics are available to all approaches, and all SP-MCTS instances use OMIS w/o S as the blueprint policy. We mark the results of OMIS w/o S in blue, highlight the results of OMIS in bold, and mark the best SP-MCTS results across all hyperparameters in red.

---

> > > ### Author Response · Authors · 2024-08-10
> > > **Supplementary Results (2/2)**
> > >
> > > Upon observation, we found that after conducting a hyperparameter search for SP-MCTS, we still arrive at conclusions similar to those drawn previously: OMIS can effectively outperform the SP-MCTS with the best hyperparameter and more effectively improve OMIS w/o S. The SP-MCTS with the best hyperparameter can sometimes effectively improve OMIS w/o S (e.g., in LBF and most of PP), while in other cases, it even makes OMIS w/o S worse (e.g., in OC and [seen:unseen]=10:5 of PP). We will include these experimental results and the relevant analysis in our revision.
> > >
> > > ## **2 Main results reported with aggregated performance and 95% confidence interval**
> > >
> > > Following your suggestion, we calculate the aggregated performance using IQM and calculate a 95% confidence interval using the bootstrapping method. The results are presented in the three tables below.
> > >
> > > ### Predator Prey
> > >
> > > | Approach | [seen:unseen]=10:0 | [seen:unseen]=10:5 | [seen:unseen]=10:10 | [seen:unseen]=5:10 | [seen:unseen]=0:10 |
> > > | --- | --- | --- | --- | --- | --- |
> > > | Meta-PG | -131.43+5.29-2.41 | -165.89+8.77-5.87 | -143.96+8.30-5.34 | -144.48+8.25-3.90 | -166.29+11.40-8.38 |
> > > | Meta-MAPG | -105.46+9.59-5.32 | -142.77+24.59-17.94 | -150.55+19.84-11.80 | -106.05+10.78-7.83 | -148.62+23.44-14.19 |
> > > | MBOM | -72.35+16.19-14.64 | -57.71+6.61-4.83 | -98.55+30.49-23.51 | -100.23+28.19-20.64 | -118.66+35.96-24.73 |
> > > | LIAM | -61.20+1.54-2.19 | -67.55+2.60-1.99 | -77.54+2.63-2.62 | -70.61+2.60-1.56 | -89.05+9.29-8.44 |
> > > | MeLIBA | -91.46+10.15-3.54 | -122.60+20.80-8.10 | -100.03+10.52-5.56 | -98.88+10.25-3.87 | -146.82+22.86-5.11 |
> > > | DRON | -88.38+14.84-3.47 | -83.53+6.21-5.38 | -179.62+40.94-2.53 | -106.33+16.86-4.90 | -146.72+24.97-2.12 |
> > > | OMIS w/o S | -53.93+1.95-1.16 | -87.07+2.36-8.24 | -91.98+4.12-3.37 | -118.65+6.93-4.79 | -117.43+4.42-3.56 |
> > > | OMIS | **-25.46+0.34-1.06** | **-22.75+1.36-0.84** | **-30.88+1.39-1.01** | **-37.98+1.63-1.43** | **-37.76+1.30-0.62** |
> > >
> > > ### Level-Based Foraging
> > >
> > > | Approach | [seen:unseen]=10:0 | [seen:unseen]=10:5 | [seen:unseen]=10:10 | [seen:unseen]=5:10 | [seen:unseen]=0:10 |
> > > | --- | --- | --- | --- | --- | --- |
> > > | Meta-PG | 0.18+0.00-0.01 | 0.17+0.01-0.00 | 0.18+0.00-0.00 | 0.17+0.01-0.01 | 0.18+0.01-0.01 |
> > > | Meta-MAPG | 0.21+0.00-0.01 | 0.21+0.01-0.01 | 0.21+0.01-0.00 | 0.21+0.01-0.01 | 0.21+0.01-0.00 |
> > > | MBOM | 0.25+0.01-0.01 | 0.24+0.02-0.01 | 0.25+0.02-0.01 | 0.25+0.01-0.01 | 0.26+0.00-0.01 |
> > > | LIAM | 0.22+0.01-0.01 | 0.22+0.01-0.02 | 0.22+0.01-0.01 | 0.22+0.01-0.01 | 0.21+0.01-0.01 |
> > > | MeLIBA | 0.21+0.01-0.01 | 0.20+0.02-0.02 | 0.20+0.01-0.02 | 0.21+0.01-0.02 | 0.21+0.01-0.01 |
> > > | DRON | 0.19+0.01-0.02 | 0.19+0.01-0.02 | 0.20+0.01-0.02 | 0.20+0.01-0.02 | 0.20+0.01-0.02 |
> > > | OMIS w/o S | 0.28+0.00-0.01 | 0.28+0.02-0.01 | 0.28+0.01-0.01 | 0.28+0.01-0.01 | 0.27+0.01-0.00 |
> > > | OMIS | **0.51+0.01-0.00** | **0.51+0.01-0.01** | **0.52+0.01-0.01** | **0.52+0.01-0.01** | **0.51+0.01-0.00** |
> > >
> > > ### OverCooked
> > >
> > > | Approach | [seen:unseen]=10:0 | [seen:unseen]=10:5 | [seen:unseen]=10:10 | [seen:unseen]=5:10 | [seen:unseen]=0:10 |
> > > | --- | --- | --- | --- | --- | --- |
> > > | Meta-PG | 108.92+8.56-32.52 | 112.78+21.57-28.72 | 118.19+5.04-35.43 | 113.30+17.30-24.67 | 111.49+16.40-24.86 |
> > > | Meta-MAPG | 130.56+6.10-41.65 | 138.19+18.85-36.37 | 135.22+17.86-24.35 | 114.44+10.31-27.64 | 125.65+16.95-27.60 |
> > > | MBOM | 146.04+4.58-49.60 | 139.84+13.54-44.60 | 136.09+5.46-42.61 | 117.12+26.29-21.94 | 132.92+9.61-30.41 |
> > > | LIAM | 94.52+27.07-24.09 | 84.22+33.41-23.48 | 88.72+27.84-23.44 | 104.53+25.59-31.93 | 102.19+22.64-18.54 |
> > > | MeLIBA | 116.77+30.64-57.55 | 117.76+24.36-63.22 | 107.02+27.12-39.25 | 109.21+20.22-43.26 | 119.21+19.07-51.22 |
> > > | DRON | 89.50+52.95-35.34 | 95.18+56.16-39.51 | 101.53+49.62-38.94 | 89.25+60.44-39.19 | 93.34+51.33-35.99 |
> > > | OMIS w/o S | 164.49+1.60-1.29 | 163.76+4.10-1.50 | 152.10+4.37-3.00 | 155.20+3.78-1.18 | 146.87+2.42-3.68 |
> > > | OMIS | **172.15+1.95-0.28** | **172.42+1.65-1.45** | **157.77+1.23-0.97** | **165.01+1.42-0.92** | **155.08+1.62-1.07** |
> > >
> > > In these tables, "+" indicates the upper confidence interval, and "-" indicates the lower confidence interval. These three tables are recalculated using the data from the experimental setup described in Figure 3 of the main text. We highlight the best results in bold.
> > >
> > > With these new statistical methods, we observe that OMIS still effectively outperforms most of the baselines mentioned in the main text and consistently improves upon the results of OMIS w/o S. In our revision, we will redraw all the figures using aggregated performance and the 95% confidence interval to strengthen the statistical significance of the results.
> > >
> > > We look forward to further discussions with you and sincerely hope that you will reconsider your score.

---

> ### Author Response · Authors · 2024-08-12
> **Response to your new comments (2/2)**
>
> > (1) how the author will make the writing clearer is unclear in the rebuttal (it'd be great if the author could provide a more detailed revision plan, …)
> >
>
> We carefully review your feedback and summarize *the areas in our initial version where clarity was lacking*:
>
> 1. Lines 39-41: We did not clearly explain how in-context learning addresses the pretraining issues mentioned in Lines 33-36; Line 45: We failed to clarify the roles and problem-solving contributions of the three components mentioned.
> 2. Line 45: We used some ambiguous phrases, such as "limited generalization abilities," "good properties," and "performance instability issues," without providing adequate explanations for them.
> 3. Line 147: We introduced two symbols, $\bar{\pi}^{-1}$ and $\pi^{-1}$, which are easily confused and difficult to understand.
> 4. Lines 148-159: We used somewhat redundant symbols $D^{\text{epi}}$ and $D^{\text{step}}_t$ without clearly explaining their specific meanings.
>
> To address these unclear areas, we have developed a **detailed revision plan**:
>
> 1. We will simplify Lines 106-119, move them before the original Line 39, and then optimize the logic to ensure a smoother connection with the surrounding paragraphs.
> 2. In Lines 32-38, we will clearly define "limited generalization abilities" and "performance instability issues" before referencing them in the original Line 45 to avoid ambiguity. In Line 50, we will provide an objective description of OMIS's theoretical properties, avoiding the use of vague terms like "good properties."
> 3. We will introduce new symbols to represent the non-stationary opponent agent, replacing $\pi^{-1}$ to prevent confusion with $\bar{\pi}^{-1}$.
> 4. We will follow your suggestion to use time-slice indices to represent partial trajectories, simplifying the notation for in-context data.
>
> Additionally, through our discussions with you, we've learned a lot from your excellent taste in writing and *identified other areas that could be improved*:
>
> 1. Line 122: We did not clearly explain how training against different opponent policies to learn BRs ensures that the actor acquires high-quality knowledge for responding to various opponents.
> 2. Line 120: Although we provided a schematic diagram and pseudocode for OMIS, the specific process of the OMIS remains unclear.
>
> For these areas, we have also developed the following **revision plans**:
>
> 1. We will briefly overview the main idea and process of ICL-based Pretraining before Line 122 to clarify the motivation behind Line 122.
> 2. We will provide a step-by-step summary of the overall OMIS process in Line 120, using Figure 1 as a reference, and refine the logic to make the approaches described in Sections 4.1 and 4.2 clearer.
>
> We will diligently implement these revision plans.
>
> > (2) I'm not sure about the significance of this method since I'm not update-to-date on opponent modeling literature.
> >
>
> At the **methodological level**, our work is built on an up-to-date and extensive overview of existing work, which reviewers like G1Ke and YkYd have appreciated. For instance, G1Ke mentioned, "The paper provides a comprehensive review of existing literature, showcasing a deep understanding of the relevant background." We logically categorized the various existing approaches into two major categories (PFA and TFA) and analyzed the issues associated with them. Based on this analysis, we proposed a new algorithm, OMIS, which effectively addresses the specific problems inherent in these two categories of approaches.
>
> At the **theoretical level**, most existing work lacks theoretical guarantees regarding generalization. In contrast, we have proven that our approach possesses the following properties: (1) When the opponent's policy is a seen one, OMIS w/o S can accurately recognize the opponent's policy and converge to the best response against it; When the opponent's policy is an unseen one, OMIS w/o S recognizes the opponent policy as the seen opponent policy with the smallest KL divergence from this unseen opponent policy and produces the best response to the recognized opponent policy. (2) OMIS's search is guaranteed to improve OMIS w/o S,  without any gradient updates.
>
> At the **experimental level**, we selected a sufficient number of representative opponent modeling baselines from both the PFA and TFA categories. OMIS consistently outperformed these baselines, regardless of whether the opponents were seen or unseen. Our supplementary experiments confirmed that OMIS can also work effectively when the transition dynamics are unknown and learned.
>
> In summary, whether at the methodological, theoretical, or experimental level, we believe our work has significantly contributed to advancing the domain of opponent modeling.
>
> ---
>
> Again, Thank you for your extensive and valuable comments, which have greatly improved our paper. We hope we have addressed all your concerns. We look forward to your continued feedback and further support for our work.

---

### Author Rebuttal · Authors · 2024-08-05

# Global Response

We extend our heartfelt thanks to AC and all the reviewers for your diligent work in evaluating our manuscript. We are deeply grateful for the insightful feedback and recommendations from each of you. In this global rebuttal comment, we provide additional experimental results and relevant analysis requested by all reviewers. The figures and tables for the results are all in the attached PDF.

## **1 (to all reviewers) Results of OMIS using learned dynamics**

To relax OMIS's reliance on ground truth transition dynamics, we implement a Model-Based OMIS (MBOMIS) to test whether OMIS can work effectively when the environment model is unknown and learned. We use the most straightforward method to learn a transition dynamic model $\hat{\mathcal{P}}$: given a state $s$ and action $a$, predicting the next state $s'$ and reward $r$ using Mean Square Error (MSE) loss. We train $\hat{\mathcal{P}}$ using the $(s, a, r, s')$ tuples from the dataset used for pretraining OMIS w/o S.

The test results against unknown non-stationary opponents are shown in **Figure 1 of the PDF**. Due to the rebuttal time constraints, we initially provided results for the PP and LBF environments. We will supplement the OC results in future revisions. Observing **Figure 1(a) of the PDF**, we find that although MBOMIS loses some performance compared to OMIS, it still effectively improves over OMIS w/o S and generally surpasses other baselines. **Figure 1(b) of the PDF** shows a similar phenomenon, where MBOMIS, despite not reaching the level of OMIS, effectively adapts to each true policy employed by the opponents compared to other baselines.

We also provide quantitative evaluation results of the learned dynamic $\hat{\mathcal{P}}$'s estimation during testing in **Table 1 of the PDF**. Observations show that $\hat{\mathcal{P}}$ generally has a relatively small MSE value in predicting the next state and reward (without normalization). Although the reward error on PP and the next state error on LBF are relatively larger, the results in **Figure 1 of the PDF** indicate that this does not significantly negatively impact the search.

## **2 (to reviewer xSNm) Results against a new search-based baseline**

We add a search-based baseline, SP-MCTS [1], to make the experiments more comprehensive. The results are shown in **Figure 2 of the PDF**. In this experiment, the true transition dynamics are available to all approaches, and SP-MCTS uses OMIS w/o S as the blueprint policy. The results indicate that OMIS can effectively outperform SP-MCTS and more effectively improve OMIS w/o S. SP-MCTS can sometimes effectively improve OMIS w/o S (e.g., in LBF and parts of PP), while in other cases, it even makes OMIS w/o S worse (e.g., in OC and parts of PP).

We presume that this is because (1) Compared to OMIS, SP-MCTS's critic and opponent model do not adaptively adjust based on the opponent's information (in-context data). Additionally, we suspect that its performance could deteriorate further if a fixed policy, rather than OMIS w/o S, were used as the blueprint policy. (2) The trade-off between exploration and exploitation in the MCTS used by SP-MCTS heavily relies on the hyperparameter $c_{\text{puct}}$. It requires extensive hyperparameter tuning to function effectively. With default hyperparameters, it may sometimes (e.g., in OC) focus too much on exploration and insufficiently on exploitation, leading to a worse policy.

[1] https://arxiv.org/pdf/2305.13206

## **3 (to reviewer xSNm) Quantitative analysis of attention weights learned by OMIS**

To rigorously evaluate whether OMIS can effectively characterize opponent policies, we conduct a quantitative analysis of the attention weights learned by OMIS by calculating the pair-wise Pearson correlation coefficients (PCC) between the attention vectors. The relevant results are shown in **Figure 4 of the PDF**. The first row is the heatmaps of the pair-wise PCC statistics of all attention vectors, and the second row shows the corresponding p-value plots for the statistics in the first row, with pairs marked in white for $p < 0.05$ and black otherwise.

The observations reveal that the attention vectors of the same opponent policy have *strong pair-wise correlations* (statistics close to $1$ and $p < 0.05$) across multiple timesteps. In contrast, the attention vectors of different opponent policies generally have no strong pair-wise correlations with each other. Although there is some pair-wise correlation between the attention vectors of different opponent policies, each opponent policy generally has the strongest pair-wise correlation with its own other attention vectors. These observations indicate that the attention weights learned by OMIS can be distinguished by different opponent policies and maintain consistency for the same opponent policy to some extent. Therefore, this analysis further demonstrates OMIS's ability to represent opponent policies based on in-context data.

## **4 (to reviewer YkYd) Results of OMIS-dyna under all ratio settings**

We supplement the results of OMIS-dyna under all [seen:unseen] opponent ratio settings, as shown in **Figure 3 of the PDF**. Observations reveal that despite the unpredictable frequency of opponent policy switches, OMIS can generally achieve good adaptation results. Additionally, the results of OMIS-dyna are close to those of *OMIS-20* (where the opponent switches policies every 20 episodes, results shown in **Figure 2 of the PDF**). In both PP and OC, there is a slight performance decline as the proportion of unseen opponents increases, likely due to the out-of-distribution behavior of the unseen opponent policies.

Should there be any further questions or concerns after reviewing our responses, we welcome continued discussions in the forthcoming phase. Your ongoing engagement is greatly valued and appreciated. Thank you again for your time, expertise, and contributions to our paper.

---

### Author Response · Authors · 2024-08-14
**Rebuttal Summary and Appreciation**

Dear Reviewers, ACs, SACs, and PCs,

We sincerely thank you for your hard work throughout the review and rebuttal process. We deeply appreciate the constructive and insightful comments provided by the reviewers after carefully reading our manuscript, which have greatly improved the quality of our work. We have also thoroughly enjoyed the in-depth discussions with all the reviewers.

---

Here, we provide a brief summary of the entire rebuttal process:

- Through the rebuttal process, we *addressed most of the reviewers' concerns*. For example:
    - Reviewer **xSNm** said: “I understand the author's idea better now after the rebuttal”
    - Reviewer **eKwk** stated: “Thank you again for addressing my questions. They have been largely answered.”
- During the rebuttal period, we supplemented our paper with additional experimental results based on the reviewers' feedback to further validate the effectiveness of our proposed OMIS (see **Global Response** for details). These new results were **unanimously recognized by the reviewers**. For example:
    - Reviewer **xSNm** said: “I think my primary concern about experiments is addressed.”
    - Reviewer **YkYd** stated: “I appreciate the efforts for conducting these many experiments requested by me and other reviewers.”
- The **novelty of our work was consistently appreciated** by reviewers **xSNm**, **G1Ke**, and **eKwk**, for which we are sincerely grateful. For example:
    - Reviewer **xSNm** said: “Using in-context learning for opponent modeling is novel and interesting”
    - Reviewer **G1Ke** stated: “The proposed method is meticulously designed and clearly articulated”
    - Reviewer **eKwk** said: “Novel combination of known components”
- The extensive and detailed discussions with reviewer **xSNm** were particularly enriching and embodied the kind of pure, meaningful academic point of view exchange, that we deeply appreciate and enjoy. We learned a great deal from our discussions with reviewer **xSNm**, including adopting a more rigorous and logical writing style and using more advanced methods to evaluate experimental results, thereby strengthening the statistical significance of the outcomes.
- Reviewer **YkYd** expressed some remaining concerns regarding the novelty and scalability of our work. We greatly appreciate his sincere and important viewpoints, and we are thankful for the opportunity to further clarify the advantages of our approach in these areas. *We provide detailed responses to address these concerns and respectfully invite the other reviewers and AC to read our discussion with Reviewer **YkYd** for a deeper understanding of our work.* Here is a summary of our response:
    - ***Regarding novelty***, our paper offers **a completely new perspective on opponent modeling** and introduces **an entirely new algorithmic framework** with **novel theoretical properties**. Our work provides an innovative understanding that ***opponent modeling is fundamentally a sequence-to-sequence problem***. **The input sequence consists of the historical data generated from interactions with the opponent, which we refer to as in-context data, while the output sequence represents the optimal sequence of actions that the self-agent needs to take.** Meanwhile, our work is the ***first to propose the concept of in-context search*, which is not only novel in the opponent modeling domain but also introduces a new methodological paradigm in the reinforcement learning field.**
    - ***Regarding scalability**,* **our approach was designed with scalability as a key consideration from the outset**, as we perfectly align with the principles outlined in **Richard S. Sutton**'s "***The Bitter Lesson***": ***One thing that should be learned from the bitter lesson is the great power of general purpose methods, of methods that continue to scale with increased computation even as the available computation becomes very great. The two methods that seem to scale arbitrarily in this way are search and learning.*** Our approach only requires **learning** a Transformer model and then performing a **search** based on it, making it ***essentially scalable***.

---

In the end, we thank you again for your time and dedication to the review process. We enjoy the rebuttal and discussion with all the reviewers. We appreciate that we receive many valuable feedbacks from the reviewers. *We believe our simple and effective approach, OMIS, provides a new perspective and a scalable way to solve complex problems in the opponent modeling & reinforcement learning community and contributes positively to the NeurIPS 2024 conference.* We hope that our work will earn your further support. We fully respect and support all your future judgment on our paper. Thank you very much.

Best regards,

All authors

---

### Decision · Program_Chairs · 2024-09-25

**Decision:**

Accept (poster)

**Comment:**

This paper proposes opponent modeling with in-context search (OMIS), where a transformer-based policy is pretrained and enhanced during the test by planning. Empirical results show OMIS outperforms pretraining-focused and testing-focused baselines.

In the initial review, most reviewers have concerns about the assumption of truth transition dynamics. During rebuttal, the experiment of OMIS based on the learned model was conducted. The performance of OMIS is still better than the baselines. This clarifies this concern.

There are still some concerns about the novelty and scalability after the rebuttal. For scalability, I agree that the paper will be stronger if it is additionally evaluated in more complex games than current environments.

Currently, I think the merits outweigh the weaknesses. The authors should include the results of OMIS with the learned model in the final version as well as promised modifications.